# CAReDiO: Enhancing Cultural Alignment of LLM via Representativeness and Distinctiveness Guided Data Optimization

## Abstract

As Large Language Models (LLMs) more deeply integrate into human life across various regions, aligning them with pluralistic cultures is crucial for improving user engagement and mitigating cultural conflicts. For this purpose, recently, different culture-specific corpora have been carefully curated, either synthesized or manually annotated. Nevertheless, inspired by culture theories, we identify two key challenges faced by these datasets: (1) *Representativeness*: These corpora fail to fully capture the target culture's core characteristics, causing insufficient cultural coverage with redundancy; (2) *Distinctiveness*: They struggle to distinguish the unique nuances of a given culture from shared patterns across other relevant ones, hindering precise cultural modelling. To handle these challenges, we introduce **CAReDiO**, a novel data optimization framework, which alternatively refines culture-sensitive questions and responses according to information-theoretic objectives in an in-context optimization manner, enhancing the cultural representativeness and distinguishability of constructed data. Extensive experiments on 15 distinct cultures demonstrate that CAReDiO can create high-quality data with richer cultural information and enable efficient alignment of small open-source or large proprietary LLMs with as few as 200 training samples, consistently outperforming previous datasets in both multi-choice and open-ended cultural benchmarks.

## 1 Introduction

As Large Language Models (LLMs) are widely integrated into human life (Bubeck et al., 2023; OpenAI, 2024; Dubey et al., 2024; Guo et al., 2025), aligning them with human values is imperative to mitigate safety risks and further improve user engagement (Ouyang et al., 2022; Wang et al., 2024b). Focusing on universal values, *e.g*, the HHH principle (Askell et al., 2021; Bai et al., 2022), most prior studies overlook the *cultural diversity* rooted in human values. As a result, LLMs trained on the corpus dominated by data that is created by English speakers and expresses Western opinions, are often biased towards Western cultures (Cao et al., 2023; Durmus et al., 2023), dissatisfying underrepresented cultural communities and raising unintended social tensions (Ryan et al., 2024). Therefore, *aligning LLMs with diverse and distinct cultural values has become both an ethical and practical necessity* (AlKhamissi et al., 2024; Tao et al., 2024).

Early efforts align LLMs with target cultures using an In-Context Learning (ICL) approach, via role-playing instructions, native-language prompts, or few-shot examples (Durmus et al., 2023; Cao et al., 2023; Kwok et al., 2024). They suffer from inconsistent performance across tasks, especially for small models, as well as expensive inference cost and privacy concerns (Saunders et al., 2022). Recently, fine-tuning culture-aware LLMs has proven a practical alternative (Li et al., 2024b). Large-scale local-language corpora are used to produce regional LLMs (Gupta et al., 2023; Nguyen et al., 2023b; Pipatanakul et al., 2023), yet language alone does not sufficiently capture cultural values (Choenni et al., 2024; Mukherjee et al., 2024; Rystrøm et al., 2025). A more precise avenue is to build dedicated, culture-specific datasets (Fung et al., 2024; Shi et al., 2024; Li et al., 2024a;b), either synthesized or manually annotated, while high-quality data demands massive annotation cost and is hard to scale.

Following this line, we ask *can we achieve cultural alignment at minimal cost by using fewer but more effective data?* To answer this question, we investigate culture theories such as the emic-etic the-

ory (Triandis et al., 1990; Hofstede & Hofstede, 2005; Miyamoto et al., 2018; Fiske & Taylor, 2020; Mostowlansky & Rota, 2020), which argues that fully understanding a culture requires two complementary perspectives: *an internal (emic) view*, capturing the highly shared beliefs and practices that bind its members, and *an external (etic) view*, highlighting the traits that differentiate itself from others.

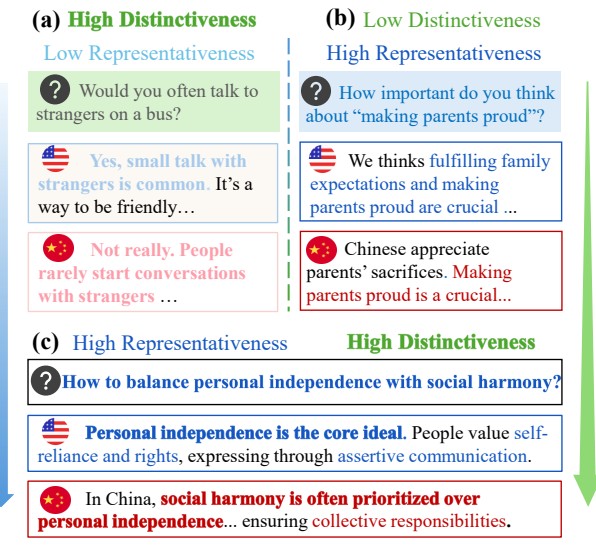

**(a) High Distinctiveness**
Low Representativeness

❓ Would you often talk to strangers on a bus?

🇺🇸 Yes, small talk with strangers is common. It's a way to be friendly…

⭐ Not really. People rarely start conversations with strangers …

**(b) Low Distinctiveness**
High Representativeness

❓ How important do you think about "making parents proud"?

🇺🇸 We thinks fulfilling family expectations and making parents proud are crucial ...

⭐ Chinese appreciate parents' sacrifices. Making parents proud is a crucial...

**(c)** High Representativeness **High Distinctiveness**

❓ How to balance personal independence with social harmony?

🇺🇸 **Personal independence is the core ideal.** People value self-reliance and rights, expressing through assertive communication.

⭐ In China, **social harmony is often prioritized over personal independence**... ensuring collective responsibilities.

Figure 1: Current data fails to fully capture both dimensions: (a) high distinctiveness (different answers in China and US) but low representativeness; (b) high representativeness (popular cultural topic) but low distinctiveness; (c) a desired example scoring high in both dimensions.

However, current datasets encounter two key challenges in reflecting both views: **Challenge 1. Representativeness**: The dataset should accurately include samples that reflect the most salient and central aspects of the target culture, rather than peripheral, less important or redundant cases (emic); **Challenge 2. Distinctiveness**: They should also capture the unique nuances of the target culture, instead of patterns shared across multiple related cultures (etic, *e.g.*, China, Japan and Korea). As shown in Fig. 1, these two dimensions are not always naturally tied together. Failing to handle these challenges hinders the precise and efficient modelling of culture-spsecific stimuli and preferences, hence hurting the efficiency and effectiveness of cultural alignment.

This work proposes CAReDiO[1], a novel LLM-empowered in-context data optimization framework for automatic cultural data construction. CAReDiO alternately generates and refines cultural questions and responses to fulfill two information-theoretic objectives: i) an *information gain objective*, inspired by the Cultural Consensus Theory (Weller, 2007), to identify data samples that better reduce the LLM's cultural uncertainty and elicit more consensus, *improving representativeness*; ii) a *culture divergence objective*, grounded in the Cognitive Conflict Theory (Limón, 2001), to enhance samples' distinguishability from non-target cultures, which theoretically performs a point-wise optimization of different cultures' JS divergence, *achieving distinctiveness*. In this way, CAReDiO can utilize any given LLMs, either the smaller open-sourced one to be aligned, *e.g.*, Llama-3.1-8B-Instruct, or a separate larger one like GPT-4o, to automatically produce more representative and distinctive data for any specific culture (shown in Fig, 2). Experiments show such data could better capture culture boundaries and enable effective alignment across diverse cultures.

Our contributions are three-fold: (1) We are the first to investigate the *representativeness* and *distinctiveness* challenges in cultural alignment data, motivated by culture theories. (2) We propose CAReDiO, an effective data optimization framework with two novel information-theoretic objectives, to tackle these challenges. (3) Using CAReDiO, we create the CARDSetcovering 15 cultures, and manifest our method can achieve better alignment across LLM backbones and under both multi-choice and open-ended benchmarks, showing superiority to larger and even manually-curated datasets.

## 2 RELATED WORK

Cultural alignment refers to adapting LLMs so that they better align with the nuanced values of diverse cultural communities. Existing research has focused on (1) evaluating culture awareness of LLMs and (2) developing methods for enhancing cultural alignment.

**Evaluation of Culture Awareness** Culture, as defined in (Adilazuarda et al., 2024), encompasses values, social norms, interpersonal behaviors and customs, etc, around which benchmarks are constructed. Many studies adopt well-established questionnaires from social sciences to analyze cultural

---

[1]**C**ultural **A**lignment via **Re**presentativeness and **Di**stinctivenss **O**ptimization.

values, such as the World Value Survey (WVS) (AlKhamissi et al., 2024), Hofstede framework (Cao et al., 2023; Masoud et al., 2023; Kharchenko et al., 2024; Sukiennik et al., 2025; Wang et al., 2023b), European Value Survey (EVS) (Tao et al., 2024) and GlobalOpinionQA (Durmus et al., 2023). More recent benchmarks construct open-ended QA data around these frameworks to better mirror real-world LLM use cases (Karinshak et al., 2024; Banerjee et al., 2024). Besides, other cultural dimensions have also been investigated: NORMSAGE (Fung et al., 2022) and NormAd (Rao et al., 2024) for social norms, and EtiCor (Dwivedi et al., 2023) for social etiquette. CulturalBench (Chiu et al., 2024) is a multiple-choice benchmark across comprehensive domains, curated and verified by humans. In most evaluations, *even advanced LLMs exhibit biases towards Western-centric values (Wang et al., 2023a), underscoring the urgency of promoting cultural alignment.*

**Approaches to Cultural Alignment**   Early efforts focus on In-Context Learning (ICL) (Dong et al., 2022), including prompting LLMs to consider from culture-specific perspectives (Durmus et al., 2023), role-playing with demographic attributes (Kwok et al., 2024; Kharchenko et al., 2024) or enriching prompts with cultural value descriptions (Choenni & Shutova, 2024). Native language prompts sometimes also improve alignment (Durmus et al., 2023; Cao et al., 2023). However, these methods depend on strong ICL capabilities and pre-existing cultural knowledge, making them less effective for smaller or weaker LLMs (Saunders et al., 2022). A more scalable solution involves fine-tuning LLMs with culturally grounded datasets (Li et al., 2024a;b) and cultural learning-inspired strategies (Liu et al., 2025; Yuan et al., 2024), *highlighting the need for high-quality cultural datasets.*

**Datasets for Cultural Alignment**   Current studies have explored four main categories of cultural datasets. The first is *large-scale local-language corpora*, which are used to train regional LLMs from English-centric models (Pires et al., 2023; Nguyen et al., 2023b; Pipatanakul et al., 2023; Abbasi et al., 2023). Nonetheless, this approach requires prohibitive computational cost and language data alone provides limited cultural specificity. The second is *culture-related data filtered from large or website corpora*. CultureInstruct (Pham et al., 2025) and CRAFT (Wang et al., 2024a) automatically identify culturally rich samples from large-scale collections. CultureBank (Shi et al., 2024) and CultureAtlas (Fung et al., 2024) collect cultural expressions from Tiktok/Reddit and Wikipedia respectively. To ensure higher quality, *manually curated datasets* are also incorporated, including NORMSAGE (Fung et al., 2022) and NORMBANK (Ziems et al., 2023) for social norms (Feng et al., 2025), CLIcK (Kim et al., 2024) and BLEnD (Myung et al., 2024) for cultural commensense (Nguyen et al., 2023a), and WVS survey for values (Haerpfer et al., 2020). Finally, *cultural datasets augmented or synthesized by LLMs with abundant cultural knowledge* is an emerging category (Yuan et al., 2024). For example, CultureLLM (Li et al., 2024a) and CulturePark (Li et al., 2024b) augment WVS results with model-generated opinions. CultureSPA (Xu et al., 2024a) synthesizes questions with shifted answers under culture-unaware and -aware settings. Though these datasets are beneficial for cultural alignment, *they could still encounter the challenges of representativeness and distinctiveness to achieve effective and efficient alignment, which are mainly optimized in this paper.*

## 3 METHODOLOGY

### 3.1 FORMALIZATION AND OVERVIEW

Define $p_{\boldsymbol{\theta}}(\mathbf{y}|\mathbf{x})$ as an LLM parameterized by $\boldsymbol{\theta}$, which generates a response $\mathbf{y}$ to a given question $\mathbf{x}$; $p_{\mathbf{c}_1}(\mathbf{x}, \mathbf{y}), \ldots, p_{\mathbf{c}_{K+1}}(\mathbf{x}, \mathbf{y})$ as the true distributions of $K+1$ different cultures. Our goal is to find a set of cultural question-response pairs $q_{\mathbf{c}}^* = \{(\boldsymbol{x}^*, \boldsymbol{y}^*)\}$ with satisfactory *representativeness* and *distinctiveness* for the target culture $\mathbf{c}$, to achieve effective and sample-efficient cultural alignment of a given backbone model $p_{\boldsymbol{\theta}}$. For this purpose, we must solve the objective below:

$$q_{\mathbf{c}}^* = \underset{(\boldsymbol{x}, \boldsymbol{y})}{\operatorname{argtopN}} \{p_{\mathbf{c}}(\boldsymbol{x}, \boldsymbol{y}) - \gamma * \frac{1}{K} \sum_{\boldsymbol{c}_k \neq \boldsymbol{c}} p_{\boldsymbol{c}_k}(\boldsymbol{x}, \boldsymbol{y})\}, \tag{1}$$

where $\gamma$ is a hyperparameter. This objective helps identify the data samples $(\boldsymbol{x}, \boldsymbol{y})$ that i) have a large probability mass $p_{\mathbf{c}}(\boldsymbol{x}, \boldsymbol{y})$, *i.e.*, salient for the target culture $\boldsymbol{c}$, and ii) have a smaller probability of being shared by the non-target cultures, ensuring both representativeness and distinctiveness.

Nevertheless, each *true* $p_{\mathbf{c}_k}(\mathbf{x}, \mathbf{y}), k = 1, \ldots, K+1$, is unavailable. Therefore, inspired by culture theories, we propose CAReDiO, an LLM-empowered in-context framework to approximate and

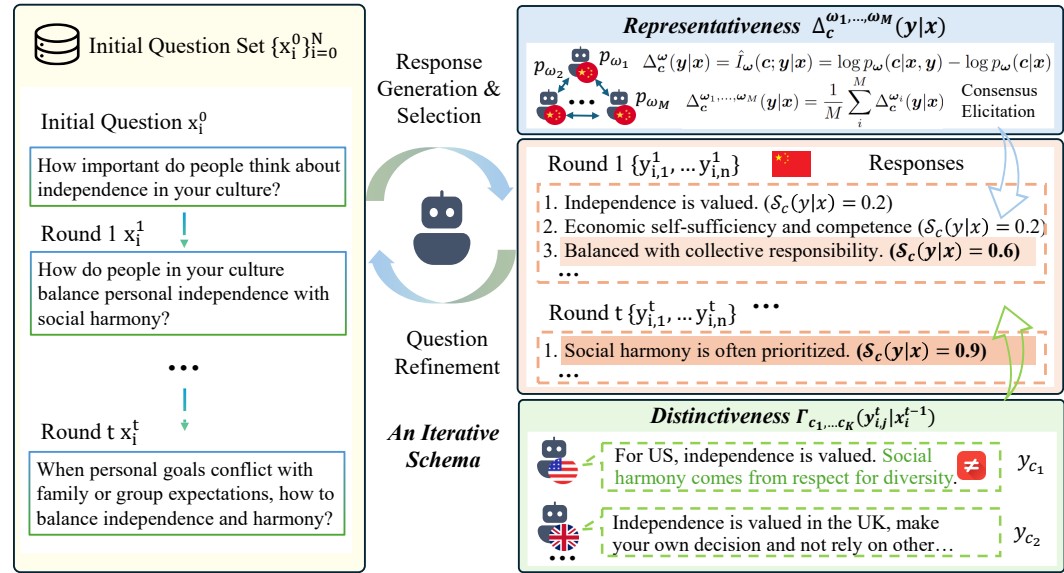

Figure 2: The CAReDiO framework, including modules for the representativeness and distinctiveness objectives, as well as an iterative schema to alternately optimize questions and responses.

optimize Eq.(1). As shown in Fig. 2, CAReDiO consists of three core components: i) an *information gain objective* to create samples that can contribute more information and further reduce $p_\theta$'s cultural bias and increase saliency $p_c(\boldsymbol{x}, \boldsymbol{y})$, *improving representativeness*; ii) a *culture divergence objective* to enhance data distinguishability from non-target cultures and decrease $p_{c_k}(\boldsymbol{x}, \boldsymbol{y}), c_k \neq c$, *improving distinctiveness*; and iii) an *iterative schema* that alternately refines the cultural question $\boldsymbol{x}$ and corresponding response $\boldsymbol{y}$ to optimize the two objectives until convergence, leading to coherent, clean, and informative alignment data. We elaborate on each module in the following subsection.

## 3.2 THE CAREDIO FRAMEWORK

We introduce our CAReDiO framework to optimize questions and answers for the target culture $\mathbf{c}$, handling the *C1: Representativeness* and *C2: Distinctiveness* challenges discussed in Sec. 1.

**Representativeness Optimization via Information Gain**    The major challenge of solving Eq.(1) lies in that $p_c(\mathbf{y}|\mathbf{x})^2$ is unavailable, and thus we can neither sample $\boldsymbol{y}$ from the true distribution nor obtain the density of $\boldsymbol{y}$. Fortunately, *Cultural Consensus Theory* (CulCT) (Weller, 2007) from cognitive anthropology and cultural psychology indicates that for a culture $\mathbf{c}$, its salient elements correspond to shared cognition of people with cultural competence. Building upon this theory, we approximate representativeness optimization as a problem of *consensus elicitation*.

We incorporate multiple LLMs, $p_{\omega_1}, \dots, p_{\omega_M}$, to mimic a group of individuals with cultural competence of the target culture $\boldsymbol{c}$, which mimics the scenario of humans participating in the rating and consensus forming process. This simulation is completed using an ICL alignment method, such as role-playing (Kwok et al., 2024; Li et al., 2024b). To ensure diversity among individuals, each $\omega_i$ could be either a heterogeneous LLM or the same one with different demographic role-plays.

Then, for each rater $p_\omega$, we quantify how well a response $\boldsymbol{y}$ reflects its cognition of culture $\boldsymbol{c}$ by the Mutual Information (MI) between $\boldsymbol{y}$ and $\boldsymbol{c}$: $I_\omega(\mathbf{c}; \mathbf{y}|\mathbf{x} = \boldsymbol{x}) = \mathbb{E}_{p_\omega(\mathbf{y}|\boldsymbol{x})}\mathbb{E}_{p_\omega(\mathbf{c}|\mathbf{y},\boldsymbol{x})}[\log p_\omega(\mathbf{c}|\boldsymbol{x}, \mathbf{y}) - \log p_\omega(\mathbf{c}|\boldsymbol{x})]$. Concretely, for a given sample $(\boldsymbol{x}, \boldsymbol{y})$ and the target culture $c$, we compute point-wise MI as the rating score of $p_\omega$, *i.e.*, $\Delta_c^\omega(\boldsymbol{y}|\boldsymbol{x})$:

$$\Delta_c^\omega(\boldsymbol{y}|\boldsymbol{x}) = \hat{I}_\omega(\boldsymbol{c}; \boldsymbol{y}|\boldsymbol{x}) = \log p_\omega(\boldsymbol{c}|\boldsymbol{x}, \boldsymbol{y}) - \log p_\omega(\boldsymbol{c}|\boldsymbol{x}). \quad (2)$$

For open-sourced LLMs, we can directly obtain the probabilities required in Eq.(2), while for black-box LLMs, we approximate each term by prompting it to return discrete scores as the probability.

---

[2]For brevity, we describe the optimization of $\mathbf{y}$, assuming $\mathbf{x}$ is obtained in all equations. In practice, we also conduct a dual refinement process for $\mathbf{x}$ to optimize $p_c(\mathbf{x})$.

Finally, we use the following score to measure the consensus and find $\boldsymbol{y}$ with the highest score:

$$\Delta_{\boldsymbol{c}}^{\boldsymbol{\omega}_1,\ldots,\boldsymbol{\omega}_M}(\boldsymbol{y}|\boldsymbol{x}) = \frac{1}{M}\sum_i^M \Delta_{\boldsymbol{c}}^{\boldsymbol{\omega}_i}(\boldsymbol{y}|\boldsymbol{x}), \tag{3}$$

In this way, for a given question, we can identify representative responses that better reflect the *shared cognition* of the target culture $\boldsymbol{c}$. Actually, Eq.(2) can be regarded as Information-Directed Sampling (IDS) (Hao et al., 2022), which identifies responses $\boldsymbol{y}$ that reinforce $p_{\boldsymbol{\omega}}$'s understanding of culture $\boldsymbol{c}$. $p_{\boldsymbol{\omega}}$ could be either $p_{\boldsymbol{\theta}}$ (the target LLM to be aligned), where Eq.(2) performs a kind of Eliciting Latent Knowledge (ELK) (Mallen et al., 2023), or a stronger one (*e.g.*, GPT-5), where it degenerates into knowledge distillation (Xu et al., 2024b). The concrete implementation is provided in Appendix B.2, and the plausibility of this consensus elicitation are empirically verified in Appendix C.1.

With this representativeness optimization method, we then give the following conclusion:

**Proposition 1** *For two samples $\boldsymbol{y}_1, \boldsymbol{y}_2$ toward the same question $\boldsymbol{x}$, if their scores from Eq.(2) satisfy $\Delta_{\boldsymbol{c}}^{\boldsymbol{\omega}}(\boldsymbol{y}_1|\boldsymbol{x}) > \Delta_{\boldsymbol{c}}^{\boldsymbol{\omega}}(\boldsymbol{y}_2|\boldsymbol{x})$, under mild conditions, using $\boldsymbol{y}_1$ for fine-tuning leads to a larger gradient in learning the true distribution $p_{\boldsymbol{c}}(\boldsymbol{x}, \boldsymbol{y})$ than using $\boldsymbol{y}_2$: $||\nabla_{\boldsymbol{\theta}} l_{\boldsymbol{\theta}}(\boldsymbol{y}_1, \boldsymbol{x})|| > ||\nabla_{\boldsymbol{\theta}} l_{\boldsymbol{\theta}}(\boldsymbol{y}_2, \boldsymbol{x})||$.*

*Proof.* See Appendix. A.

**Distinctiveness Optimization by Culture Divergence** As demonstrated in Fig. 1, highly representative samples for $\boldsymbol{c}$ won't naturally be distinguishable from other non-target ones, as closely connected cultures, *e.g.*, China, Japan and Korea, often share similar values, norms and behaviors. To achieve precise cultural modelling, we must better capture culture boundaries and construct distinguishable $\boldsymbol{y}$. Here, we resort to Cognitive Conflict Theory (CogCT) (Cosier & Rose, 1977), which indicates cognitive conflicts among cultures can provoke self-reflection on their own culture.

Technically, we set the target culture $\boldsymbol{c} = \boldsymbol{c}_{K+1}$, and $\boldsymbol{c}_1, \ldots, \boldsymbol{c}_K$ as the non-target ones for convenience, and use the generalized JS divergence (Englesson & Azizpour, 2021) between the data distribution $q_{\boldsymbol{c}}$ and $p_{\boldsymbol{c}_1}, \ldots, p_{\boldsymbol{c}_K}$, *i.e.*, $\mathrm{GJS}_{\alpha,\boldsymbol{w}}[q(\mathbf{y}|\boldsymbol{x}), p_{\boldsymbol{c}_1}(\mathbf{y}|\boldsymbol{x}), \ldots, p_{\boldsymbol{c}_K}(\mathbf{y}|\boldsymbol{x})]$, to measure distinctiveness. $\alpha, \boldsymbol{w} = (w_1, \ldots, w_K) > 0$ are weights (hyper-parameters) for each distribution with $\alpha + \sum_{i=1}^K w_i = 1$. Nevertheless, the difficulty still lies in that each true culture distribution $p_{\boldsymbol{c}_k}$ is unattainable. Therefore, we use the following formulation $\Gamma_{\boldsymbol{c}_1,\ldots,\boldsymbol{c}_K}(\boldsymbol{y}|\boldsymbol{x})$ to approximate distinctiveness instead:

$$\Gamma_{\boldsymbol{c}_1,\ldots,\boldsymbol{c}_K}(\boldsymbol{y}|\boldsymbol{x}) = \phi(\boldsymbol{y},\boldsymbol{x})\left[\log\frac{\phi(\boldsymbol{y},\boldsymbol{x})}{1-\phi(\boldsymbol{y},\boldsymbol{x})} + \log\frac{1-\alpha}{2\alpha}\right] + \log(1-\phi(\boldsymbol{y},\boldsymbol{x})), \tag{4}$$

where $\phi(\boldsymbol{y},\boldsymbol{x})$[3] is a classifier to give the probability that the response $\boldsymbol{y}$ (together with $\boldsymbol{x}$) does NOT come from any of the $K$ non-target cultures. The classifier can be either fine-tuned or approximated ones like *llm-as-judge*. Since high-quality annotated cultural data is rare, we implement it using an LLM $p_{\omega}$ and the OpenAI text-embedding-3-small model. Given a specific $(\boldsymbol{x}, \boldsymbol{y})$, we ask $p_{\omega}$ to generate responses $\boldsymbol{y}_{\boldsymbol{c}}$ for the target culture $\boldsymbol{c}$ and $(\boldsymbol{y}_{\boldsymbol{c}_1}, \ldots, \boldsymbol{y}_{\boldsymbol{c}_K})$ for other non-target cultures. Then, we encode them with the embedding model and approximate $\phi(\boldsymbol{y}, \boldsymbol{x})$ as $\phi(\boldsymbol{y},\boldsymbol{x}) = \frac{\exp(\mathrm{sim}(\boldsymbol{e}_{\boldsymbol{y}},\boldsymbol{e}_{\boldsymbol{c}}))}{\mathrm{sim}(\boldsymbol{e}_y,\boldsymbol{e}_c)+\sum_{k=1}^K \mathrm{sim}((\boldsymbol{e}_y,\boldsymbol{e}_{c_k}))}$, where $\boldsymbol{e}_{\boldsymbol{y}}, \boldsymbol{e}_{\boldsymbol{c}}, \boldsymbol{e}_{\boldsymbol{c}_k}$ are text embeddings and $\mathrm{sim}(\cdot, \cdot)$ measures similarity. We discuss more choices of classifiers in Appendix C.2

For the above approximation of the distinctiveness objective, we provide a conclusion:

**Proposition 2** *For any give $\boldsymbol{x}$ and $\boldsymbol{y}$, if the classifier error $|\phi(\boldsymbol{y},\boldsymbol{x}) - p^*(\boldsymbol{y} \notin p_{\boldsymbol{c}_1}, \ldots, p_{\boldsymbol{c}_K}|\boldsymbol{y},\boldsymbol{x})| < \epsilon$, and the classifier is not over-confident, i.e., $\phi(\boldsymbol{y},\boldsymbol{x}) < \eta$, then maximizing Eq.(4) is an approximated point-wise maximization of the lower bound of $\mathrm{GJS}_{\alpha,\mathbf{w}}[q(\mathbf{y}|\boldsymbol{x}), p_{\boldsymbol{c}_1}(\mathbf{y}|\boldsymbol{x}), \ldots, p_{\boldsymbol{c}_K}(\mathbf{y}|\boldsymbol{x})]$, and the approximation error $\mathcal{E}$ is bounded by $\mathcal{E} < \epsilon|\log\frac{\eta(1-\alpha)}{2(1-\eta)\alpha}|$.*

*Proof.* See Appendix. A.

Prop. 2 implies we can directly use a reliable classifier to score each created $\boldsymbol{y}$ according to Eq.(4) and select the top-$N$ ones to form the dataset $q_{\boldsymbol{c}}$. This process actually maximizes a lower bound of the true

---

[3]We abbreviate $p_{\phi}(\boldsymbol{y} \notin p_{\boldsymbol{c}_1}, \ldots, p_{\boldsymbol{c}_K}|\boldsymbol{y},\boldsymbol{x})$ or $p_{\phi}(\boldsymbol{x} \notin p_{\boldsymbol{c}_1}, \ldots, p_{\boldsymbol{c}_K}|\boldsymbol{y},\boldsymbol{x})$ as $\phi(\boldsymbol{y},\boldsymbol{x})$.

distinctiveness and elicits cultural differences, even if we cannot access the real culture distributions. Empirical verification of the classifier error bound in practice is provided in Appendix C.2.

Eventually, we combine the two objectives and use the following score for data optimization:

$$\mathcal{S}_{\boldsymbol{c}}(\boldsymbol{y}|\boldsymbol{x}) = \lambda_1 \cdot \Delta_{\boldsymbol{c}}^{\boldsymbol{\omega}_1,\dots,\boldsymbol{\omega}_M}(\boldsymbol{y}|\boldsymbol{x}) + \lambda_2 \cdot \Gamma_{\boldsymbol{c}_1,\dots,\boldsymbol{c}_K}(\boldsymbol{y}|\boldsymbol{x}) + \lambda_3 \cdot \mathbb{E}_{(\boldsymbol{x}',\boldsymbol{y}')\sim q_{\boldsymbol{c}}}\left[\mathcal{K}\left((\boldsymbol{x},\boldsymbol{y}),(\boldsymbol{x}',\boldsymbol{y}')\right)\right]. \quad (5)$$

$\lambda_1$, $\lambda_2$, $\lambda_3$ are hyperparameters, and $\mathcal{K}(\cdot,\cdot)$ represents sematic distance to ensure data diversity. Note that we describe only the optimization of $\boldsymbol{y}$ for brevity, and conduct a dual computation for $\mathcal{S}_{\boldsymbol{c}}(\boldsymbol{x}|\boldsymbol{y})$.

**Iterative Data Optimization**   Based on the information-theoretic objectives and an initial set of cultural questions $\{x_i^0\}_{i=0}^N$, we iteratively and alternately create and refine the questions $\boldsymbol{x}$ and responses $\boldsymbol{y}$ for a specified target culture $\boldsymbol{c}$. Concretely, utilizing an LLM $p_{\boldsymbol{\omega}}$ and classifier $\phi(\boldsymbol{y},\boldsymbol{x})$ (in all our experiments, $\boldsymbol{\theta}=\boldsymbol{\omega}$), at the $t$-th iteration, we fix the question $\boldsymbol{x}^{t-1}$ from the last iteration, and then instruct the LLM to generate the response $\boldsymbol{y}^t$. To enhance distinctiveness, we prompt $p_{\boldsymbol{\omega}}$ with both the question $\boldsymbol{x}$ and answers generated for other non-target cultures, $\boldsymbol{y}_{\boldsymbol{c}_1}, \boldsymbol{y}_{\boldsymbol{c}_2}, \dots, \boldsymbol{y}_{\boldsymbol{c}_K}$, then require it to consider the unique nuances of culture $c$ for response generation. Together with representativeness, we prompt $p_{\boldsymbol{\omega}}$ with different roles to obtain diverse responses $(\boldsymbol{y}_1, \dots, \boldsymbol{y}_n)$. Then, we compute the score for each $\boldsymbol{y}_i$ and select the one that maximizes Eq.(5) as final $\boldsymbol{y}^t$. Once we obtain the optimal $\boldsymbol{y}^t$, we further refine $\boldsymbol{x}^t$ to improve the score $\mathcal{S}_{\boldsymbol{c}}(\boldsymbol{x}|\boldsymbol{y})$. We provide $p_{\boldsymbol{\omega}}$ with all generated $(\boldsymbol{y}_1, \dots, \boldsymbol{y}_n)$ and their scores in the $t$-th iteration, and then $p_{\boldsymbol{\omega}}$ refines $\boldsymbol{x}^{t-1}$ to increase the generation prob-

---

**Algorithm 1:** The CAREDiO Framework

**Input:** Maximum number of iterations $T$, the LLM $p_{\boldsymbol{\omega}}$, the classifier $\phi(\boldsymbol{y},\boldsymbol{x})$, target culture $\boldsymbol{c}$, non-target culture $\boldsymbol{c}_1, \dots, \boldsymbol{c}_K$

**Output:** Optimized data $q_{\boldsymbol{c}}^* = \{(\boldsymbol{x}_i,\boldsymbol{y}_i)\}_{i=0}^N$

1 Initialize a set of cultural questions $\{\boldsymbol{x}_i^0\}_{i=0}^N$
2 **for** $t = 1, 2, \dots, T$ **do**
3    **for** $i = 1, 2, \dots, N$ **do**
4       Generate multiple responses $\{\boldsymbol{y}_{i,1}^t, \dots, \boldsymbol{y}_{i,n}^t\}$ from the LLM;
5       Calculate score $\mathcal{S}_{\boldsymbol{c}}(\boldsymbol{y}_{i,j}^t|\boldsymbol{x}_i^{t-1})$ for each $\boldsymbol{y}_{i,j}^t$ by Eq.(5);
6       Select the highest-scoring one as $\boldsymbol{y}_i^t$;
7       Ask $p_{\boldsymbol{\omega}}$ to refine $\boldsymbol{x}_i^{t-1}$ to $\{\hat{\boldsymbol{x}}_{i,j}^t, \dots, \hat{\boldsymbol{x}}_{i,m}^t\}$;
8       Calculate $\mathcal{S}_{\boldsymbol{c}}(\hat{\boldsymbol{x}}_{i,j}^t|\boldsymbol{y}_i^t)$;
9       Select the highest-scoring one as $\boldsymbol{x}_i^t$

---

ability of representative and distinctive responses and suppress those with lower scores. To be compatible with black-box LLMs, the optimization process is performed through LLM ICL without any training, until convergence or reaching early stopping criteria. The complete algorithm is summarized in Algorithm 1. Detailed prompts and implementations are provided in Appendix B.2.

## 3.3 CARDSET CONSTRUCTION AND ALIGNMENT

**CARDSet Creation**   To validate the effectiveness of CAReDiO , we use it to create a cultural alignment dataset CARDSet. First, we leverage the LLM $p_{\boldsymbol{\omega}}$ to initiate questions $\{\boldsymbol{x}_i^0\}_{i=1}^N$ around diverse topics from core culture aspects, including cultural values, *e.g.*, Hofstede Cultural Dimensions (Hofstede & Hofstede, 2005), norms and behavioral practices (See Appendix. B.1 for topic details). Specifically, several questions are first generated for each topic, then we employ the Self-Instruct approach (Wang et al., 2022) to extend more distinct questions for each (100 in this work). Four common question formats are adopted to align with the practical usage of LLMs: scenarios-based, value-oriented, open-ended and multiple-choice questions (Detailed formats in Appendix B.3).

With questions $\{\boldsymbol{x}_i^0\}_{i=1}^N$, we perform the iterative data optimization process as shown in Algo. 1. Concretely, we implement Eq.(3) by prompting $p_{\boldsymbol{\omega}}$ to role-play a group of individuals from the target culture $\boldsymbol{c}$. To incorporate comprehensive knowledge and enhance reliability, these individuals are set in three types: (i) *general people with various demographics* sampled from the WVS data of $\boldsymbol{c}$; (ii) *cultural experts with different backgrounds*, such as sociologists; and (iii) *cross-cultural researchers*. We use 15 general people, 5 cultural experts, and 3 cross-cultural researchers. We use different $p_{\boldsymbol{\omega}}$ to synthesize multiple versions of CARDSet for comparison experiments in Tab. 3 and Fig. 4 (b).

**Alignment Fine-tuning**   Through the above optimization process, we can obtain a great deal of cultural data $q_{\boldsymbol{c}}^* = \{(\boldsymbol{x}^*,\boldsymbol{y}^*)\}$ reinforcing culture *representativeness* and *distinctiveness*, where each sample has a score $\mathcal{S}_{\boldsymbol{c}}(\boldsymbol{x},\boldsymbol{y})$. However, training on the entire dataset incurs high computational costs

and some data might be redundant. Therefore, we rank all these samples based on their score $\mathcal{S}_c$ and sequentially select those according to the pre-defined computational budget. To ensure data diversity, we compute the similarity between the subsequent candidate and the selected ones, omitting those with a similarity score higher than $\tau = 0.85$. With responses generated for other cultures in the distinctiveness optimization step as dispreferred ones, we can fine-tune cultural LLMs via SFT or DPO. To ensure a fair comparison, we follow most baselines to use SFT in all experiments.

## 4 EXPERIMENTS

### 4.1 EXPERIMENTAL SETTINGS

**Evaluation Datasets and Metrics**   We measure cultural alignment with four benchmarks targeting distinct aspects. Statistical information is listed in Table 1.

(1) **CulturalBench** (Chiu et al., 2024): A manually curated benchmark with 1,227 four-choice questions for cultural knowledge, spanning 45 regions and 17 topics. There are two variants *Easy* and *Hard*, where *Hard* transforms each item into four binary questions and the LLM should judge all binary questions correctly. *Accuracy* is calculated on the ground truth for each individual culture.

Table 1: Statistics of evaluation benchmarks.

| Dataset | Types | #Samples | Metrics |
|---|---|---|---|
| CulturalBench | Multiple-Choice | 1,227 | Accuracy |
| Prism | Open-Ended QA | 468 | Quality Rating |
| GlobalOpinionQA | Questionnaire | 2,556 | Accuracy |
| WVS | Questionnaire | 260 | Consistency |

(2) **Prism** (Kirk et al., 2025): It contains conversations between 1,500 participants across 75 countries and 21 LLMs. We maintain value-related questions raised by people from difference countries with two criteria: i) the question involves cultural topics such as *relationship management* and *abortion*; and ii) responses could vary meaningfully across cultures. For assessment, we introduce both llm-as-judge (Gemini-2.5-Pro) and native annotators to rate the response *quality* on a 1-5 scale.

(3) **GlobalOpinionQA** (Durmus et al., 2023): It compiles items from Global Attitudes surveys (GAS) and World Value Survey (WVS). To avoid overlap with the next dataset, we retain only the GAS subset. Each item consists of a question, multiple choices and the choice distributions across various countries. We report *accuracy* as whether the model's prediction matches the top-1 human choice.

(4) **World Value Suryve (WVS)** (Haerpfer et al., 2020): A questionnaire surveying people's values across 13 topics. It collects real responses from people across countries. We compute the *consistency* between the predictions of an LLM $p_\theta$ and the real answers from the culture $c$ following Xu et al. (2024a). More details about these benchmarks and human evaluation are provided in Sec. D.1.

**Baselines**   To comprehensively assess our approach, we conduct cultural alignment on LLMs from different families and scales, including proprietary GPT-4.1 and GPT-5, open-source models including LLaMA-3.1-8B-Instruct (Dubey et al., 2024), Qwen2.5-7B-Instruct (Team, 2024) and Gemma-3-27B-IT (Team et al., 2025). Across all backbones, we adopt the **Role-Play** baseline that applies system prompts to simulate individuals from specific cultural backgrounds. Besides, we finetune cultural LLMs with our CARDSet and multiple datasets derived from different sources.

As summarized in Tab. 2, six cultural datasets are compared. **CultureLLM** (Li et al., 2024a) and **CulturePark** (Li et al., 2024b) are augmented by GPT-4-Turbo based on the real WVS data. **CultureSPA** (Xu et al., 2024a) is WVS-style opinion data synthesized by LLMs. **CultureBank** (Shi et al., 2024) and **CultureInstruct** (Pham et al., 2025) are culture-relevant text filtered from Tiktok/Reddit and the DOLMA corpus (Soldaini et al., 2024) respectively. **CultureData** is constructed by ourselves through merging all public manually created cultural datasets, such as NORMBANK (Ziems et al., 2023) and CultureAtlas (Fung et al., 2024). For fair comparisons, we employ 1,000 samples for each culture except for CultureInstruct which is a mixed cultural dataset lacking explicit cultural labels.

More details about baselines and implementations can be found in Appendix D.4, D.5.

Table 2: Statistics of cultural datasets. Cult. Points: Culture points representing distinctive cultural aspects extracted from the dataset by GPT-4.1; Sim and SB are cosine similarity and Self-BLEU within each dataset; Cult. Sim is cosine similarity between subsets of different cultures.

| Datasets | Source | #sample | Avg.L ↑ | #Cult. Points ↑ | Sim ↓ | SB ↓ | Cult. Sim ↓ |
|---|---|---|---|---|---|---|---|
| CultureLLM | WVS augmentation | 1,000 each | 48.6 | 245.2 | 0.246 | 0.616 | 0.246 |
| CulturePark | WVS augmentation | 1,000 each | 68.6 | 494.6 | 0.235 | 0.406 | 0.223 |
| CultureSPA | LLM-synthetic | 1,000 each | 46.7 | 517.6 | 0.261 | 0.410 | 0.264 |
| CultureBank | web platform | 18,396 total | 87.4 | 442.0 | 0.229 | **0.167** | 0.187 |
| CultureInstruct | pretrain corpus | 46,878 total | 191.1 | - | - | - | - |
| CultureData | manual annotation | 1,000 each | 16.8 | 1521.0 | **0.199** | 0.330 | **0.127** |
| CARDSet | LLM-synthetic | 1,000 each | **200.4** | **2027.0** | 0.251 | 0.324 | 0.202 |

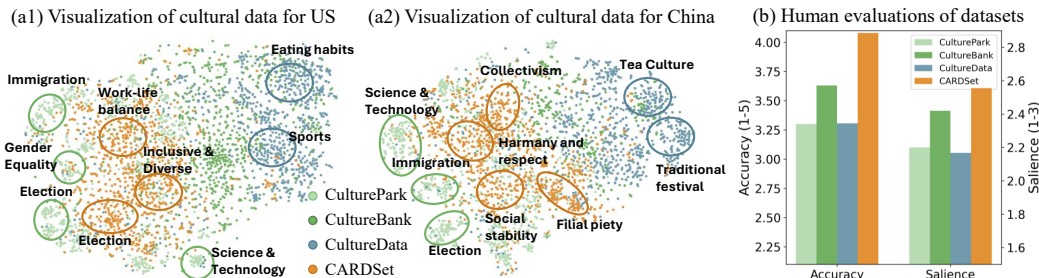

(a1) Visualization of cultural data for US    (a2) Visualization of cultural data for China    (b) Human evaluations of datasets

Figure 3: TSNE visualization and human evaluation of cultural datasets.

## 4.2 CULTURAL DATASET ANALYSIS

Before delving into the performance of cultural alignment, we first compare the quality of CARD-Set generated by our CAReDiO framework with existing cultural datasets introduced in Sec. 4.1.

**Quantitative Analysis** As shown in Tab. 2, textual samples in CARDSet are generally longer and contain richer cultural information (with more cultural points in the raw text extracted by GPT-4.1). This suggests that CARDSet is more informative and better captures core cultural factors. Moreover, CARDSet exhibits lower intra- and inter-cultural similarity, indicating that the dataset encodes more diverse and unique cultural knowledge compared to prior baselines.

**Visualization Analysis** We further apply t-SNE to visualize the embedding space of cultural texts in Fig. 3. For CulturePark, it covers important but generic value topics for different countries, such as '*Gender Equality*'. It might be hard to distinguish subtle cultural variations. Regarding CultureData, it captures unique aspects of different cultures, like the 'Tea Culture' for China, while often at a superficial level and loosely connected to deeper cultural norms and values. By contrast, CARDSet perfectly overcomes both challenges, highlighting representative and distinctive aspects that tie directly to cultural cores. For example, CARDSet captures a defining value *Inclusive & Diverse* for the US and the *Filial Piety* for China. For the whole distribution, CARDSet also locates at a joint region of other datasets, suggesting it provides both broader coverage and core factors.

**Human Evaluation on Data Quality** To complement automatic evaluation, we recruit native annotators from the corresponding cultures to assess the data quality along two dimensions: 1) Accuracy (1-5), which means the consensus level of this data to the culture; 2) Salience (1-3), representativeness and importance of the cultural aspect. As shown in Fig. 3 (b), CARDSet exhibits significant superiority to other datasets across both dimensions, highlight the effectiveness of our method. More details about annotator recruitment and guidance are provided in Appendix D.3.

## 4.3 CULTURAL ALIGNMENT PERFORMANCE

**Settings**. We conduct alignment across 15 cultures and 4 different LLM backbones to compare the alignment performance of CAReDiO against various baselines. Tab. 3 reports the scores averaged across 15 cultures on all four benchmarks, using Qwen2.5-7B-Instruct, Gemma-3-27B-IT and GPT-4.1 as backbone being aligned respectively. Detailed per-culture results and experiments with

Table 3: Evaluation results of cultural alignment across four benchmarks. 'CB' denotes Cultural-Bench; 'Average' represents the mean score over all benchmarks. * marks the best results across all backbones. For each LLM, the best and second-best results are highlighted in bold and underlined.

| Family | Method | CB-Easy | CB-Hard | Prism | GlobalOpinionQA | WVS | Average |
|---|---|---|---|---|---|---|---|
| Proprietaty LLMs | GPT-5 | 88.79 | 59.54 | 2.187 | 46.27 | 62.38 | 60.14 |
| | GPT-5 + Role-Play | 89.55 | 59.99 | 4.519 | 60.08* | 70.44* | 74.09* |
| Qwen2.5-7B-Instruct | Raw Model | 72.01 | 38.90 | 2.103 | 53.28 | 59.68 | 53.18 |
| | Role-Play | 72.38 | 36.73 | _3.364_ | 55.83 | 64.96 | 59.44 |
| | CultureLLM | 71.79 | 34.79 | 3.121 | _57.11_ | **65.49** | 58.32 |
| | CulturePark | 71.99 | 34.41 | 3.107 | 56.47 | 57.83 | 56.57 |
| | CultureSPA | 70.92 | 36.25 | 3.108 | 52.83 | 62.00 | 56.83 |
| | CultureBank | 72.28 | 27.34 | 3.193 | 56.43 | 62.47 | 56.48 |
| | CultureInstruction | 72.77 | 27.75 | 3.346 | **57.78** | 63.25 | 57.69 |
| | CultureData | _72.83_ | _40.11_ | 3.354 | 57.44 | 64.69 | _60.43_ |
| | CAReDiO | **73.48** | **40.20** | **3.871** | 56.23 | _65.26_ | **62.51** |
| Gemma-3-27B-IT | Raw Model | _82.11_ | 46.59 | 2.174 | 51.83 | 64.77 | 57.76 |
| | Role-Play | 81.33 | _48.28_ | _4.571_ | 54.84 | 67.22 | 68.62 |
| | CultureLLM | 80.46 | 46.31 | 4.441 | 58.15 | 66.99 | 68.14 |
| | CulturePark | 81.85 | 46.34 | 4.474 | **59.74** | 65.95 | _68.67_ |
| | CultureSPA | 81.40 | 48.00 | 4.431 | 56.59 | 67.76 | 68.48 |
| | CultureBank | 81.82 | 41.88 | 4.323 | 55.89 | 67.11 | 66.63 |
| | CultureInstruction | 76.18 | 18.19 | 3.525 | _59.22_ | 61.42 | 57.10 |
| | CultureData | 81.83 | 44.28 | 4.032 | 58.33 | **68.02** | 66.62 |
| | CAReDiO | **82.56** | **48.88** | **4.627*** | 58.25 | _67.96_ | **70.04** |
| GPT-4.1 | Raw Model | 89.82 | 59.45 | 2.131 | 52.69 | 60.91 | 61.10 |
| | Role-Play | 89.29 | _63.47_ | _4.270_ | 53.76 | **69.85** | _72.35_ |
| | CultureBank | **90.80*** | 60.00 | 4.226 | **56.76** | 68.11 | 72.04 |
| | CAReDiO | _90.32_ | **63.54*** | **4.336** | _56.64_ | 69.66 | **73.37** |

Llama-3.1-8B-Instruct are provided in Appendix F.1. Fig. 4 (a) presents human evaluation results of response quality (win rate aggregated across US, China, Japan and Poland cultures), where the LLM being aligned is Qwen2.5-7B-Instruct, compared to the role-playing baseline, CulturePark, and role-playing with GPT-4.1. In Fig. 4 (b), we compare the alignment performance of CAReDiO instantiated by different LLMs (Qwen2.5-7B-Instruct, GPT-4.1 and GPT-5) for synthesis, while fixing the backbone as Qwen2.5-7B-Instruct. Observing these results, we have two key findings below.

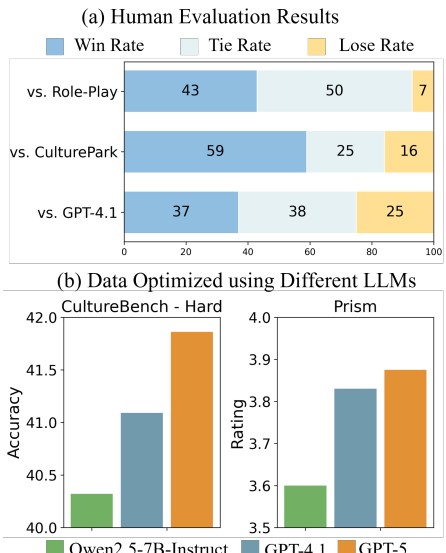

Figure 4: Results of cultural alignment.

First, *CAReDiO consistently enhances cultural alignment across LLM families and scales, including strong proprietary ones*. Compared to the raw models (Qwen2.5-7B, Gemma-3-27B, GPT-4.1) and the role-playing variants, CAReDiO achieves significant gains on all benchmarks. Interestingly, Fig. 4 (b) shows that using more capable LLM backbones for synthesis leads to further improvement, while data generated by the raw model (*i.e.*, Qwen2.5-7B-Instruct) also yields clear gains. This fully demonstrates that the improvement of cultural alignment by our framework does not solely derive from knowledge distillation but also the information-theoretic-objectives designed for representativeness and distinctiveness.

Second, *CAReDiO outperforms baselines on most benchmarks, especially CultureBench and Prism*. CultureBench is a manually curated benchmark about extensive cultural aspects, while questions in Prism come from real-world interactions. Superior performance on these data highlights the practical robustness and adaptability of our method. On GlobalOpinionQA and WVS, CAReDiO lags slightly behind CultureLLM, we guess it is due to that CultureLLM is directly augmented from actual WVS data and thus have an advantage in similar evaluations (more discussion about the possible data leakage is in Appendix D.6). We also consider synthesizing more data around this format to enhance the results in the future. Notably, for the advanced GPT-5,

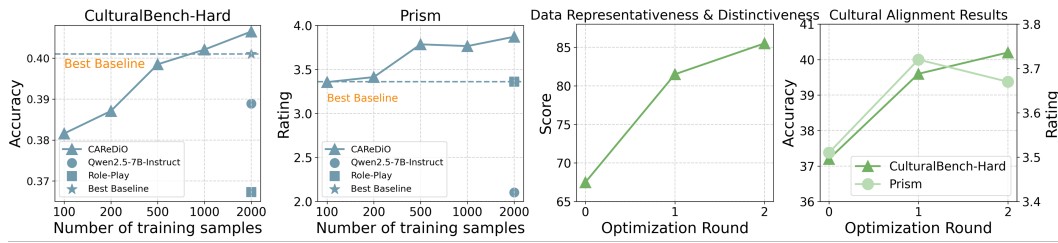

(a) Results with different numbers of training samples    (b) Results with data across optimization rounds

Figure 5: Results of two hyper-parameters: number of training samples and optimization round.

CAReDiO achieves comparable or better results on Prism. Our framework can be extended to using GPT-5 for data synthesis to improve itself once GPT-5 is available for fine-tuning in the future.

**Human Evaluation on Alignment** We also conduct human evaluation with *native annotators* from the US, China, Japan and Poland (three per culture). Showing questions from Prism and responses generated by different methods, they evaluate the quality (consensus level) from 1 (conflict the culture) to 5 (highly aligned with the culture). For each culture, we label 50-100 samples and report the average win rate in Fig. 4 (a). *Human annotators consistently rate responses by CAReDiO higher than those from baselines.* This demonstrates that our approach not only improves automatic benchmark scores but also produces outputs perceived as more culturally aligned by humans from target cultures. More details about annotation guidelines and human agreement are provided in Appendix D.3.

### 4.4 HYPERPARAMETERS ANALYSIS

**Number of Training Samples** We conduct experiments by continuously increasing the training samples from 100 to 2,000, where higher-scoring samples are selected first. We use Qwen2.5-7B-Instruct both for data synthesis and as the target model being aligned, and report the results averaged over 15 cultures. As shown in Fig. 5 (a), performance improves as more samples are introduced. Importantly, the earlier selected samples contribute more significant performance gains. Such observations indicate that our dataset is diverse enough to continuously provide learning benefits, while prioritized samples with higher representativeness and distinctiveness are more effective for cultural alignment, supporting the efficiency of our approach. On the Prism benchmark, our model reaches top performance with as few as 100 samples. This reduction in training overhead is highly valuable for fine-tuning-based methods.

**Optimization Rounds**. We also compare the cultural datasets generated in different optimized rounds, *i.e.*, the hyperparameter $T$ in Algorithm 1. Again, we use Qwen2.5-7B-Instruct for both data synthesis and for being the aligned model, and report the results averaged across over 15 cultures. As shown in Fig. 5 (b), the representativeness and distinctiveness score of data continuously increases along the iterative process, especially in the first round. Correspondingly, the alignment performance also improves. Besides, we found the performance gain is mostly achieved in the first round; thus, our framework is also efficient in data synthesis to minimize the generation cost.

More experiments about effectiveness and reliability are offered in the Appendix, including an ablation study in Appendix 15, fine-grained distinctiveness analysis among related cultures (China/Japan/Korea) in Appendix F.3, analysis about results variance across cultures and benchmark types in Appendix F.4, analysis about the number of individuals in representative optimization in Appendix F.5, robustness to the classifier in Appendix F.6, intuitive case study in Appendix F.7.

### 5 CONCLUSION

This paper addresses the challenges of representativeness and distinctiveness in cultural alignment datasets by introducing CAReDiO, an LLM-empowered data optimization framework for automatic cultural data construction. It involves an iterative process to generates and refines questions and responses for two information-theoretic objectives, thus enhancing representativeness and distinctiveness. Using the constructed dataset CARDSet covering 15 cultures, we demonstrate the superiority of CAReDiO over baseline datasets. Limitations and future directions are discussed in Appendix G.

## ETHICS STATEMENT

This paper introduces CAReDiO, a novel framework to enhance cultural alignment of LLMs. We are aware of the potential ethical implications and societal impact of this line of work, and we emphasize the importance of responsible development. For transparency and reproducibility, we provide implementation details in the Appendix and commit to releasing the necessary code and data upon acceptance. Given the cultural biases that persist in current LLMs and the associated risks, our framework is specifically designed to improve cultural alignment and is not intended for malicious use. While our experiments focus on 15 cultures, the framework is generalizable to a wide range of cultures, which we believe contributes to greater fairness and inclusivity. Furthermore, by improving the efficiency of alignment, our approach makes it more feasible to support underrepresented cultures with limited resources.

## REPRODUCIBILITY STATEMENT

Due to space limitations, many technical details such as derivations, implementations, and some experimental settings could not be included in the main body and have instead been provided in the Appendix. Specifically, the Appendix contains: (1) derivation of CAReDiO algorithm in Appendix A, (2) details for dataset creation such as the topic definition and prompts in Appendix B, (3) more detailed experimental settings in Appendix D and (4) detailed experimental results in Appendix F. We submit the core code of our method as the supplementary materials for clarity and commit to release the necessary code and data upon acceptance to support reproducibility.

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

# A SUPPLEMENTS FOR DERIVATION

Define $p_\theta(\mathbf{y}|\mathbf{x})$ as an LLM which generates a response $\mathbf{y}$ to a given question $\mathbf{x}$; $\mathbf{c}_1, \ldots, \mathbf{c}_K, \mathbf{c}_{K+1}$ as $K+1$ different cultures. Our goal is to find the best question-response pair $(\mathbf{x}^*, \mathbf{y}^*)$ which satisfies: i) $(\mathbf{x}^*, \mathbf{y}^*) = \mathrm{argmax}_{(\mathbf{x},\mathbf{y})} p_{\mathbf{c}_i}(\mathbf{x}, \mathbf{y})$ where $\mathbf{c}_i$ is the target culture; and ii) $(\mathbf{x}^*, \mathbf{y}^*) = \mathrm{argmin}_{(\mathbf{x},\mathbf{y})} \frac{1}{K} \sum_{k, k \neq i} p_{\mathbf{c}_k}(\mathbf{x}, \mathbf{y})$. Requirement i) follows our *Representativeness* rule and Requirement ii) is in line with *Distinctiveness*. We should show how it can be approximated and solved.

**Representativeness Optimization** The major challenge of $(\mathbf{x}^*, \mathbf{y}^*) = \mathrm{argmax}_{(\mathbf{x},\mathbf{y})} p_{\mathbf{c}}(\mathbf{x}, \mathbf{y})$ lies in that the true distribution of the target culture $p_{\mathbf{c}}(\mathbf{x}, \mathbf{y})$ is unavailable, and thus we cannot either sample from it or get the density. We resort to the *Cultural Consensus Theory*, which shows the "culturally correct" answer is determined by the shared beliefs of people with cultural competence. Based on this theory, we assume that each large enough LLM $p_\theta(\mathbf{y}|\mathbf{x})$ possesses sufficient competence but it is usually unelicited. Therefore, we approximate the Representativeness objective as a *consensus elicitation* problem, and find the best $\mathbf{y}$[4] that maximizes $I_\theta(\mathbf{c}; \mathbf{y}|\mathbf{x} = \boldsymbol{x})$:

$$I_\theta(\mathbf{c}; \mathbf{y}|\mathbf{x} = \boldsymbol{x}) = \mathbb{E}_{p_\theta(\mathbf{y}|\boldsymbol{x})} \int p_\theta(\mathbf{c}|\mathbf{y}, \boldsymbol{x}) \log \frac{p_\theta(\mathbf{c}|\mathbf{y}, \boldsymbol{x}) p_\theta(\mathbf{y}|\boldsymbol{x})}{p_\theta(\mathbf{c}|\boldsymbol{x}) p_\theta(\mathbf{y}|\boldsymbol{x})} d\mathbf{c}$$

$$= \mathbb{E}_{p_\theta(\mathbf{y}|\boldsymbol{x})} \mathbb{E}_{p_\theta(\mathbf{c}|\mathbf{y},\boldsymbol{x})} \left[ \log p_\theta(\mathbf{c}|\boldsymbol{x}, \mathbf{y}) - \log p_\theta(\mathbf{c}|\boldsymbol{x}) \right]. \tag{6}$$

For a sampled $\boldsymbol{y}$, we then use point-wise mutual information for Eq.(6) and use the following information score to guide the data optimization process in practice:

$$\Delta_{\boldsymbol{c}}^{\boldsymbol{\theta}}(\boldsymbol{y}) = \hat{I}_{\boldsymbol{\theta}}(\boldsymbol{c}; \boldsymbol{y}|\boldsymbol{x}) = \log p_{\boldsymbol{\theta}}(\boldsymbol{c}|\boldsymbol{x}, \mathbf{y}) - \log p_{\boldsymbol{\theta}}(\boldsymbol{c}|\boldsymbol{x}) \tag{7}$$

Eq.(7) represents a form of Information-Directed Sampling (IDS) (Hao et al., 2022), which helps find the response $\boldsymbol{y}$ that reinforces the LLM $p_{\boldsymbol{\theta}}$' understanding of culture $\boldsymbol{c}$. When $p_{\boldsymbol{\theta}}$ is the target LLM (to be aligned) itself, the optimization process can be regarded as a kind of Eliciting Latent Knowledge (ELK) (Mallen et al., 2023); when $p_{\boldsymbol{\theta}}$ is a larger LLM (potentially with better culture competence), *e.g.*, GPT-5, we conduct typical knowledge distillation. In practice, we use multiple LLMs to perform a better consensus elicitation:

$$\Delta_{\boldsymbol{c}}^{\boldsymbol{\theta}_1, \ldots, \boldsymbol{\theta}_M}(\boldsymbol{y}) = \sum_i^M \Delta^{\boldsymbol{\theta}_i}(\boldsymbol{y}), \tag{8}$$

where each $\boldsymbol{\theta}_i$ could be either a heterogeneous LLM or the same one with different individual settings. With this approximated representativeness objective, we then give the following conclusion:

*Theorem 1. For two samples $\boldsymbol{y}_1, \boldsymbol{y}_2$ toward the same question $\boldsymbol{x}$, assume their probabilities under the true cultural distribution do not differ significantly, i.e., $|p_{\boldsymbol{c}}(\boldsymbol{y}_1|\boldsymbol{x}) - p_{\boldsymbol{c}}(\boldsymbol{y}_2|\boldsymbol{x})| < \epsilon$, which holds for culturally plausible candidate answers, if their scores from Eq.(7) satisfy $\Delta_{\boldsymbol{c}}^{\boldsymbol{\theta}}(\boldsymbol{y}_1) > \Delta_{\boldsymbol{c}}^{\boldsymbol{\theta}}(\boldsymbol{y}_2)$, then using $y_1$ for fine-tuning leads to a larger gradient than using $y_2$: $||\nabla_{\boldsymbol{\theta}} l_{\boldsymbol{\theta}}(\boldsymbol{y}_1, \boldsymbol{x})|| > ||\nabla_{\boldsymbol{\theta}} l_{\boldsymbol{\theta}}(\boldsymbol{y}_2, \boldsymbol{x})||$.*

*Proof.* Assume we have true samples from the target culture $\boldsymbol{c}$, which forms the empirical distribution $p_{\boldsymbol{c}}(\mathbf{x}, \mathbf{y})$, and we use these samples to train the LLM $p_{\boldsymbol{\theta}}$ with the following loss:

$$\mathcal{L}(\boldsymbol{\theta}) = -\mathbb{E}_{(\boldsymbol{x}, \boldsymbol{y}) \sim p_{\boldsymbol{c}}} \left[ \log p_{\boldsymbol{\theta}}(\boldsymbol{y}|\boldsymbol{x}) \right]. \tag{9}$$

Then, we have the gradient for the parameters $\theta$:

$$\nabla_{\boldsymbol{\theta}} \mathcal{L}(\boldsymbol{\theta}) = -\mathbb{E}_{(\boldsymbol{x}, \boldsymbol{y}) \sim p_{\boldsymbol{c}}} \left[ \nabla_{\boldsymbol{\theta}} l_{\boldsymbol{\theta}}(\boldsymbol{y}, \boldsymbol{x}) \right], \; l_{\boldsymbol{\theta}}(\boldsymbol{y}, \boldsymbol{x}) = \log p_{\boldsymbol{\theta}}(\boldsymbol{y}|\boldsymbol{x}). \tag{10}$$

We then consider the gradient magnitude $||\nabla_{\boldsymbol{\theta}} l_{\boldsymbol{\theta}}(\boldsymbol{y}, \boldsymbol{x})||$ and demonstrate $||\nabla_{\boldsymbol{\theta}} l_{\boldsymbol{\theta}}(\boldsymbol{y}, \boldsymbol{x})|| \propto [\log p_{\boldsymbol{c}}(\boldsymbol{y}|\boldsymbol{x}) - \log p_{\boldsymbol{\theta}}(\boldsymbol{y}|\boldsymbol{x})]$. Suppose the model conducts softmax to compute the probability of output in the last layer, with the parameters $\boldsymbol{w}$. For brevity, we consider $\boldsymbol{y}$ as one single token (which is typical in multiple-choice questions), and thus we have:

$$\boldsymbol{z}_{\boldsymbol{y}}(\boldsymbol{x}) = \boldsymbol{w}_{\boldsymbol{y}}^T \cdot h(\boldsymbol{x}),$$

$$p_{\boldsymbol{\theta}}(\boldsymbol{y}|\boldsymbol{x}) = \frac{\exp(\boldsymbol{z}_{\boldsymbol{y}}(\boldsymbol{x}))}{\sum_i \exp(z_i(x))},$$

$$\log p_{\boldsymbol{\theta}}(\boldsymbol{y}|\boldsymbol{x}) = \boldsymbol{z}_{\boldsymbol{y}}(\boldsymbol{x}) - \log \sum_i \exp(\boldsymbol{z}_i(\boldsymbol{x})) \tag{11}$$

---

[4]We do an iterative optimization of $\mathbf{y}$ and $\mathbf{x}$ separately. For brevity, we fix $\mathbf{x} = \boldsymbol{x}$ and optimize $\mathbf{y}$.

where $h(\boldsymbol{x})$ is the input of the last layer. Computing the gradient for $\boldsymbol{w}_i$, we have:

$$\frac{\partial \log p_{\boldsymbol{\theta}}(\boldsymbol{y}|\boldsymbol{x})}{\partial w_i} = (\mathbb{I}[i = y] - p_{\boldsymbol{\theta}}(i|\boldsymbol{x})) \cdot h(\boldsymbol{x}). \tag{12}$$

Thus, we have $||\nabla_{\boldsymbol{\theta}} l_{\boldsymbol{\theta}}(\boldsymbol{y}, \boldsymbol{x})|| = ||\mathbb{I}[i = y] - p_{\boldsymbol{\theta}}(i|\boldsymbol{x})|| \cdot ||h(\boldsymbol{x})||$. Since the number of candidate tokens is usually huge for an LLM, the probability $p_{\boldsymbol{\theta}}(i|\boldsymbol{x})$ for tokens $i(\neq y)$ is usually very small. For simplicity, we consider the case $i = y$ that mainly affects the gradient magnitude. Defining $\delta(\boldsymbol{x}, \boldsymbol{y}) = \log p_{\boldsymbol{c}}(\boldsymbol{y}|\boldsymbol{x}) - \log p_{\boldsymbol{\theta}}(\boldsymbol{y}|\boldsymbol{x})$, we present $\log p_{\boldsymbol{\theta}}(\boldsymbol{y}|\boldsymbol{x}) = \log p_{\boldsymbol{c}}(\boldsymbol{y}|\boldsymbol{x}) - \delta(\boldsymbol{x}, \boldsymbol{y})$. Substituting $p_{\boldsymbol{\theta}}(\boldsymbol{y}|\boldsymbol{x})$ with this term and ignore $||h(\boldsymbol{x})||$ that has no correlation with $\boldsymbol{y}$, we obtain

$$||\nabla_{\boldsymbol{\theta}} l_{\boldsymbol{\theta}}(\boldsymbol{y}, \boldsymbol{x})|| = ||\mathbb{I}[i = y] - p_{\boldsymbol{\theta}}(i|\boldsymbol{x})|| \cdot ||h(\boldsymbol{x})|| \propto 1 - e^{\log p_{\boldsymbol{c}}(\boldsymbol{y}|\boldsymbol{x}) - \delta(\boldsymbol{x}, \boldsymbol{y})} \tag{13}$$

For plausible training samples $(\boldsymbol{x}, \boldsymbol{y})$ from the target culture $\boldsymbol{c}$, they have a large score in $\log p_{\boldsymbol{c}}(\boldsymbol{y}|\boldsymbol{x})$ and only yield a narrow probability difference range. Thus, $\delta(\boldsymbol{x}, \boldsymbol{y})$ plays a major role in determining the value of Eq.(13). This indicates that using a $(\boldsymbol{y}, \boldsymbol{x})$ with a larger score $\delta(\boldsymbol{x}, \boldsymbol{y})$ to fine-tune the LLM $p_{\boldsymbol{\theta}}$ can leads to a larger gradient magnitude, accelerating the cultural learning towards $p_{\boldsymbol{c}}(\boldsymbol{x}, \boldsymbol{y})$.

Since the true value of $\log p_{\boldsymbol{c}}(\boldsymbol{y}|\boldsymbol{x})$ is still unavailable, we use $p_{\boldsymbol{\theta}}$'s self-judgement to approximate it, that is, $\log p_{\boldsymbol{c}}(\boldsymbol{y}|\boldsymbol{x}) \approx \log p_{\boldsymbol{\theta}}(\boldsymbol{y}|\boldsymbol{x}, \boldsymbol{c})$. Then, we have

$$\begin{aligned}
\delta(\boldsymbol{x}, \boldsymbol{y}) &= \log p_{\boldsymbol{c}}(\boldsymbol{y}|\boldsymbol{x}) - \log p_{\boldsymbol{\theta}}(\boldsymbol{y}|\boldsymbol{x}) \\
&\approx \log p_{\boldsymbol{\theta}}(\boldsymbol{y}|\boldsymbol{x}, \boldsymbol{c}) - \log p_{\boldsymbol{\theta}}(\boldsymbol{y}|\boldsymbol{x}) \quad \text{(using Bayes Equation)} \\
&= \log \frac{p_{\boldsymbol{\theta}}(\boldsymbol{y}|\boldsymbol{x}, \boldsymbol{c}) \cdot p_{\boldsymbol{\theta}}(\boldsymbol{x}, \boldsymbol{c})}{p_{\boldsymbol{\theta}}(\boldsymbol{y}|\boldsymbol{x}) \cdot p_{\boldsymbol{\theta}}(\boldsymbol{x}, \boldsymbol{c})} \\
&= \log \frac{p_{\boldsymbol{\theta}}(\boldsymbol{c}|\boldsymbol{y}, \boldsymbol{x}) \cdot p_{\boldsymbol{\theta}}(\boldsymbol{y}|\boldsymbol{x}) \cdot p_{\boldsymbol{\theta}}(\boldsymbol{x})}{p_{\boldsymbol{\theta}}(\boldsymbol{y}|\boldsymbol{x}) \cdot p_{\boldsymbol{\theta}}(\boldsymbol{c}|\boldsymbol{x}) \cdot p_{\boldsymbol{\theta}}(\boldsymbol{x})} \\
&= \log p_{\boldsymbol{\theta}}(\boldsymbol{c}|\boldsymbol{y}, \boldsymbol{x}) - \log p_{\boldsymbol{\theta}}(\boldsymbol{c}|\boldsymbol{x}) \\
&= \Delta_{\boldsymbol{c}}^{\boldsymbol{\theta}}(\boldsymbol{y}|\boldsymbol{x}),
\end{aligned} \tag{14}$$

which indicates that using samples $\boldsymbol{y}$ with a larger score from Eq.(7) to finetune $p_{\boldsymbol{\theta}}$ can approximately leads to a larger gradient magnitude, accelerating the cultural learning towards $p_{\boldsymbol{c}}(\boldsymbol{x}, \boldsymbol{y})$.

**Distinctiveness Optimization** To optimize $(\mathbf{x}^*, \mathbf{y}^*) = \operatorname{argmin}_{(\mathbf{x}, \mathbf{y})} \frac{1}{K} \sum_{k, k \neq i} p_{\mathbf{c}_k}(\mathbf{x}, \mathbf{y})$, we refer to the *Cognitive Conflict Theory*, and elicit cultural differences from conflicts. For a given question $\boldsymbol{x}$, assume we have collected a set of $\boldsymbol{y}$ by Eq.(7), which forms an empirical distribution $q(\boldsymbol{y}|\boldsymbol{x})$ for the target culture, *e.g.*, Japan, we aim to find the best $q$ with minimal overlap with non-target cultures, *e.g.*, Korea, Singapore and UK, $\boldsymbol{c}_1, \ldots, \boldsymbol{c}_K$.

Concretely, we use the following objective:

$$\phi(\boldsymbol{y}, \boldsymbol{x}) \left[ \log \frac{\phi(\boldsymbol{y}, \boldsymbol{x})}{1 - \phi(\boldsymbol{y}, \boldsymbol{x})} + \log \frac{1 - \alpha}{2\alpha} \right] + \log(1 - \phi(\boldsymbol{y}, \boldsymbol{x})) = \Gamma(\boldsymbol{y}|\boldsymbol{x}), \tag{15}$$

$$q^*(\boldsymbol{y}|\boldsymbol{x}) = \operatorname*{argtop}_{\boldsymbol{y}} \Gamma(\boldsymbol{y}|\boldsymbol{x}) \tag{16}$$

where $\phi(\boldsymbol{y}, \boldsymbol{x})$ is a classifier to give the probability that the response $\boldsymbol{y}$ comes from any of the K non-target culture, and $\alpha \in [0, 1]$ is a hyperparamter (the weight of the target culture).

*Theorem 2. For any give $\boldsymbol{x}$ and $\boldsymbol{y}$, if the classifier error $|\phi(\boldsymbol{y}, \boldsymbol{x}) - p^*(\boldsymbol{y} \notin p_{\boldsymbol{c}_1}, \ldots, p_{\boldsymbol{c}_K}|\boldsymbol{y}, \boldsymbol{x})| < \epsilon$, and the classifier is not over-confident, i.e., $\phi(\boldsymbol{y}, \boldsymbol{x}) < \eta$, then maximizing Eq.(15) is an approximated point-wise maximization of the lower bound of $GJS_{\alpha, \mathbf{w}}[q(\boldsymbol{y}|\boldsymbol{x}), p_{\boldsymbol{c}_1}(\boldsymbol{y}|\boldsymbol{x}), \ldots, p_{\boldsymbol{c}_K}(\boldsymbol{y}|\boldsymbol{x})]$, and the approximation error $\mathcal{E}$ is bounded by $\mathcal{E} < \epsilon |\log \frac{\eta(1-\alpha)}{2(1-\eta)\alpha}|$.*

*Proof.* For a given question $\boldsymbol{x}$, assume we have collected a set of $\boldsymbol{y}$ by Eq.(7), which forms an empirical distribution $q(\mathbf{y}|\boldsymbol{x})$ for the target culture, *e.g.*, Japan, we aim to find the best $q$ with minimal overlap with non-target cultures, *e.g.*, Korea, Singapore and UK, $\boldsymbol{c}_1, \ldots, \boldsymbol{c}_K$. We optimize:

$$q^*(\mathbf{y}|\boldsymbol{x}) = \operatorname*{argmax}_{\boldsymbol{q}} GJS_{\alpha, \mathbf{w}}[q(\mathbf{y}|\boldsymbol{x}), p_{\boldsymbol{c}_1}(\mathbf{y}|\boldsymbol{x}), \ldots, p_{\boldsymbol{c}_K}(\mathbf{y}|\boldsymbol{x})], \tag{17}$$

where GJS is Generalized Jensen divergence, and $\alpha$, $\boldsymbol{w} = (w_1, \ldots, w_K) > 0$ are weights for each distribution with $\alpha + \sum_{i=1}^{K} w_i = 1$.

For brevity, we omit $\boldsymbol{x}$. Since $1 - \alpha = \sum_{i=1}^{K} w_k$, define $\beta_i = \frac{w_i}{1-\alpha}$, and the average non-target culture distribution as $\hat{p} = \frac{\sum_{i=1}^{K} \boldsymbol{w}_i p_{\boldsymbol{c}_i}}{1-\alpha}$, we have $m = \alpha p + \sum_{i=1}^{K} w_i p_{\boldsymbol{c}_i}$, and thus $\text{GJS}_{\alpha,\mathbf{w}}[q, p_{\boldsymbol{c}_1}, \ldots, p_{\boldsymbol{c}_K}] = \alpha \text{KL}[q||m] + (1-\alpha)\sum_{i=1}^{K} \beta_i \text{KL}[p_{\boldsymbol{c}_i}||m] = \alpha q + (1-\alpha)\hat{p}$. Then, we further have:

$$\text{GJS}_{\alpha,\mathbf{w}}[q, p_{\boldsymbol{c}_1}, \ldots, p_{\boldsymbol{c}_K}]$$

$$= \alpha \mathbb{E}_q[\log q - \log m] + (1-\alpha)\sum_{i=1}^{K} \beta_i \mathbb{E}_{p_{\boldsymbol{c}_i}}[\log p_{\boldsymbol{c}_i} - \log m]$$

$$= \alpha \mathbb{E}_q[\log q - \log m] + (1-\alpha)\left[\mathbb{E}_{\hat{p}}\log\hat{p} - \sum_{i=1}^{K} \beta_i \mathbb{E}_{p_{\boldsymbol{c}_i}}\log m + \sum_{i=1}^{K} \beta_i \mathbb{E}_{p_{\boldsymbol{c}_i}}\log p_{\boldsymbol{c}_i} - \mathbb{E}_{\hat{p}}\log\hat{p}\right]$$

$$= \alpha \mathbb{E}_q[\log q - \log m] + (1-\alpha)\left[\mathbb{E}_{\hat{p}}\log\hat{p} - \sum_{i=1}^{K} \beta_i \mathbb{E}_{p_{\boldsymbol{c}_i}}\log m + \text{GJS}_{\boldsymbol{\beta}}[p_{\boldsymbol{c}_1}, \ldots, p_{\boldsymbol{c}_K}]\right]$$

$$= \alpha \text{KL}[q||m] + (1-\alpha)\text{KL}[\hat{p}||m] + (1-\alpha)\text{GJS}_{\boldsymbol{\beta}}[p_{\boldsymbol{c}_1}, \ldots, p_{\boldsymbol{c}_K}]$$

$$= \text{GJS}_\alpha[q, m] + (1-\alpha)\text{GJS}_{\boldsymbol{\beta}}[p_{\boldsymbol{c}_1}, \ldots, p_{\boldsymbol{c}_K}]$$

$$\geq \text{GJS}_\alpha[q, m]. \tag{18}$$

Once the mix weight $\boldsymbol{w}$ is determined, $\text{GJS}_{\boldsymbol{\beta}}[p_{\boldsymbol{c}_1}, \ldots, p_{\boldsymbol{c}_K}]$ only relies on $p_{\boldsymbol{c}_1}, \ldots, p_{\boldsymbol{c}_K}$, irrelevant to $q$. Therefore, we only maximize $\text{GJS}_\alpha[q, m]$. However, each true $p_{\boldsymbol{c}_i}$ is unknown. To maximize it, we further define a binary variable $\mathbf{s} \in \{0, 1\}$, which indicates the source of a given response $\boldsymbol{y}$ for a fixed question $\boldsymbol{x}$. When $\boldsymbol{y} \sim q(\boldsymbol{y}|\boldsymbol{x})$, $\mathbf{s} = 0$, when $\boldsymbol{y} \sim \hat{p}(\boldsymbol{y}|\boldsymbol{x})$, $\mathbf{s} = 1$. We then maximize the mutual information $I(\mathbf{s}; \mathbf{y}|\mathbf{x} = \boldsymbol{x})$. We then also have:

$$I(\mathbf{s}; \mathbf{y}|\mathbf{x} = \boldsymbol{x})$$

$$= p(\mathbf{s} = 0|\boldsymbol{x})\text{KL}[p(\mathbf{y}|\mathbf{s} = 0, \boldsymbol{x})||p(\mathbf{y}|\boldsymbol{x})] + p(\mathbf{s} = 1|\boldsymbol{x})\text{KL}[p(\mathbf{y}|\mathbf{s} = 1, \boldsymbol{x})||p(\mathbf{y}|\boldsymbol{x})]$$

$$= \alpha \text{KL}[q(\mathbf{y}|\boldsymbol{x})||p(\mathbf{y}|\boldsymbol{x})] + (1 - \alpha))\text{KL}[\hat{p}(\mathbf{y}|\boldsymbol{x})||p(\mathbf{y}|\boldsymbol{x})]$$

$$= \alpha \text{KL}[q(\mathbf{y}|\boldsymbol{x})||m(\mathbf{y}|vx)] + (1 - \alpha))\text{KL}[\hat{p}(\mathbf{y}|\boldsymbol{x})||m(\mathbf{y}|vx)]$$

$$= \text{GJS}_\alpha[q, m]. \tag{19}$$

Therefore, maximizing $I(\mathbf{s}; \mathbf{y}|\mathbf{x} = \boldsymbol{x})$ is equivalent to maximizing $\text{GJS}_\alpha[q, m]$.

By the Barber–Agakov bound (Barber & Agakov, 2004), we have

$$I(\mathbf{s}; \mathbf{y}|\mathbf{x} = \boldsymbol{x}) \geq \mathbb{E}_{p(\mathbf{y}|\boldsymbol{x})}\mathbb{E}_{p(\mathbf{s}|\mathbf{y},\mathbf{x})}[\log q_\phi(\mathbf{s}|\mathbf{y}, \boldsymbol{x}) - \log p(\mathbf{s}|\boldsymbol{x})]. \tag{20}$$

By also fixing a given $\mathbf{y}$, we have a point-wise mutual information estimation as:

$$\hat{I}(\boldsymbol{s}, \boldsymbol{y}|\boldsymbol{x})$$

$$\geq p(\mathbf{s} = 0|\boldsymbol{y}, \boldsymbol{x})\left[\log \frac{q_\phi(\mathbf{s} = 0|\boldsymbol{y}, \boldsymbol{x})}{q_\phi(\mathbf{s} = 1|\boldsymbol{y}, \boldsymbol{x})} + \log \frac{1-\alpha}{\alpha}\right] + \log \frac{q_\phi(\mathbf{s} = 1|\boldsymbol{y}, \boldsymbol{x})}{2}$$

$$\approx \boldsymbol{\phi}(\boldsymbol{y}, \boldsymbol{x})\left[\log \frac{\boldsymbol{\phi}(\boldsymbol{y}, \boldsymbol{x})}{1 - \boldsymbol{\phi}(\boldsymbol{y}, \boldsymbol{x})} + \log \frac{1-\alpha}{2\alpha}\right] + \log(1 - \boldsymbol{\phi}(\boldsymbol{y}, \boldsymbol{x}))$$

$$= \Gamma(\mathbf{y}), \tag{21}$$

where $q_\phi$ is a classifier parameterized by $\phi$, $e.g.$, GPT-5, to predict whether $\mathbf{y}$ is from the reference culture distribution, and we abbreviate it as $\boldsymbol{\phi}(\boldsymbol{y}, \boldsymbol{x})$. Since the true probability $p(\mathbf{s} = 0|\boldsymbol{y}, \boldsymbol{x})$ is unknown, we also approximate it with $\boldsymbol{\phi}(\boldsymbol{y}, \boldsymbol{x})$.

From the derivation above, we conclude that optimizing $\Gamma(\mathbf{y})$ is is equivalent to optimizing a point-wise lower bound of $\text{GJS}_{\alpha,\mathbf{w}}[q, p_{\boldsymbol{c}_1}, \ldots, p_{\boldsymbol{c}_K}]$. Assume the error of this classifier $|\boldsymbol{\phi}(\boldsymbol{y}, \boldsymbol{x}) - p(\mathbf{s} = 0|\boldsymbol{y}, \boldsymbol{x})| < \epsilon$ and the classifier is not over-confident, $i.e.$, $\boldsymbol{\phi}(\boldsymbol{y}, \boldsymbol{x}) < \eta$, we can easily have the approximation error $< \epsilon|\log \frac{\eta(1-\alpha)}{2(1-\eta)\alpha}|$.

We use two iterative steps to optimize $\mathcal{S}(\boldsymbol{x}, \boldsymbol{y})$.

**Question Generation Step** At the first iteration, we generate questions from scratch. In later iterations, we fix the optimal sampled response $y$ and refine $x$ to optimize $\mathcal{S}(\boldsymbol{x}|\boldsymbol{y})$. This step mainly involves: i) enhancing $p_{\boldsymbol{c}}(\boldsymbol{x})$, ii) the representativeness of $x$, and iii) the possibility of $x$ that can increase the distinctiveness.

**Response Generation Step** We fix the question and generate the optimal response $y$. This step mainly involves: i) enhancing $p_{\boldsymbol{c}}(\boldsymbol{y}|\boldsymbol{x})$, ii) the representativeness of $(x, y)$; iii) the distinctiveness.

Table 4: Notation Table

| Variable | Description |
| --- | --- |
| $\mathbf{x}$ | the question |
| $\mathbf{y}$ | the response |
| $(\mathbf{x}, \mathbf{y})$ | a cultural data sample |
| $\theta$ | parameters of a target LLM to be aligned |
| $p_{\boldsymbol{\theta}}(\mathbf{y}|\mathbf{x})$ | an LLM to be aligned |
| $p_{\mathbf{c}}(\mathbf{x}, \mathbf{y})$ | the true data distribution of the target culture $c$ |
| $p_{\mathbf{c}_k}(\mathbf{x}, \mathbf{y})$ | the true data distribution of the non-target culture $c_k$ |
| $K$ | the number of non-target cultures |
| $q_{\boldsymbol{c}}^*$ | the goal set of cultural data, with satisfactory representativeness and distinctiveness |
| $p_{\boldsymbol{\omega}}$ | an LLM that simulates an individual with cultural competence of the target culture $\boldsymbol{c}$ |
| $p_{\boldsymbol{\omega}_i}$ | the i-th LLM for simulation |
| $I(\mathbf{c}; \mathbf{y}|\mathbf{x})$ | the mutual information between $\mathbf{c}$ and $\mathbf{y}$ given $\mathbf{x}$ as the condition |
| $\Delta_{\boldsymbol{c}}^{\boldsymbol{\omega}}(y|x)$ | the opproximated representativeness score by an LLM $p_{\boldsymbol{\omega}}$ |
| $\Delta_{\boldsymbol{c}}^{\boldsymbol{\omega}_1,\dots,\boldsymbol{\omega}_M}(y|x)$ | the opproximated representativeness score by the group of simulating LLMs |
| $M$ | the number of cultural individuals participating in consensus elicitation |
| $\Gamma_{\boldsymbol{c}_1,\dots,\boldsymbol{c}_K}(\boldsymbol{y}|\boldsymbol{x})$ | the opproximated distinctiveness score |
| $\phi(x, y)$ | the classifier that estimates the probability that $y$ does NOT come from $\boldsymbol{c}_1, \dots, \boldsymbol{c}_K$ |
| $\text{sim}(\cdot, \cdot)$ | cosine similarity between text embeddings |
| $\epsilon$ | the classifier error |
| $\eta$ | the classifier confidence |
| $\mathcal{E}$ | the approximate error bound of Propasiotion 2 |
| $S_{\boldsymbol{c}}(\boldsymbol{y}|\boldsymbol{x})$ | the score of $\boldsymbol{y}$ for data optimization |
| $\lambda_1, \lambda_2, \lambda_3$ | hypermarameters to trade-off representativeness and distinctiveness |
| $\{x_i^0\}_{i=1}^N$ | the initial question set for cultural data optimization |
| $t$ | the iteration index |
| $N$ | the total number of samples for optimization |
| $n$ | the number of candidate responses generated in each optimization step |
| $m$ | the number of candidate questions generated in each optimization step |

# B SUPPLEMENTS FOR CARDSET DATA CONSTRUCTION

## B.1 SUPPLEMENTS FOR CULTURAL TOPICS

We construct a cultural framework through integrating diverse definitions of cultures from multiple disciplines such as ethics and value. The framework contains diverse topics as follows.

**I. Cultural Values**

- **Schwartz's Theory of Basic Values**: Self-direction, Stimulation, Hedonism, Achievement, Power, Security, Tradition, Conformity, Benevolence, and Universalism.

- **Hofstede Cultural Dimensions** (Hofstede & Hofstede, 2005): Power Distance Index, Individualism vs. Collectivism, Uncertainty Avoidance Index, Masculinity vs. Femininity, Long-Term Orientation, and Indulgence vs. Restraint.
- **World Value Survey** (AlKhamissi et al., 2024): Social Values, Attitudes & Stereotypes, Happiness and Well-being, Social Capital, Trust & Organizational Membership, Economic Values, Corruption, Migration, Security, Neighborhood Safety & Disorder, Postmaterialist Index, Science & Technology, Religious Values, Ethical Values and Norms, Political Interest & Political Participation, Political Culture & Political Regimes.

Definition about these value dimensions can be referred to the corresponding theory.

**II. Social Norms**

- **Gender Roles**: Refers to cultural expectations and behaviors assigned to genders. Key elements include roles in the family, workplace, and society, as well as attitudes toward gender equality and stereotypes.
- **Respect Elders**: Explores how elders are treated and regarded in society. Key elements include deference, caregiving, decision-making authority, and intergenerational relationships.
- **Family Obligations**: Refers to the responsibilities and expectations individuals have toward their family, including financial support, caregiving, and prioritizing family over personal needs.
- **Justice and Fairness**: Encompasses cultural attitudes toward fairness, equality, and the application of justice. Key elements include perceptions of legal systems, social equality, and ethical decision-making.
- **Individual Rights**: Individual Rights [Ethics and Norms]: Focuses on the emphasis placed on personal freedoms, autonomy, and individual rights within society. Key elements include freedom of speech, privacy, and access to opportunities.
- **Social Norms**: Refers to unwritten rules and expectations governing appropriate behavior in social settings. Key elements include dress codes, public behavior, and communication styles.
- **Moral Duties and Altruism**: Explores the cultural emphasis on moral obligations and selfless acts for the welfare of others. Key elements include charity, volunteerism, and moral responsibility.
- **Environmental Ethics**: Refers to cultural attitudes and practices toward nature and the environment. Key elements include sustainability, conservation, and ecological responsibility.

**III. Behavioral Practices**

- **Social Relationship**: Examines the relationships within different social groups, including family, friends, colleagues, acquaintances, and strangers. Key elements include hierarchy, trust, intimacy, and obligations.
- **Work Behaviors**: Focuses on behaviors, hierarchies, and expectations in professional and business environments. Key elements include authority, teamwork, and professional etiquette.
- **Economic Behaviors**: Explores cultural attitudes toward money, wealth, and economic activities. Key elements include saving habits, spending patterns, and attitudes toward entrepreneurship.
- **Education System and Relationship**: Explores the structure, relationships, and norms within educational institutions, such as schools. Key elements include authority, learning methods, and examination systems.
- **Religious and Ceremonial Behaviors**: Rituals, festivals, and traditions tied to religious or secular practices. Key elements include rites of passage, community celebrations, and individual practices.

B.2 SUPPLEMENTS FOR OPTIMIZATION PROMPTS

CAReDiOis an in-context data optimization framework, without any training. The primary prompts used in the framework are illustrated in the following figures. Corresponding to Algorithm 1,

- Fig 6 shows the prompt for question initialization with the Self-Instruct approach (Line 1).

You are a {country} culture expert with deep knowledge of cultures, values and social practices of {country} and other countries.

Given a cultural value topic from the Schwartz Theory of Basic Values, Hofstede's Cultural Dimensions, or World Value Survey, etc, your task is to generate 3-5 diverse questions or scenarios around this topic to explore the representative values, beliefs and social norms of {country} culture.

You should follow these guidelines for generation:
1. Topic Relevance: Question should explore cultural characteristics around the given topic.
2. Culture Consensus: Question should focus on values, beliefs and social practices that are representative, widely accepted and frequently appeared in {country} culture. Avoid niche, rare, or extreme viewpoints.
3. Culture Representativeness: Question should capture generalized values, beliefs and social practices of {country} culture that can be guidance across various situations, not be too specific to a single event or context.
4. Diversity: Each generated question should cover different aspects of the topic and differ from the provided example questions.
5. Neutral phrasing: Do not explicitly mention '{country}' in the question, so that it can be used to interview people from any cultural backgrounds to uncover the distinctiveness, i.e., avoid using like "In {country}, how do people...?" or "In {country} culture, what is the view on...?" in the question.

Generated questions should be one of the following types:
- Scenario-based question: Present a realistic value-related scenario and ask how people might react.
- Value-oriented question: Ask how people might prioritize or value certain principles or beliefs.
- Open-ended question: Encourage people to share their thoughts, feelings, and experiences related to a cultural aspect in detail.
- Likert-scale attitude question: Ask respondents to indicate their degree of agreement or priority regarding specific cultural value statements, like such as questions in World Value Survey or other surveys.

Here is the cultural topic that you need to generate questions for. (The provided definition is only a reference but not a complete description, you can further explore it based on your knowledge)
Cultural Topic: {topic}

Example Questions
[example 1]: {example_1}
[example 2]: {example_2}

Please generate 3-5 new questions following the guidelines above. Output exactly in the following format, without any other text.
[question 1]: <text of the question> (scenario-based question / value-oriented question / open-ended question / Likert-scale attitude question)
[question 2]: <text of the question> (...)
... (up to 5 questions)

The generated questions are:
"""

Figure 6: Prompt for initializing questions using the Self-Instruct approach.

- Fig. 7 shows the prompt for response generation, maximizing representativeness and distinctiveness (Line 4).

- Fig. 8 shows the prompt for point-wise MI calculation in Eq.(2) (Line 5).

- Fig 9 includes the prompts for role-playing of individuals in three types.

- Fig 10 presents the prompt for question revision in Line 7.

We will also open-source the codes and synthetic datasets for reproducibility.

### B.3 SUPPLEMENTS FOR QUESTION FORMATS

To align with the practical usage of LLMs, we consider four common question formats in our synthesis process, listed as follows.

- Scenario-based question: Present a realistic value-related scenario and ask how people might react.

You are a {role-play instruction}, thus having deep knowledge of {country} culture values, beliefs and social practices.

Given a culture-related question, please answer it with your expertise and knowledge about {country} culture. You should follow the guidelines below:
1. Highlight points that are related to the question, representative and widely accepted in {country} culture.
2. Highlight aspects that are representative and generalized, not specific to a single event or context.
3. Emphasize points that are distinctive and unique to {country} culture, compared to other cultures. To help you think more deeply, several responses from people of other countries are provided. Reflect on these responses and consider how {country} culture might approach this topic differently.
4. Focus only on the cultural answer; do not restate the question, explain the task, or mention this prompt.
5. Answer the question within 150 words, then output the answer text only, with no additional commentary.

Here is the question:
{question}

Responses from other countries:
{other_response}

Your response is:"""

Figure 7: Prompts for generating representative and distinctive responses.

You are a {role-play instruction}, thus having deep knowledge of {country} culture values, beliefs and social practices.

Your task is to judge how representative the provided cultural points to a culture-related question are to {country} culture **from your own viewpoints as this expert**.

Here is the culture-related question:
{question}

Cultural points:
{cultural_points}

Please evaluate the representativeness of each cultural point to {country} culture using a 5-likert scale: 1 (not representative at all), 2 (slightly representative), 3 (somewhat representative), 4 (mostly representative), 5 (very representative)

Figure 8: Prompts for scoring the point-wise MI for each response.

- Value-oriented question: Ask how people might prioritize or value certain principles or beliefs.

- Open-ended question: Encourage people to share their thoughts, feelings, and experiences related to a cultural aspect in detail.

- Likert-scale attitude question: Ask respondents to indicate their degree of agreement or priority regarding specific cultural value statements, like such as questions in World Value Survey or other surveys.

### General people with various demographics sampled from WVS

*"Now you are a **{country}** citizen with the following demographic profile: **{demographic_info}** You understand **{country}** culture based on your own experiences and viewpoints as this persona, including the cultural values, beliefs and social practices. Your task is ..."*

### Cultural experts, such as sociologists and cultural psychologist

*"You are a **{country}** culture expert, with 15 years of experience as a **{role}.** Thus, you have deep knowledge of **{country}** culture values, traditions and social practices. Your task is to ... from your own viewpoints as this expert"*

### Cross-cultural researchers

*"You are a **{foreign}** culture expert, with 15 years of experience as a cross-cultural researcher. Thus, you have deep knowledge of cultural values, beliefs and social practices of various cultures across the world. Your task is to ... from your own viewpoints as this foreign culture expert"*

Figure 9: Prompts for role-playing of individuals in three types, used in the point-wise MI calculation.

You are a {role-play instruction}, thus having deep knowledge of {country} culture values, beliefs and social practices.

Given a culture-related question under a topic, you need to refine it so that it can better elicit widely accepted, representative and distinctive features of {country} culture towards this question compared to other cultures.

To refine the question to achieve both a higher representativeness reward and a higher distinctiveness reward, you should consider several improvement directions:
1. Cultural Consensus: Analyze both the high-scoring and low-scoring cultural points to identify which are more widely accepted features of {country} culture, then, refine the question to i) keep the high-scoring points, ii) avoid low-scoring points, and iii) encourage new relevant high-scoring points that are not yet covered.
2. Cultural Representativeness: Ensure the question is not tied to a single event or context, but capture points generalized across various situations.
3. Cultural Distinctiveness: Identify features where {country} significantly differ from other cultures, then refine the question to better elicit these distinctive features while avoiding points common across cultures.
4. Question-Answer Consistency: Ensure the question is coherent with the high-scoring points, avoid overly vague or generic wording.

Here is the input:
{Information about responses generated in this iteration}
{Representativeness score for each response}
{Distinctiveness score for each response}

Please refine the above question.

Figure 10: Prompts for refining questions.

## C   SUPPLEMENTS FOR METHOD

### C.1   FEASIBILITY AND EFFECTIVENESS OF CONSENSUS ELICITION

To obtain cultural data with higher representativeness, we draw on Cultural Consensus Theory (CCT) (Weller, 2007) and approximate this objective as a consensus elicitation problem. This approximation relies on two assumptions: (1) each LLM $p_\omega$ possesses sufficient cultural competence to identify responses that reflect the target culture; and (2) one single LLM may encode incomplete and biased knowledge of the target culture, whereas multiple LLMs prompted with different personas

$p_{\omega_1}, \ldots, p_{\omega_M}$ can help extract cultural knowledge from diverse perspectives and vote for a more representative consensus, mitigating bias. Below, we discuss the rationale behind these assumptions and empirically validate both.

(1) **Prompting an LLM to simulate cultural roles can indeed assign it with higher cultural competence**. Known as In-Context Alignment (ICA), conditioning an LLM with attributes, such as role, persona and culture, has been a widely adopted method in cultural alignment and shown efficacy (Durmus et al., 2023; Kwok et al., 2024; Kharchenko et al., 2024; Choenni & Shutova, 2024). Moreover, our experiments in Table 3 also demonstrate that the Role-Play baseline significantly improves the original base model, e.g., $57.75 \rightarrow 68.62$ on Gemma-3-27B-IT, showing that ICA indeed enhances an LLM's cultural competence.

(2) **The performance of cultural alignment can be further enhanced by extracting the consensus from multiple role-playing LLMs.** Using GPT-5 as the backbone LLM, for each culture, we prompt it to simulate seven roles with distinct personas: i) three general people with different demographics randomly sampled from the WVS data of the target culture; ii) three cultural experts such as sociologist; and iii) one cross-cultural researcher who comes from another culture but studies global cultural patterns. First, we evaluate these personas individually on GlobalOpinionQA and measure their mutual agreement in Table 4. Then, we extract their consensus on each test question and compare the performance between individual roles and the consensus results in Table 6. Two key observations emerge:

- **Different roles show divergent cultural perceptions.** We observe obvious differences in the outputs of distinct roles. This indicates that incorporating multiple personas can help activate diverse cultural knowledge.
- **Consensus of multiple roles further improves stability and accuracy.** Obviously, aggregating multiple roles yields better and more stable (lower std across cultures) results, with especially larger gains in low-resource settings, e.g., Russia and Poland, where model biases are more prominent.

Table 5: The average and minimum agreement among multiple roles.

|  | United States | Germany | Japan | Russia | Poland | Mexico | Nigeria |
|---|---|---|---|---|---|---|---|
| Average Agreement | 0.86 | 0.85 | 0.81 | 0.80 | 0.81 | 0.79 | 0.80 |
| Minimum Agreement | 0.83 | 0.78 | 0.72 | 0.69 | 0.74 | 0.70 | 0.74 |

Table 6: Performance comparison between single LLM and multi-LLM consensus on the GlobalOpinionQA dataset, using GPT-5 as the LLM backbone.

| Method | Avg. | Std | United States | Germany | Japan | Russia | Poland | Mexico | Nigeria |
|---|---|---|---|---|---|---|---|---|---|
| Single Role | 58.63 | 2.758 | 64.5 | 62.81 | 56.47 | 56.65 | 59.71 | 59.4 | 58.84 |
| Multi Consensus | 59.92 | 2.573 | 65.17 | 64.99 | 58.24 | 59.21 | 62.63 | 60.45 | 59.94 |
| $\Delta$ | 1.29 | - | 0.67 | 2.18 | 1.77 | 2.56 | 2.92 | 1.05 | 1.10 |

These results collectively demonstrate the effectiveness reliability of consensus elicitation (CCT) from multiple LLMs. Moreover, note that this process does not rely on perfect persona simulation. We recognize precisely simulating individuals is an unresolved challenge, but it is NOT our purpose. Instead, CCT only assumes the presence of diverse agents with sufficient cultural competence to identify shared cultural knowledge. The experiments above show that such approximated persona playing is sufficient to obtain cultural improvement.

## C.2 Empirical Verification of Cultural Classifier and Error Bound

**Error bound of Classifier** Since the true culture distributions are unattainable, we introduce a classifier, which estimates the probability that a response $y$ does NOT come from any of the $K$ non-target cultures, and approximate the distinctiveness objective using Eq.(4). Correspondingly, we provide Proposition 2 as i) an interpretation of Eq.(4), showing that our method truly exploits the differences between true cultural distributions, and ii) a guarantee that our approximated implementation indeed

Table 7: Error bound and accuracy of cultural classifiers implemented with different architectures.

| | Cultural Response Similarity | | | | LLM-as-Judge (GPT-5-Thinking) | | | |
| | United States | China | Japan | Poland | United States | China | Japan | Poland |
| --- | --- | --- | --- | --- | --- | --- | --- | --- |
| Accuracy | 0.9051 | 0.9350 | 0.9450 | **0.9350** | **0.9650** | **0.9550** | **0.9700** | 0.9100 |
| F1-score | 0.9064 | 0.9366 | 0.9442 | **0.9319** | **0.9648** | **0.9577** | **0.9714** | 0.9100 |
| $\epsilon$ | **0.1682** | 0.1803 | **0.0973** | **0.0964** | 0.1981 | **0.1358** | 0.1206 | 0.1635 |
| $\eta$ | 0.9519 | 0.9559 | 0.9304 | 0.9675 | 1.0000 | 1.0000 | 1.0000 | 0.8490 |
| $\mathcal{E}$ | **0.0160** | **0.0330** | **0.0290** | **0.0485** | 0.7952 | 0.5452 | 0.5061 | 0.1902 |

optimizes the distinctiveness objective. Although Prop. 2 holds in theory, it also indicates that the tightness of the approximation error bound $\mathcal{E}$ depends on two classifier-related factors. We therefore conduct empirical verification about whether these conditions hold in practice.

First, we construct an evaluation set of 800 samples $(\boldsymbol{x}, \boldsymbol{y}, p^*, \boldsymbol{c})$ covering four distinct cultures (200 samples for each), where $(\boldsymbol{x}, \boldsymbol{y})$ are question-response pairs sampled from our synthetic data and $p^*$ are annotated and averaged across three native speakers per culture. Following the consensus-rating protocol described in Appendix D.3, we use a 5-point scale and convert the rating score into $p^* \in [0, 1](1 \rightarrow 0.1, 2 \rightarrow 0.3, 3 \rightarrow 0.6, 4 \rightarrow 0.85, 5 \rightarrow 0.95)$. Using the classifier $\phi(\boldsymbol{x}, \boldsymbol{y})$ implemented by $p_{\boldsymbol{\omega}}$ and the openai text-embedding-3-small model, we measure the error bound $\epsilon$ and confidence $\eta$ for each culture. The classifier error $\epsilon$ is calculated as the average $|\phi(x, y) - p^*|$ across the 200 samples, and the classifier confidence $\eta$ as the max value of $\phi(x, y)$.

As shown in Table 7, *the approximation error $\mathcal{E}$ is smaller than 0.05 for most countries*. This is very small compared to the typical range of GJS, which spans $[0, \log K]$ for $K + 1$ distributions ($K = 14$ in this work, and the maximum of GJS is 2.64). Thus, the approximation is sufficiently accurate for practical use. Besides, the efficacy of Eq.(4) (distinctiveness optimization) has also been empirically verified by the strong overall performance of CAReDiO across all benchmarks, further supporting that Proposition 2 holds in practice.

**Accuracy and Sensitivity of Classifier**    We further conduct experiments to assess the reliability of our classifier. Besides, we claim that the classifier can be implemented in multiple ways, thus we also compare the current embedding-based version with another alternative, i.e., llm-as-judge using GPT-5-thinking as the backbone. The results are shown in Table 7, where we report five metrics: i) average probability error $\epsilon = |\phi(x, y) - p^*|)$; ii) maximum confidence $\eta$; iii) the approximation error bound $\mathcal{E}$ mentioned in Proposition 2; iii) classification accuracy, Acc.; and iv) classification F1.

We can see: a) *Our simple classifier based on cultural response similarity is highly accurate (with F1 > 0.9)*. We infer this is because semantic differences between cultures are not small, even simple methods perform well, while previous work has largely ignored distinctiveness. b) Note that in Eq.(4) we do not use the binary prediction label, but rather the probability $\phi(x, y)$. Therefore, *the approximation error bound $\mathcal{E}$ matters more, which is very small (smaller than 0.05) compared to its upper bound (2.64) in this work.* c) *LLM-as-judge with the advanced GPT-5-Thinking achieves a little better accuracy and F1, but also larger probability error $\epsilon$*, and thus leads to much worse error bound $\mathcal{E}$. This is because these off-the-shelf LLMs often fail to provide continuous score/probability.

Overall, these results above demonstrate that our simple classifier is sufficiently stable and accurate for our optimization objective. It is also worth noting that $\phi(x, y)$ can be implemented in different architectures. This is not our core contribution, but just an implementation of our novel optimization method (Algorithm 1). We can explore more structures in the future.

# D    SUPPLEMENTS FOR EXPERIMENTAL SETTINGS

## D.1    MORE DETAILS ABOUT BENCHMARKS

We introduce comprehensive benchmarks of various categories for extensive evaluation, each with distinct evaluation protocols and metrics. We present the details as follows.

(1) **Value Questionnaires Evaluation**.

- **GlobalOpinionQA** (Durmus et al., 2023): This dataset compiles 2,556 items from cross-national value questionnaires, i.e., Global Attitudes surveys (GAS, about public opinion, social issues and demographic trends in the U.S. and worldwide) and World Value Survey. GAS covers topics like politics, media, technology, religion, race and ethnicity; while WVS focuses on people's beliefs and values across the world, how these beliefs change over time, and the social and political impact of these beliefs. Each item presents an opinion-related question with multiple answer choices, along with the probability distribution of choices across various countries. For evaluation, we compate the accuracy of the model's prediction based on the ground truth.

- **WVS** (Haerpfer et al., 2020): This is a public questionnaire that investigates people's values across 13 topics, such as social values, attitudes and stereotypes. It collects real responses from people across different countries. We compute the *alignment* between an LLM $P_\theta$ and a culture $C$ following the metric in (Xu et al., 2024a): $Align(P_\theta, C) = (1 - \frac{\text{Euclidean}(\mathcal{A}_{P_\theta}, \mathcal{A}_C)}{max_d istance}) \times 100$. $\mathcal{A}_{P_\theta}$ and $\mathcal{A}_C)$ denotes the model's and human's answers on all questions respectively, and 'max_distance' is the maximum possible option difference used for normalization.

(2) **Multiple-Choice Cultural Knowledge Evaluation**.

- **CulturalBench** (Chiu et al., 2024): This manual dataset contains 1,227 four-choice questions for assessing LLMs' cultural knowledge, spanning 45 regions and 17 cultural topics. We adopt its CulturalBench-Hard version which transforms each multi-choice item into four binary true/false questions and requires the LLM to evaluate all options correctly. Accuracy is calculated on the ground truth.

(3) **Open-ended Question Evaluation**

- **Prism** (Kirk et al., 2025): This dataset includes real conversations between 1,500 diverse participants from 75 countries and 21 LLMs. We filter a subset of questions for evaluation based on two criteria: i) the question is explicitly or potentially related to cultural topics such as relationship management and discussion on abortion; and ii) several cultures exhibit clear differences in responses. We also use GPT-4o to evaluate the culture-awareness of the responses, from 1 to 5.

### D.2 LICENSE OF DATASETS

GlobalOpinionQA (Durmus et al., 2023) is under cc-by-nc-sa-4.0 license. CulturalBench (Chiu et al., 2024) is under cc-by-4.0 license. And Prism (Kirk et al., 2025) is under cc license. CultureBank (Shi et al., 2024) is under MIT license.

### D.3 DETAILS ABOUT HUMAN EVALUATION

**Human Recruitment** We recruited three native annotators for each country through a professional vendor company. Annotators were selected to vary in gender and age group to enhance labeling diversity and quality. Each annotator was compensated between \$7.5–\$15 per hour according to their local income level, which is substantially higher than the minimum wage in their respective regions. This annotation project underwent a full review and received approval from the Institutional Review Board (IRB).

**Annotation Guidance** We designed two annotation tasks: 1) Accuracy (1-5), which means the consensus level of this data to the culture; 2) Saliance(1-3), representativeness and importance of the cultural aspect. The human evaluation task in Section 4.3 also follows the "Accuracy" annotation task. Their criteria are as follows.

**Consensus Rating (1–5)**:

- **1 – Conflict / Mismatch**: The response conflicts with or deviates from the mainstream values, norms, or behaviors of the target culture, or reflects the core values of a different culture.

- **2 – Neutral / Generic**: The response is internationalized or culturally neutral, usable across cultures but lacking distinctive features of the target culture.

- **3 – Moderate Fit, With Cultural Cues**: The response aligns with the target culture and contains relevant cultural references, but details are limited, and the expression remains generic or templated.

- **4 – Good Fit, With Distinct Features**: The response explicitly reflects core cultural characteristics (e.g., linguistic style, value preferences, or contextual knowledge), with added details or examples, and no cultural misunderstandings.

- **5 – Excellent Fit**: The response provides accurate, detailed, and nuanced representation of the culture's core values, beliefs, or practices. It is highly natural and satisfactory from the perspective of a cultural insider.

**Importance Rating (1–3)**:

- 1 – **Low Representativeness**: The content is culturally neutral, irrelevant, or represents marginal/secondary aspects of the culture.

- 2 – **Moderate Representativeness**: The content aligns with the culture and is thematically related, but reflects secondary or surface-level aspects (e.g., customs like food or clothing) rather than core values.

- 3 – **High Representativeness**: The content captures central cultural values, core beliefs, or highly representative features of the target culture.

Table 8: Inter-annotator agreement on different methods across cultures.

| IAA | China | Japan | Poland |
|---|---|---|---|
| Role-Play (Qwen2.5-7B) | 0.879 | 0.596 | 0.654 |
| Role-Play (GPT-4.1) | 0.867 | 0.571 | 0.624 |
| CulturePark | 0.848 | 0.652 | 0.752 |
| CAReDiO | 0.806 | 0.880 | 0.795 |
| Avg | 0.850 | 0.675 | 0.706 |

**Quality Control** Since this task is inherently subjective, we do not enforce a single "correct" label. Instead, we focus on minimizing noise due to the inconsistent understanding of the rubric. To this end, we provide annotators with the above fine-grained scoring rubrics and conduct mandatory training sessions before annotation. This ensures that disagreements stem from genuine cultural differences rather than misunderstandings of the guidelines.

We evaluate inter-annotator agreement across tasks and cultures, reporting the results in Table 8. The average pairwise agreement reaches 85%, while full agreement across all three annotators was achieved in approximately 65–70% of cases, indicating a strong agreement for subjective NLP/LLM evaluation tasks (with most values exceeding 0.7). These results demonstrate that our annotation process is reliable and that annotator divergence remains within an acceptable and meaningful range. For each sample, we applied a majority-vote strategy to merge individual ratings into the final label.

### D.4 MORE DETAILS ABOUT BASELINES

**Role-Play**: We instruct LLMs to simulate individuals from specific cultural backgrounds by providing a system prompt *"You are a {country} chatbot that knows {country} culture very well. Please answer the following questions according to your knowledge about the culture of country.".*

**Culturally Fine-tuned LLMs**: Recent studies about cultural alignment fall into this category, all of which depend on supervised fine-tuning but collect training data in different ways.

- CultureLLM (Li et al., 2024a) employs 50 questions from the World Value Survey (WVS) with answers of the corresponding culture as seed data and augment semantically equivalent samples for training using a powerful LLM.

- CulturePark (Li et al., 2024b) builds an LLM-powered multi-agent communication framework, where agents playing roles of different cultures discuss about the topics from World Value Surveys thus high-quality cultural data is collected.

- CultureSPA (Xu et al., 2024a) uncovers representative data of specific cultures by activating the LLM's internal culture knowledge. It first synthesizes survey questions across cultural topics and identify the data that are different with culture-unaware and culture-aware prompting.

- CultureBank (Shi et al., 2024) collects self-narratives of diverse culture-aware scenarios such as working, immigration and traveling from the online community TikTok. It merges samples across all cultures to train a common model and applies the model through prompt engineering.

- CultureInstruct (Pham et al., 2025) is automatically constructed from public web sources using a specialized LLM to generate culturally relevant instructions, resulting in 430K samples spanning tasks from standard NLP to complex reasoning. The dataset explicitly incorporates 11 cultural topics, ensuring diversity and coverage across multiple cultural dimensions. To keep a similar scale with other baselines, we applies 10% of the whole collection for fine-tuning in this paper, around 43k samples.

- CultureData is a dataset created by us through merging all public manually curated cultural benchmarks, including NormBank (Ziems et al., 2023), CulturalAtlas (Fung et al., 2024), CLIcK (Kim et al., 2024), Cancle, BLEnD (Myung et al., 2024), SeaEval () and NormAd (Rao et al., 2024). For datasets such as NormBank, where each entry contains multiple structured components, we use LLMs to reformat them into plain text suitable for fine-tuning. The number of samples varies across cultures: among the 15 cultures considered in our experiments, Italy, Singapore, Poland, and Nigeria contain slightly fewer than 1,000 samples each, while others—most notably the UK—have substantially more.

## D.5 Implementation Details

We access proprietary LLMs via their official APIs, and follow the open-source code to produce cultural data or directly use the released datasets for other baselines. Our experiments cover 15 cultures selected from diverse regions with varying representation and popularity. Experiments are completed using NVIDIA A100 (80G). For fine-tuning based cultural alignment, we apply the general performance benchmark MMLU as a validation set. Early stopping is applied when the model's MMLU score decreases by more than 10% relative to the raw model performance. We would release the code and synthesized data for reproduction.

During the CARDSet creation process, we synthsize $N = 100$ questions for each topic at first. For representativeness optimization, we set 15 general people, 5 cultural experts and 3 cross cultural researchers.

## D.6 Discussion on Data Leakage

As shown in Line 475-477, we explicitly mention this potential data leakage and the possible inflated gains caused by it. In this case, we still keep these questionnaire-style benchmarks in our evaluation for two reasons below.

(1) **Comparing the impact of leakage by WVS**. Several baselines, e.g., CultureLLM, CulturePark, CultureSPA, are structurally integrated with WVS, and it's infeasible to remove WVS-related data without fundamentally altering these baselines. However, CAReDiO uses only WVS demographics, not its questions or responses. As discussed in Sec.4.3 (Line 482-485), the evaluation on WVS helpds show the limitation of baselines, which perform well only on these two benchmarks but fail on other OOD ones, indicating potential leakage and overfitting.

(2) **Comparing the generalization ability of different methods**. To mitigate possible leakage from WVS, we take two actions: (i) **We remove the WVS split in GlobalOpinionQA** and only keep the GAS subset for evaluation. (ii) **We intentionally introduce two additional benchmarks in different formats for comparison**. As shown in Table 3, our method achieves significantly larger gains on these non-WVS benchmarks than on WVS/GlobalOpinionQA, while baselines, especially the three rooted in WVS, only perform well on WVS (on non-WVS sets, even worse than Role-Play). This demonstrates CAReDiO's improvements are not driven by leakage and supports its robustness and generality.

# E  LLM USAGE DISCLOSURE

In accordance with ICLR 2026's author guidelines regarding the use of Large Language Models, we confirm that ChatGPT was used solely for minor polishment purposes, *e.g.*, correcting grammatical errors and refining the phrasing of certain sentences in the main text of this paper. At no point were LLMs involved in generating research ideas, designing experiments, conducting analyses, or drafting the substantive essential content. All research contributions, analyses, and conclusions presented herein are entirely the original work of the authors.

# F  SUPPLEMENTS FOR RESULTS

## F.1  CULTURAL ALIGNMENT PERFORMANCE

Cultural alignment performance with Llama-3.1-8B-Instruct as the backbone is listed in Tab. 9

Table 9: Evaluation results of cultural alignment on four different benchmarks

| Method | CB-Easy | CB-Hard | Prism | GlobalOpinionQA | WVS | Average |
|---|---|---|---|---|---|---|
| Llama-3.1-8B-Instruct | 68.76 | 36.84 | 2.051 | 54.50 | 61.10 | 52.45 |
| Role-Play | 68.83 | 38.58 | 3.575 | 54.10 | 63.93 | 59.39 |
| CultureLLM | 71.85 | 37.36 | 3.371 | 56.07 | 62.57 | 59.05 |
| CulturePark | 69.48 | 38.66 | 2.787 | 56.47 | 61.22 | 56.32 |
| CultureSPA | 69.10 | 38.22 | 2.814 | 54.09 | 58.49 | 55.24 |
| CultureBank | 67.97 | 12.38 | 3.403 | 57.14 | 57.90 | 52.69 |
| CultureInstruction | 46.23 | 7.75 | 3.274 | 56.36 | 51.69 | 45.50 |
| CultureData | 68.93 | 36.75 | 3.414 | 51.37 | 60.68 | 57.20 |
| CAReDiO | 71.38 | 40.03 | 4.205 | 55.97 | 63.89 | 63.07 |

Here, we present the alignment performance for each culture across the four datasets in Table 10, Table 11, Table 12, Table 13 and Table 14.

Table 10: Cultural alignment performance across various cultures on the GlobalOpinionQA dataset.

| Models | US | UK | Germany | Italy | China | Japan | Korea | India | Singapore | Indonesia | Russia | Poland | Mexico | Nigeria |
|---|---|---|---|---|---|---|---|---|---|---|---|---|---|---|
| gpt-5 | 55.43 | 52.45 | 51.58 | 46.33 | 40.46 | 46.32 | 45.25 | 47.66 | 33.87 | 40.88 | 43.48 | 48.80 | 48.66 | 46.55 |
| gpt-5 + Role-Play | 64.50 | 65.27 | 62.81 | 61.02 | 57.44 | 56.47 | 60.04 | 59.36 | 67.74 | 51.82 | 56.65 | 59.71 | 59.40 | 58.84 |
| Llama-3.1-8B-Instruct | 61.81 | 59.56 | 58.56 | 52.82 | 47.38 | 53.97 | 53.87 | 52.77 | 61.29 | 50.46 | 47.19 | 54.12 | 55.52 | 53.73 |
| Role-Play | 62.15 | 59.32 | 59.32 | 50.71 | 51.57 | 48.53 | 48.24 | 57.45 | 54.84 | 51.98 | 51.92 | 58.24 | 52.99 | 50.14 |
| CultureLLM | 58.12 | 53.85 | 56.60 | 50.71 | 50.52 | 58.53 | 55.46 | 59.15 | 54.84 | 52.43 | 56.27 | 58.24 | 54.48 | 65.75 |
| CulturePark | 61.37 | 60.61 | 61.72 | 50.71 | 55.56 | 51.62 | 54.40 | 67.23 | 53.23 | 52.43 | 53.58 | 58.24 | 54.93 | 54.97 |
| CultureSPA | 61.14 | 57.58 | 60.74 | 50.99 | 53.25 | 50.44 | 50.53 | 55.11 | 50.00 | 52.89 | 51.92 | 59.71 | 54.33 | 48.62 |
| CultureBank | 64.50 | 61.07 | 60.74 | 53.11 | 54.30 | 50.44 | 52.64 | 58.94 | 56.45 | 55.47 | 56.01 | 58.64 | 62.69 | 54.97 |
| CultureInstruction | 59.91 | 58.74 | 60.09 | 55.08 | 53.67 | 55.74 | 57.57 | 51.28 | 59.68 | 58.66 | 57.03 | 54.52 | 52.54 | 54.56 |
| CultureData | 59.13 | 54.90 | 52.89 | 49.15 | 46.96 | 44.85 | 50.18 | 55.11 | 46.77 | 59.73 | 46.29 | 53.72 | 55.82 | 43.65 |
| CAReDiO | 62.04 | 59.67 | 62.81 | 50.85 | 53.46 | 49.12 | 55.46 | 54.89 | 62.90 | 53.50 | 54.73 | 55.05 | 59.40 | 49.72 |
| Qwen2.5-7B-Instruct | 55.77 | 57.34 | 55.94 | 52.68 | 46.54 | 52.94 | 54.23 | 54.68 | 53.23 | 48.33 | 49.87 | 54.92 | 56.42 | 53.04 |
| Role-Play | 58.34 | 57.81 | 57.58 | 53.67 | 53.25 | 54.85 | 52.64 | 58.30 | 62.90 | 47.42 | 56.14 | 57.45 | 57.61 | 53.59 |
| CultureLLM | 59.91 | 60.02 | 56.38 | 53.67 | 54.30 | 51.32 | 55.81 | 64.47 | 59.68 | 50.91 | 56.91 | 57.45 | 57.01 | 61.74 |
| CulturePark | 60.58 | 59.32 | 58.67 | 53.67 | 53.70 | 53.70 | 58.72 | 62.90 | 48.02 | 56.39 | 57.45 | 58.51 | 53.87 |
| CultureSPA | 57.00 | 55.83 | 59.32 | 51.55 | 50.52 | 50.15 | 45.77 | 54.89 | 46.77 | 47.26 | 54.86 | 58.24 | 55.22 | 52.21 |
| CultureBank | 60.58 | 59.67 | 58.02 | 53.81 | 51.36 | 54.56 | 51.06 | 61.70 | 58.06 | 52.58 | 56.01 | 56.78 | 58.96 | 56.91 |
| CultureInstruction | 61.03 | 60.14 | 59.21 | 53.53 | 52.83 | 55.15 | 55.63 | 64.47 | 62.90 | 53.04 | 55.63 | 57.05 | 57.31 | 61.05 |
| CultureData | 60.47 | 59.32 | 59.43 | 55.79 | 53.04 | 56.76 | 55.28 | 57.02 | 62.90 | 51.37 | 59.59 | 59.04 | 57.76 | 56.35 |
| CAReDiO | 59.69 | 57.81 | 59.54 | 54.52 | 53.88 | 54.56 | 52.46 | 58.09 | 62.90 | 47.42 | 56.77 | 57.85 | 57.61 | 54.14 |
| Gemma-3-27B-IT | 57.33 | 59.09 | 59.54 | 56.07 | 42.14 | 55.74 | 56.69 | 48.94 | 40.32 | 43.31 | 47.70 | 55.59 | 53.13 | 50.00 |
| Role-Play | 62.37 | 58.74 | 58.67 | 51.41 | 49.48 | 54.71 | 55.63 | 51.06 | 59.68 | 48.78 | 55.24 | 56.12 | 53.43 | 52.49 |
| CultureLLM | 59.24 | 58.74 | 61.72 | 51.41 | 56.81 | 60.52 | 62.85 | 53.40 | 75.81 | 51.82 | 58.18 | 56.12 | 58.96 | 48.48 |
| CulturePark | 65.40 | 64.57 | 65.21 | 51.41 | 54.93 | 57.94 | 62.50 | 59.36 | 75.81 | 51.67 | 58.70 | 56.12 | 55.22 | 57.46 |
| CultureSPA | 63.61 | 58.86 | 62.38 | 57.49 | 51.15 | 53.82 | 58.10 | 50.21 | 62.90 | 52.43 | 56.65 | 55.45 | 58.81 | 50.41 |
| CultureBank | 61.93 | 59.44 | 58.34 | 54.80 | 52.20 | 51.76 | 55.28 | 53.40 | 69.35 | 50.30 | 55.75 | 52.66 | 53.88 | 53.31 |
| CultureInstruction | 62.37 | 66.20 | 63.14 | 53.25 | 49.69 | 55.74 | 57.04 | 59.79 | 70.97 | 54.56 | 59.21 | 58.51 | 57.31 | 61.33 |
| CultureData | 61.81 | 65.03 | 63.69 | 51.98 | 50.10 | 56.91 | 58.27 | 53.19 | 72.58 | 52.13 | 59.72 | 57.05 | 61.49 | 52.62 |
| CAReDiO | 63.83 | 62.24 | 60.41 | 53.67 | 53.46 | 56.30 | 58.10 | 54.89 | 70.97 | 51.67 | 59.21 | 56.74 | 58.81 | 56.35 |
| gpt-4.1 | 59.91 | 60.37 | 56.92 | 55.23 | 41.09 | 54.26 | 55.99 | 51.06 | 46.77 | 43.77 | 49.36 | 56.91 | 55.67 | 50.28 |
| Role-Play | 59.57 | 61.89 | 60.31 | 54.94 | 54.72 | 50.88 | 56.16 | 37.23 | 48.39 | 49.70 | 56.01 | 56.52 | 52.09 | 54.28 |
| CultureBank | 64.73 | 61.31 | 60.63 | 57.06 | 53.46 | 56.32 | 57.75 | 51.70 | 58.06 | 48.63 | 58.31 | 57.31 | 55.82 | 53.59 |
| CAReDiO | 65.29 | 64.69 | 60.41 | 57.06 | 55.97 | 54.71 | 51.41 | 51.70 | 58.06 | 48.63 | 58.31 | 57.31 | 55.80 | 53.59 |

## F.2  ABLATION STUDY

Table 11: Cultural alignment performance across various cultures on the WVS dataset.

| Models | US | UK | Germany | China | India | Russia | Mexico | Nigeria |
|---|---|---|---|---|---|---|---|---|
| gpt-5 | 70.61 | 77.61 | 73.12 | 59.41 | 54.38 | 60.46 | 53.28 | 50.17 |
| gpt-5 + Role-Play | 76.35 | 78.35 | 74.76 | 67.58 | 68.51 | 67.36 | 61.47 | 69.16 |
| Llama-3.1-8B-Instruct | 65.22 | 68.68 | 68.13 | 61.40 | 54.25 | 65.70 | 52.97 | 52.43 |
| Role-Play | 70.82 | 70.30 | 68.92 | 61.44 | 58.59 | 62.28 | 58.51 | 60.57 |
| CultureLLM | 70.23 | 69.13 | 70.85 | 60.25 | 57.59 | 58.82 | 59.02 | 54.66 |
| CulturePark | 66.68 | 69.47 | 64.71 | 61.66 | 58.73 | 56.49 | 57.61 | 54.43 |
| CultureSPA | 61.85 | 62.09 | 65.88 | 52.34 | 54.36 | 54.06 | 58.14 | 59.24 |
| CultureBank | 63.76 | 64.57 | 66.31 | 54.29 | 54.45 | 54.35 | 50.99 | 54.46 |
| CultureInstruction | 55.81 | 58.17 | 55.16 | 47.61 | 48.79 | 50.69 | 51.25 | 46.07 |
| CultureData | 66.32 | 67.40 | 63.27 | 57.20 | 57.92 | 59.21 | 55.91 | 58.26 |
| CAReDiO | 69.64 | 68.66 | 65.12 | 63.68 | 61.25 | 61.75 | 57.30 | 63.71 |
| Qwen2.5-7B-Instruct | 69.98 | 68.20 | 67.18 | 54.85 | 56.24 | 56.78 | 53.24 | 50.98 |
| Role-Play | 70.01 | 74.16 | 71.83 | 59.59 | 63.06 | 63.22 | 57.26 | 60.56 |
| CultureLLM | 71.10 | 71.22 | 69.29 | 66.85 | 63.39 | 62.26 | 57.75 | 62.09 |
| CulturePark | 63.05 | 60.95 | 61.20 | 55.88 | 53.37 | 55.89 | 56.59 | 55.72 |
| CultureSPA | 64.91 | 70.36 | 68.23 | 59.18 | 58.46 | 61.31 | 54.42 | 59.10 |
| CultureBank | 68.27 | 70.10 | 69.04 | 58.69 | 59.64 | 59.67 | 55.01 | 59.33 |
| CultureInstruction | 72.34 | 65.16 | 67.91 | 61.12 | 62.45 | 61.90 | 55.00 | 60.15 |
| CultureData | 72.86 | 71.89 | 69.62 | 60.98 | 62.47 | 65.19 | 55.51 | 59.02 |
| CAReDiO | 70.99 | 75.02 | 71.89 | 59.64 | 62.83 | 64.25 | 56.82 | 60.62 |
| Gemma-3-27B-IT | 70.61 | 74.30 | 72.59 | 64.19 | 58.98 | 64.23 | 58.32 | 54.98 |
| Role-Play | 75.19 | 73.44 | 72.53 | 61.86 | 66.81 | 59.52 | 61.56 | 66.84 |
| CultureLLM | 74.41 | 72.64 | 71.41 | 64.06 | 63.51 | 61.67 | 60.55 | 67.69 |
| CulturePark | 72.34 | 71.07 | 70.76 | 66.03 | 61.71 | 61.57 | 60.41 | 63.76 |
| CultureSPA | 74.45 | 77.17 | 71.47 | 66.44 | 65.98 | 59.88 | 61.25 | 65.48 |
| CultureBank | 72.06 | 70.60 | 72.65 | 59.79 | 71.09 | 57.76 | 64.06 | 68.84 |
| CultureInstruction | 65.33 | 64.89 | 63.52 | 58.78 | 66.23 | 54.50 | 53.75 | 64.35 |
| CultureData | 73.24 | 72.20 | 71.98 | 68.49 | 66.61 | 64.87 | 61.23 | 65.53 |
| CAReDiO | 75.67 | 73.44 | 72.16 | 64.44 | 65.63 | 64.18 | 61.88 | 66.25 |
| gpt-4.1 | 71.52 | 75.97 | 71.68 | 57.75 | 50.99 | 59.57 | 52.22 | 47.60 |
| Role-Play | 75.33 | 78.47 | 73.54 | 66.82 | 67.22 | 67.36 | 62.22 | 67.82 |
| CultureBank | 73.79 | 78.19 | 71.71 | 64.86 | 63.64 | 66.84 | 59.66 | 66.18 |
| CAReDiO | 72.99 | 80.54 | 73.16 | 66.93 | 67.22 | 67.36 | 62.22 | 66.86 |

Table 12: Cultural alignment performance across various cultures on the CulturalBench-Easy dataset.

| Models | US | UK | Germany | Italy | China | Japan | Korea | India | Singapore | Indonesia | Russia | Poland | Romania | Mexico | Nigeria |
|---|---|---|---|---|---|---|---|---|---|---|---|---|---|---|---|
| gpt-5 | 100.00 | 92.00 | 93.75 | 83.33 | 83.05 | 88.68 | 92.68 | 80.43 | 86.96 | 80.77 | 80.00 | 87.50 | 93.33 | 93.88 | 95.45 |
| gpt-5 + Role-Play | 95.00 | 96.00 | 96.88 | 88.89 | 84.75 | 90.57 | 90.24 | 80.43 | 86.96 | 80.77 | 80.00 | 87.50 | 100.00 | 89.80 | 95.45 |
| Llama-3.1-8B-Instruct | 70.00 | 72.00 | 68.75 | 69.44 | 62.71 | 77.36 | 51.22 | 76.09 | 56.52 | 69.23 | 60.00 | 87.50 | 60.00 | 67.35 | 86.36 |
| Role-Play | 70.00 | 72.00 | 71.88 | 75.00 | 52.54 | 79.25 | 48.78 | 76.09 | 56.52 | 65.38 | 56.67 | 83.33 | 73.33 | 65.31 | 86.36 |
| CultureLLM | 70.00 | 72.00 | 68.75 | 75.00 | 59.32 | 79.25 | 65.85 | 76.09 | 65.22 | 73.08 | 70.00 | 83.33 | 66.67 | 71.43 | 81.82 |
| CulturePark | 70.00 | 72.00 | 68.75 | 75.00 | 59.32 | 79.25 | 51.22 | 73.91 | 56.52 | 69.23 | 63.33 | 83.33 | 66.67 | 67.35 | 86.36 |
| CultureSPA | 70.00 | 72.00 | 68.75 | 75.00 | 54.24 | 79.25 | 48.78 | 73.91 | 60.87 | 65.38 | 66.67 | 83.33 | 66.67 | 65.31 | 86.36 |
| CultureBank | 75.00 | 76.00 | 62.50 | 72.22 | 55.93 | 75.47 | 46.34 | 73.91 | 56.52 | 69.23 | 46.67 | 83.33 | 66.67 | 63.27 | 81.82 |
| CultureInstruction | 50.00 | 36.00 | 53.12 | 52.78 | 35.59 | 60.38 | 46.34 | 45.65 | 43.48 | 50.00 | 36.67 | 37.50 | 33.33 | 48.98 | 63.64 |
| CultureData | 70.00 | 72.00 | 62.50 | 69.44 | 59.32 | 79.25 | 65.85 | 71.74 | 60.87 | 65.38 | 60.00 | 79.17 | 73.33 | 63.27 | 86.36 |
| CAReDiO | 75.00 | 76.00 | 78.12 | 72.22 | 54.24 | 77.36 | 65.22 | 56.52 | 65.38 | 60.00 | 79.17 | 80.00 | 63.27 | 81.82 | 86.36 |
| Qwen2.5-7B-Instruct | 70.00 | 84.00 | 75.00 | 61.11 | 74.58 | 77.36 | 53.66 | 82.61 | 78.26 | 57.69 | 70.00 | 83.33 | 73.33 | 75.51 | 63.64 |
| Role-Play | 80.00 | 84.00 | 71.88 | 72.22 | 74.58 | 75.47 | 60.98 | 76.09 | 65.22 | 57.69 | 76.67 | 91.67 | 66.67 | 73.47 | 59.09 |
| CultureLLM | 80.00 | 80.00 | 71.88 | 72.22 | 74.58 | 75.47 | 63.41 | 80.43 | 65.22 | 61.54 | 70.00 | 91.67 | 53.33 | 73.47 | 63.64 |
| CulturePark | 75.00 | 76.00 | 71.88 | 72.22 | 71.19 | 73.58 | 60.98 | 80.43 | 60.87 | 61.54 | 60.00 | 83.33 | 80.00 | 75.51 | 77.27 |
| CultureSPA | 80.00 | 76.00 | 68.75 | 72.22 | 64.41 | 75.47 | 58.54 | 69.57 | 60.87 | 61.54 | 80.00 | 83.33 | 80.00 | 69.39 | 63.64 |
| CultureBank | 80.00 | 80.00 | 71.88 | 63.89 | 69.49 | 77.36 | 60.98 | 82.61 | 65.22 | 61.54 | 63.33 | 79.17 | 86.67 | 69.39 | 72.73 |
| CultureInstruction | 85.00 | 80.00 | 71.88 | 72.22 | 71.19 | 77.36 | 65.85 | 76.09 | 65.85 | 69.23 | 63.33 | 87.50 | 73.33 | 73.47 | 77.27 |
| CultureData | 80.00 | 80.00 | 71.88 | 72.22 | 74.58 | 75.47 | 63.42 | 78.26 | 60.87 | 61.54 | 83.33 | 91.67 | 66.67 | 73.47 | 59.09 |
| CAReDiO | 80.00 | 76.00 | 75.00 | 75.00 | 76.27 | 81.13 | 60.98 | 76.09 | 65.22 | 61.54 | 76.67 | 79.17 | 73.33 | 77.55 | 68.18 |
| Gemma-3-27B-IT | 95.00 | 88.00 | 78.12 | 75.00 | 81.36 | 79.25 | 78.26 | 78.26 | 82.61 | 69.23 | 80.00 | 79.17 | 100.00 | 83.67 | 86.36 |
| Role-Play | 95.00 | 84.00 | 81.25 | 72.22 | 83.05 | 81.13 | 73.17 | 78.26 | 78.26 | 73.08 | 80.00 | 83.33 | 93.33 | 77.55 | 86.36 |
| CultureLLM | 90.00 | 80.00 | 81.25 | 72.22 | 77.97 | 79.25 | 78.05 | 78.26 | 82.61 | 69.23 | 80.00 | 83.33 | 86.67 | 81.63 | 86.36 |
| CulturePark | 95.00 | 84.00 | 78.12 | 72.22 | 81.36 | 83.02 | 73.17 | 78.26 | 86.96 | 73.08 | 80.00 | 83.33 | 93.33 | 79.59 | 86.36 |
| CultureSPA | 95.00 | 88.00 | 87.50 | 69.44 | 79.66 | 79.25 | 70.73 | 73.91 | 73.91 | 76.92 | 86.67 | 83.33 | 86.67 | 83.67 | 86.36 |
| CultureBank | 95.00 | 88.00 | 78.12 | 77.78 | 76.27 | 77.36 | 78.05 | 82.61 | 73.91 | 69.23 | 80.00 | 91.67 | 93.33 | 79.59 | 86.36 |
| CultureInstruction | 95.00 | 84.00 | 65.62 | 69.44 | 55.93 | 73.58 | 73.17 | 80.43 | 69.57 | 80.77 | 76.67 | 75.00 | 86.67 | 79.59 | 77.27 |
| CultureData | 95.00 | 88.00 | 84.38 | 72.22 | 79.66 | 79.25 | 75.61 | 80.43 | 82.61 | 73.08 | 76.67 | 83.33 | 93.33 | 77.55 | 86.36 |
| CAReDiO | 95.00 | 88.00 | 81.25 | 72.22 | 76.27 | 79.25 | 70.73 | 78.26 | 82.61 | 76.92 | 83.33 | 91.67 | 93.33 | 87.76 | 81.82 |
| gpt-4.1 | 100.00 | 96.00 | 90.62 | 88.89 | 84.75 | 88.68 | 92.68 | 82.61 | 86.96 | 84.62 | 83.33 | 91.67 | 93.33 | 87.76 | 95.45 |
| Role-Play | 100.00 | 100.00 | 90.62 | 86.11 | 83.05 | 90.57 | 90.24 | 84.78 | 78.26 | 80.77 | 86.67 | 91.67 | 100.00 | 85.71 | 90.91 |
| CultureBank | 100.00 | 92.00 | 90.62 | 91.67 | 84.75 | 90.57 | 90.24 | 86.96 | 82.61 | 88.46 | 86.67 | 91.67 | 100.00 | 85.71 | 100.00 |
| CAReDiO | 100.00 | 100.00 | 90.62 | 91.67 | 86.44 | 90.57 | 90.24 | 86.96 | 82.61 | 80.77 | 86.67 | 91.67 | 100.00 | 85.71 | 90.91 |

We perform an ablation study to evaluate the independent contributions of the representativeness and distinctiveness objectives in our framework. Specifically, we compare two variants against both the backbone LLM and the full CAReDiO:

Table 13: Cultural alignment performance across various cultures on the CulturalBench-Hard dataset.

| Models | US | UK | Germany | Italy | China | Japan | Korea | India | Singapore | Indonesia | Russia | Poland | Romania | Mexico | Nigeria |
|---|---|---|---|---|---|---|---|---|---|---|---|---|---|---|---|
| gpt-5 | 55.00 | 68.00 | 78.12 | 55.56 | 59.32 | 71.70 | 60.98 | 47.83 | 60.87 | 50.00 | 56.67 | 50.00 | 46.67 | 55.10 | 77.27 |
| gpt-5 + Role-Play | 70.00 | 72.00 | 78.12 | 58.33 | 62.71 | 79.25 | 53.66 | 47.83 | 56.52 | 50.00 | 60.00 | 50.00 | 60.00 | 46.94 | 54.55 |
| Llama-3.1-8B-Instruct | 40.00 | 48.00 | 43.75 | 36.11 | 27.12 | 47.17 | 26.83 | 28.26 | 17.39 | 57.69 | 30.00 | 29.17 | 33.33 | 46.94 | 40.91 |
| Role-Play | 50.00 | 44.00 | 50.00 | 44.44 | 25.42 | 52.83 | 19.51 | 28.26 | 13.04 | 46.15 | 36.67 | 37.50 | 46.67 | 38.78 | 45.45 |
| CultureLLM | 50.00 | 40.00 | 43.75 | 44.44 | 30.51 | 50.94 | 31.71 | 34.78 | 13.04 | 38.46 | 30.00 | 37.50 | 46.67 | 36.73 | 31.82 |
| CulturePark | 50.00 | 48.00 | 43.75 | 44.44 | 30.51 | 49.06 | 26.83 | 39.13 | 17.39 | 38.46 | 20.00 | 40.00 |  | 44.90 | 50.00 |
| CultureSPA | 45.00 | 44.00 | 43.75 | 36.11 | 32.20 | 49.06 | 26.83 | 39.13 | 17.39 | 46.15 | 30.00 | 37.50 | 33.30 | 42.86 | 50.00 |
| CultureBank | 30.00 | 24.00 | 6.25 | 13.89 | 6.78 | 20.75 | 9.76 | 19.57 | 4.35 | 15.38 | 3.33 | 8.33 | 0.00 | 14.29 | 9.09 |
| CultureInstruction | 5.00 | 12.00 | 12.50 | 2.78 | 6.78 | 15.09 | 12.20 | 2.17 | 4.35 | 11.54 | 6.67 | 8.33 | 6.67 | 10.20 | 0.00 |
| CultureData | 45.00 | 52.00 | 37.50 | 44.44 | 32.20 | 54.72 | 26.83 | 30.43 | 17.39 | 26.92 | 26.67 | 33.33 | 46.67 | 40.82 | 36.36 |
| CAReDiO | 45.00 | 48.00 | 46.88 | 33.33 | 32.20 | 52.83 | 36.96 | 13.04 | 46.15 | 26.67 | 33.33 | 40.00 | 55.10 | 45.45 | 45.45 |
| Qwen2.5-7B-Instruct | 55.00 | 52.00 | 34.38 | 41.67 | 38.98 | 56.60 | 31.71 | 36.96 | 21.74 | 34.62 | 43.33 | 29.17 | 26.67 | 30.61 | 50.00 |
| Role-Play | 55.00 | 52.00 | 34.38 | 38.89 | 37.29 | 56.60 | 29.27 | 30.43 | 21.74 | 34.62 | 40.00 | 29.17 | 26.67 | 28.57 | 36.36 |
| CultureLLM | 55.00 | 40.00 | 37.50 | 38.89 | 33.90 | 56.60 | 19.57 | 30.43 | 21.74 | 38.46 | 36.67 | 29.17 | 20.00 | 22.45 | 36.36 |
| CulturePark | 45.00 | 48.00 | 40.62 | 38.89 | 25.42 | 43.40 | 26.83 | 34.78 | 21.74 | 34.62 | 23.33 | 29.17 | 33.33 | 34.69 | 36.36 |
| CultureSPA | 35.00 | 44.00 | 43.75 | 36.11 | 32.20 | 58.49 | 19.51 | 32.61 | 21.74 | 30.77 | 33.33 | 33.30 |  | 40.82 | 45.45 |
| CultureBank | 40.00 | 48.00 | 21.88 | 27.78 | 13.56 | 33.96 | 21.95 | 34.78 | 13.04 | 15.38 | 16.67 | 20.83 | 33.33 | 32.65 | 36.36 |
| CultureInstruction | 40.00 | 44.00 | 12.50 | 30.56 | 18.64 | 41.51 | 29.27 | 30.43 | 21.74 | 30.77 | 16.67 | 20.83 | 6.67 | 40.82 | 45.45 |
| CultureData | 45.00 | 56.00 | 40.62 | 50.00 | 32.20 | 60.38 | 29.27 | 39.13 | 21.74 | 42.31 | 36.67 | 33.33 | 26.67 | 42.86 | 45.45 |
| CAReDiO | 50.00 | 52.00 | 43.75 | 44.44 | 30.51 | 54.72 | 26.83 | 39.13 | 21.74 | 38.46 | 43.33 | 37.50 | 33.33 | 32.65 | 54.55 |
| Gemma-3-27B-IT | 60.00 | 52.00 | 43.75 | 36.11 | 44.07 | 60.38 | 39.13 | 30.43 |  | 50.00 | 53.33 | 29.17 | 53.33 | 48.98 | 59.09 |
| Role-Play | 75.00 | 56.00 | 62.50 | 36.11 | 45.76 | 62.26 | 41.46 | 36.96 | 30.43 | 61.54 | 50.00 | 29.17 | 46.67 | 44.90 | 45.45 |
| CultureLLM | 75.00 | 56.00 | 59.38 | 36.11 | 45.76 | 50.94 | 36.59 | 50.00 | 30.43 | 38.46 | 43.33 | 29.17 | 33.33 | 51.02 | 59.09 |
| CulturePark | 70.00 | 56.00 | 56.25 | 36.11 | 40.68 | 58.49 | 41.46 | 47.83 | 26.09 | 42.31 | 43.33 | 29.17 | 26.67 | 57.14 | 63.64 |
| CultureSPA | 80.00 | 52.00 | 62.50 | 36.11 | 37.29 | 66.04 | 41.46 | 45.65 | 34.78 | 53.85 | 46.67 | 37.50 | 33.33 | 42.86 | 50.00 |
| CultureBank | 60.00 | 48.00 | 34.38 | 25.00 | 42.37 | 49.06 | 39.02 | 45.65 | 26.09 | 42.31 | 40.00 | 41.67 | 20.00 | 51.02 | 63.64 |
| CultureInstruction | 35.00 | 36.00 | 9.38 | 13.89 | 10.17 | 16.98 | 14.63 | 19.57 | 17.39 | 19.23 | 13.33 | 25.00 | 0.00 | 28.57 | 13.64 |
| CultureData | 60.00 | 56.00 | 34.38 | 38.89 | 30.51 | 43.40 | 53.66 | 30.43 | 30.43 | 38.46 | 43.33 | 45.83 | 46.67 | 53.06 | 59.09 |
| CAReDiO | 75.00 | 56.00 | 65.62 | 38.89 | 45.76 | 60.38 | 39.02 | 39.13 | 26.09 | 61.54 | 46.67 | 37.50 | 46.67 | 44.90 | 50.00 |
| gpt-4.1 | 60.00 | 76.00 | 71.88 | 63.89 | 54.24 | 79.25 | 53.66 | 47.83 | 47.83 | 61.54 | 50.00 | 58.33 | 33.33 | 61.22 | 72.73 |
| Role-Play | 70.00 | 80.00 | 81.25 | 61.11 | 59.32 | 75.47 | 58.54 | 47.83 | 60.87 | 50.00 | 46.67 | 62.50 | 53.33 | 63.27 | 81.82 |
| CultureBank | 50.00 | 72.00 | 78.12 | 52.78 | 62.71 | 71.70 | 53.66 | 47.83 | 56.52 | 50.00 | 50.00 | 54.17 | 53.33 | 65.31 | 81.82 |
| CAReDiO | 65.00 | 88.00 | 78.12 | 61.11 | 59.32 | 75.47 | 63.41 | 47.83 | 60.87 | 50.00 | 46.67 | 62.50 | 53.33 | 63.27 | 78.18 |

Table 14: Cultural alignment performance across various cultures on the Prism dataset.

| Models | US | UK | Germany | Italy | China | Japan | Korea | India | Indonesia | Russia | Poland | Romania | Mexico | Nigeria |
|---|---|---|---|---|---|---|---|---|---|---|---|---|---|---|
| gpt-5 | 2.867 | 2.407 | 2.351 | 2.024 | 1.917 | 2.444 | 2.037 | 2.213 | 2.259 | 1.981 | 2.071 | 2.019 | 2.013 | 2.013 |
| gpt-5 + Role-Play | 4.553 | 4.627 | 4.868 | 4.553 | 4.306 | 4.508 | 4.691 | 4.267 | 5.000 | 4.278 | 4.496 | 4.333 | 4.393 | 4.393 |
| Llama-3.1-8B-Instruct | 2.520 | 2.173 | 2.033 | 2.033 | 1.861 | 2.286 | 1.963 | 2.093 | 2.000 | 1.981 | 1.972 | 1.926 | 1.960 | 1.913 |
| Role-Play | 3.853 | 3.607 | 3.728 | 3.780 | 3.667 | 3.381 | 3.432 | 3.627 | 3.741 | 3.444 | 3.574 | 2.963 | 3.680 | 3.573 |
| CultureLLM | 3.500 | 3.303 | 3.298 | 3.780 | 3.444 | 3.397 | 3.420 | 3.133 | 3.407 | 3.389 | 3.574 | 2.926 | 3.240 | 3.387 |
| CulturePark | 2.827 | 3.020 | 2.570 | 3.780 | 2.611 | 2.349 | 2.642 | 2.907 | 2.074 | 2.519 | 3.574 | 2.222 | 3.027 | 2.893 |
| CultureSPA | 3.180 | 2.647 | 2.746 | 2.797 | 2.556 | 2.683 | 2.605 | 2.733 | 3.556 | 2.519 | 2.369 | 3.278 | 2.813 | 2.920 |
| CultureBank | 3.573 | 3.473 | 3.579 | 3.439 | 3.583 | 3.222 | 3.222 | 3.533 | 3.593 | 3.185 | 3.199 | 3.259 | 3.420 | 3.360 |
| CultureInstruction | 3.747 | 3.413 | 3.175 | 3.325 | 3.333 | 2.984 | 3.296 | 3.253 | 3.741 | 2.963 | 3.213 | 2.833 | 3.213 | 3.347 |
| CultureData | 3.360 | 3.427 | 3.544 | 3.618 | 3.278 | 3.381 | 3.235 | 3.240 | 3.815 | 3.519 | 3.351 | 2.963 | 3.520 | 3.547 |
| CAReDiO | 4.487 | 4.507 | 4.518 | 4.244 | 4.028 | 3.905 | 4.136 | 4.067 | 4.370 | 4.407 | 4.170 | 3.333 | 4.373 | 4.327 |
| Qwen2.5-7B-Instruct | 2.447 | 2.273 | 2.272 | 1.967 | 1.917 | 2.365 | 2.062 | 2.187 | 2.074 | 2.000 | 1.986 | 1.907 | 2.027 | 1.960 |
| Role-Play | 3.373 | 3.247 | 3.667 | 3.488 | 3.611 | 3.095 | 3.321 | 3.107 | 3.630 | 3.444 | 3.369 | 2.778 | 3.560 | 3.407 |
| CultureLLM | 3.180 | 3.013 | 3.482 | 3.488 | 3.278 | 2.873 | 3.000 | 2.907 | 2.852 | 2.833 | 3.369 | 2.796 | 3.173 | 3.453 |
| CulturePark | 3.193 | 3.220 | 3.149 | 3.488 | 3.111 | 2.810 | 2.877 | 2.987 | 3.556 | 2.981 | 3.369 | 2.722 | 2.927 | 3.113 |
| CultureSPA | 2.913 | 3.333 | 3.057 | 3.417 | 2.810 | 3.037 | 2.880 | 3.593 | 3.111 | 3.106 | 3.106 | 3.020 | 3.293 | 2.833 |
| CultureBank | 3.220 | 3.439 | 3.081 | 3.167 | 3.349 | 3.160 | 3.000 | 3.481 | 3.074 | 3.149 | 2.796 | 3.207 | 3.267 | 3.313 |
| CultureInstruction | 3.313 | 3.465 | 3.480 | 3.556 | 3.317 | 3.173 | 3.320 | 3.556 | 3.278 | 3.085 | 3.000 | 3.427 | 3.507 | 3.360 |
| CultureData | 3.233 | 3.313 | 3.746 | 3.545 | 3.194 | 3.063 | 3.130 | 3.373 | 3.926 | 3.352 | 3.369 | 2.778 | 3.407 | 3.533 |
| CAReDiO | 3.967 | 3.860 | 4.395 | 3.724 | 3.861 | 3.794 | 3.802 | 3.747 | 4.000 | 3.889 | 3.766 | 3.315 | 4.160 | 3.907 |
| Gemma-3-27B-IT | 2.800 | 2.300 | 2.377 | 2.024 | 1.972 | 2.524 | 2.111 | 2.093 | 2.185 | 2.000 | 2.078 | 1.963 | 2.027 | 1.987 |
| Role-Play | 4.607 | 4.727 | 4.789 | 4.593 | 4.278 | 4.413 | 4.642 | 4.680 | 4.778 | 4.611 | 4.496 | 4.278 | 4.580 | 4.520 |
| CultureLLM | 4.367 | 4.560 | 4.632 | 4.593 | 4.222 | 4.238 | 4.395 | 4.387 | 4.889 | 4.370 | 4.496 | 4.111 | 4.387 | 4.520 |
| CulturePark | 4.487 | 4.640 | 4.781 | 4.593 | 4.389 | 4.302 | 4.556 | 4.253 | 4.741 | 4.389 | 4.496 | 4.148 | 4.513 | 4.347 |
| CultureSPA | 4.500 | 4.600 | 4.684 | 4.561 | 4.167 | 4.159 | 4.407 | 4.280 | 4.815 | 4.407 | 4.390 | 4.370 | 4.153 | 4.540 |
| CultureBank | 4.520 | 4.513 | 4.447 | 4.268 | 4.250 | 4.206 | 4.469 | 4.320 | 4.704 | 4.056 | 4.057 | 4.000 | 4.373 | 4.333 |
| CultureInstruction | 3.820 | 3.547 | 3.614 | 3.415 | 3.639 | 3.381 | 3.580 | 3.440 | 3.667 | 3.259 | 3.383 | 3.259 | 3.400 | 3.940 |
| CultureData | 4.113 | 3.177 | 3.772 | 4.390 | 3.944 | 4.190 | 3.926 | 3.560 | 4.296 | 4.222 | 4.305 | 4.278 | 3.800 | 4.480 |
| CAReDiO | 4.720 | 4.753 | 4.833 | 4.520 | 4.639 | 4.508 | 4.568 | 4.720 | 4.926 | 4.426 | 4.610 | 4.259 | 4.687 | 4.607 |
| gpt-4.1 | 2.680 | 2.213 | 2.211 | 2.049 | 1.944 | 2.397 | 2.074 | 2.187 | 2.111 | 2.019 | 2.028 | 1.963 | 1.980 | 1.980 |
| Role-Play | 4.560 | 4.360 | 4.605 | 4.285 | 4.222 | 4.254 | 4.062 | 4.067 | 4.630 | 4.093 | 4.170 | 4.093 | 4.187 | 4.187 |
| CultureBank | 4.507 | 4.400 | 4.579 | 4.073 | 4.194 | 4.238 | 4.210 | 4.013 | 4.593 | 4.037 | 4.184 | 3.722 | 4.193 | 4.220 |
| CAReDiO | 4.627 | 4.493 | 4.430 | 4.285 | 4.639 | 4.095 | 4.444 | 4.067 | 4.630 | 4.093 | 4.170 | 4.093 | 4.187 | 4.447 |

- Only Rep: this variant applies Eq.(3) to calculate the scores and select the best samples following Algorithm 1, optimizing representativeness only.
- Only Dist: in this version, Eq.(4) is employed for scoring and sample selection, focusing only on distinctiveness.

The comparison results are shown in Table 15, and we summarize several key findings.

(1) *Each of representativeness and distinctiveness objectives yields certain improvements compared to the simple Role-playing baseline.* Specifically, "Only Rep" improves 15.3% averaged on benchmarks, while only Dist improves 13.5%. (2) *The combination (full CAReDiO) achieves the best performance,*

Table 15: Results of the ablation study on each independent optimization objective.

| Method | CB-Easy | CB-Hard | Prism | GlobalOpinion | Average | Improvement |
|---|---|---|---|---|---|---|
| Qwen2.5-7B-Instruct | 72.01 | 38.90 | 2.103 | 53.28 | 51.56 | - |
| Role-Play | 72.38 | 36.73 | 3.364 | 55.83 | 58.05 | +12.59% |
| Only Rep | 74.53 | 38.70 | 3.478 | 54.94 | 59.43 | +15.26% |
| Only Dist | 72.18 | 38.14 | 3.436 | 54.95 | 58.50 | +13.45% |
| CAReDiO | 73.48 | 40.20 | 3.871 | 56.23 | 61.83 | +19.91% |

e.g., +20% compared to the backbone LLM. (3) *Optimizing representativeness alone outperforms distinctiveness alone*, indicating that current LLMs contain biased cultural knowledge while Eq.(3) more effectively mitigates them. Adding Dist on top of Rep brings additional gains, suggesting that Eq.(4) helps identify clearer cultural boundaries and resolve ambiguity.

**More Intuitive Discussion about Gains**    To help readers understand, we provide a more intuitive explanation for why CAReDiO, especially the two objectives, can outperform other methods on cultural alignment.

Following the line of synthesizing cultural data with LLMs, we acknowledge two key assumptions (which often hold in practice): (i) The backbone LLMs, e.g., GPT-4.1, have encoded sufficient knowledge about specific cultures and can be elicited in a certain way. (ii) The cultural knowledge distribution learned by the LLMs is imperfect and biased, which can be more serious in weaker models. Based on these assumptions, we observed that most prior methods directly apply simple role-playing, and usually produce biased or ambiguous data.

This improvement by CAReDiO is obtained by explicitly shaping the data distribution along two theoretically grounded criteria derived from cultural theory:

* *Representativeness*: CAReDiO utilizes multi-LLM ensemble to maximize the shared understandings of the target culture while minimizing irrelevant noise (corner cases) of generated samples (Eq.(2)-(3)), which helps mitigate the bias of single LLM role-playing. Thus, optimizing only representativeness yields improvements. In Table 3, with (weaker) Qwen2.5-7B-Instruct as backbone, on difficult tasks (CB-Hard), Role-Play performs much worse than CAReDiO (40.20 > 36.73).

* *Distinctiveness*: CAReDiO captures characteristics unique to the target culture rather than generic patterns shared across related cultures (Eq.(4)). This further distinguishes ambiguous data samples located near cultural boundaries. This is verified in Appendix F.3. For closely related cultures, e.g., Japan/Korea/China, CAReDiO achieves larger performance gaps between target and non-target cultures, $\Delta=5.3 > 2.4$, while ensuring better performance on each target one.

Therefore, through CAReDiOis an in-context optimization framework, its concrete designs are grounded in information-theoretic derivation to produce higher-quality cultural data than baselines, explaining its superior empirical performance.

### F.3    PERFORMANCE ON DISTINCTIVENESS AMONG RELATED CULTURES

We incorporate the distinctiveness objective in CAReDiOto help identify clearer cultural boundaries and resolve ambiguity. To validate its effectiveness, we conduct a focused analysis on culturally proximate regions, i.e., China, Japan and Korea, and evaluate whether CAReDiOimproves fine-grained distinctiveness among them. Analysis is conducted from two perspectives below.

**Distinctiveness of synthesized training data**    We embed CAReDiO-synthesized data for China, Japan, Korea, and those from the CulturePark benchmark, using OpenAI text-embedding-3-small API, then conduct TSNE dimensionality reduction. The visualization is shown in Fig. 11. We observe that **CAReDiO produces much more clearly separated clusters, while CulturePark exhibits substantial cross-cultural overlap**. To quantify this, we compute the average inter-cluster centroid distance: CAReDiO achieves 0.47 compared to CulturePark's 0.29. This substantial margin indicates that CAReDiO's synthesized data provides significantly finer distinctiveness than the baseline.

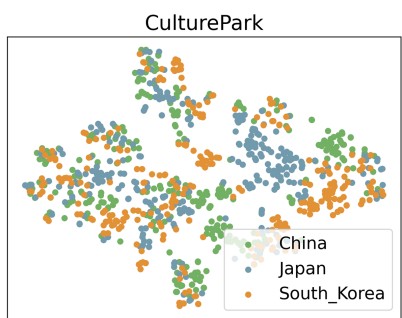 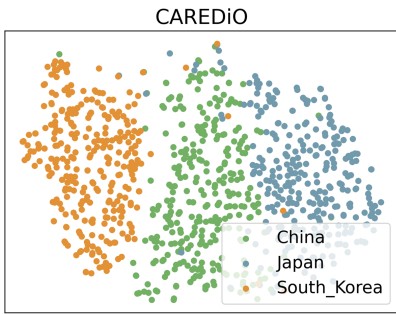

Figure 11: TSNE visualization of cultural alignment data for related cultures China/Japan/Korea, from CulturePark and CAReDiO respectively.

Table 16: Confusion matrices of cross-cultural evaluation on GlobalOpinionQA and Prism, for CulturePark and our CAReDiO framework.



(a) CulturePark on GlobalOpinionQA

|  | China | Japan | Korea | Δ ↑ | Con. Acc ↓ |
|---|---|---|---|---|---|
| China | 52.41 | 53.35 | 51.59 | -0.06 | 38.34 |
| Japan | 42.74 | 53.38 | 52.18 | 5.92 | 30.86 |
| Korea | 49.47 | 50.47 | 51.41 | 1.44 | 35.13 |

(b) CAReDiO on GlobalOpinionQA

|  | China | Japan | Korea | Δ ↑ | Con. Acc ↓ |
|---|---|---|---|---|---|
| China | 53.88 | 50.84 | 49.47 | 3.73 | 34.69 |
| Japan | 45.25 | 54.56 | 53.70 | 5.08 | 33.73 |
| Korea | 45.23 | 45.54 | 52.46 | 7.07 | 27.29 |

(c) CulturePark on Prism

|  | China | Japan | Korea | Δ ↑ |
|---|---|---|---|---|
| China | 1.0 | 0.7283 | 0.7214 | 0.2752 |
| Japan | 0.7283 | 1.0 | 0.7297 | 0.2710 |
| Korea | 0.7214 | 0.7298 | 1.0 | 0.2744 |

(d) CAReDiO on Prism

|  | China | Japan | Korea | Δ ↑ |
|---|---|---|---|---|
| China | 1.0 | 0.7122 | 0.6948 | 0.2965 |
| Japan | 0.7122 | 1.0 | 0.6756 | 0.3061 |
| Korea | 0.6948 | 0.6756 | 1.0 | 0.3148 |



**Cross-Cultural Confusion Matrices** We further evaluate models aligned to each target culture using either CAReDiO or CulturePark dataset, then measure how each model performs on non-target cultures to derive confusion matrices. We assessed two cultural benchmarks: GlobalOpinionQA and Prism.

Table 16(a) and (b) illustrate the assessment results on GlobalOpinionQA. The first three columns report the accuracy on each culture; $\Delta$ is the average performance gap between the target culture and non-target cultures. As for Con.Acc, we extract the subset from GlobalOpinion where the non-target culture shows different answers and report the model's alignment with the answers from the non-target culture, thus the lower the better.

We can see: (i) *Models aligned via CAReDiOperform better on the target culture while worse on non-target cultures*, achieving improved cultural specificity than CulturePark (larger performance gaps $\Delta$, 5.3 ¿ 2.4). ii) On the conflicting subset (different answers are expected from distinct cultures), CAReDiO models exhibit lower alignment on non-target cultures (Acc. on Conflicting, 31.9 < 34.8).

Table 16(c) and (d) present the evaluation results on the Prism dataset. Using 200 open-ended questions sampled from the Prism dataset, we measured pairwise similarity between responses produced by China-, Japan-, and Korea-aligned models. CAReDiOmodels display lower cross-cultural similarity (higher $\Delta$) than CulturePark (0.31 > 0.27), confirming better separation across related cultures.

Across all analyses, cluster distinguishability, centroid distances, confusion matrices, and cross-cultural response similarity, **CAReDiO consistently demonstrates stronger ability to make fine-grained distinctions between closely related cultures.**

F.4  ANALYSIS OF RESULTS VARIANCE

Table 17: The average and standard deviation of improvement over the backbone LLM across cultures.

|  | Average | Std | % of Improved Culture | % of Decreased Culture |
|---|---|---|---|---|
| CB-Easy | 1.47 | 6.88 | 73.30% | 26.70% |
| CB-Hard | 1.30 | 4.85 | 73.30% | 26.70% |
| Prism | 35.35 | 4.61 | 100% | 0 |
| GlobalOpinionQA | 2.95 | 3.10 | 100% | 0 |
| WVS | 5.00 | 2.47 | 100% | 0 |

Detailed per-culture evaluation results across all four benchmarks have been included in Table 12, 13, 14, 10 and Table 11. To address the concern that the reliability of major conclusions delivered by the average results in Table 3 could be impacted by the high variance raised from particular cultures or question types, we analyze the mean and std of CAReDiO's performance across all benchmarks and cultures.

**Average and standard deviation of gains across cultures**    As shown in Table 17, *on three benchmarks (Prism, GlobalOpinionQA and WVS), CAReDiO yields small/acceptable variance (std) compared to the average of the improvement over the backbone LLM*. Besides, we also observe *CAReDiO achieves improvements on most cultures*, e.g., 100% improvement ratio on Prism, GlobalOpinionQA and WVS, 73.3% on CultureBench).

Table 18: The improvement ratio of CAReDiO across each question type and cultures.

| Question Type | US | UK | Germany | Italy | China | Japan | South Korea | India |
|---|---|---|---|---|---|---|---|---|
| Multi-Choice Knowledge | 2.60% | -4.76% | 13.63% | 14.69% | -9.73% | 0.78% | -0.87% | -1.01% |
| Value Questionnaire | 4.24% | 5.41% | 6.73% | 3.49% | 12.25% | 3.06% | -3.26% | 8.98% |
| Open-ended Question | 62.12% | 69.82% | 93.44% | 89.32% | 101.41% | 60.42% | 84.38% | 71.33% |

| Question Type | Singapore | Indonesia | Russia | Poland | Romania | Mexico | Nigeria |
|---|---|---|---|---|---|---|---|
| Multi-Choice Knowledge | -8.33% | 8.88% | 4.76% | 11.78% | 12.49% | 4.68% | 8.12% |
| Value Questionnaire | 18.17% | -1.88% | 13.49% | 5.34% | 6.72% | 10.51% | 2.07% |
| Open-ended Question | - | 92.86% | 94.45% | 89.63% | 73.83% | 105.23% | 99.34% |

**Improvement ratio across each question type**    We classify all benchmarks into three categories of question type: i) Multi-choice question about culture knowledge (CB-Easy, CB-Hard), ii) Value Questionnaire (GlobalOpinionQA and WVS) and iii) Open-ended question (Prism). As shown in Table 18, CAReDiO obtains consistent improvements on most cultures and question types, with two general conclusions:

(1) *CAReDiO performs better on moderately low-resource cultures*, e.g., Germany and Russia. This is because our representativeness objective (Eq.(3)) helps elicit consensus and mitigates LLMs' inherent bias towards minority cultures.

(2) *CAReDiO achieves larger improvements on open-ended questions*. This is because our synthesized data takes the form of $(x, y)$ QA pairs. We adopt this format due to that most baselines are rooted in multiple-choice questions (e.g., CulturePark and CultureSPA) and tend to overfit to that format, and thus fail to generalize to more realistic open-ended QA tasks. However, CAReDiO still shows improvements across diverse question types, indicating that this form of data offers better generalization.

These findings confirm that our major conclusion—that CAReDiOconsistently and significantly enhances cultural alignment—is robust and not driven by high-variance artifacts in specific cultures or question types.

F.5    NUMBER OF INDIVIDUALS IN REPRESENTATIVENESS OPTIMIZATION

Drawing on the Cultural Consensus Theory, we incorporate multiple LLM-simulated individuals to perform consensus elicitation for representativeness optimization. We conduct experiments to

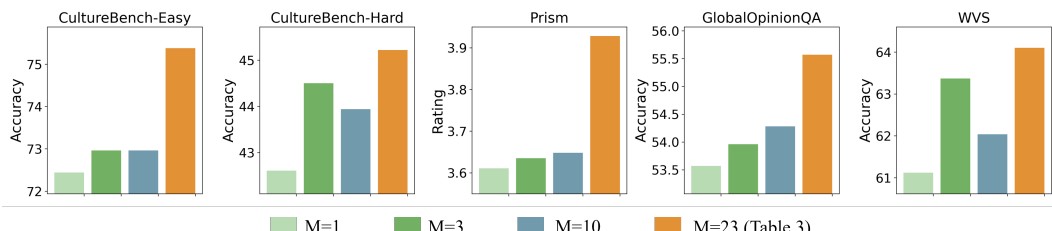

Figure 12: Performance comparison of CAReDiO with different numbers of simulated individuals (i.e., $M$ in the figure) for representativeness optimization.

analyze the impact of the number of individuals involved in this process, i.e., the hyper-parameter $M$ in Eq.(3). Again, we use Qwen2.5-7B-Instruct for both data synthesis and for being the aligned model, and report the results averaged across multiple countries (Germany, Italy, China, Japan, Russia and Nigeria).

As shown in Fig. 12, *introducing multiple LLM-simulated individuals indeed significantly enhances the performance*, demonstrating the effectiveness of consensus elicitation for optimizing representativeness and mitigating the bias embedded in the original LLM. *As a larger number of individuals become available, the results are further improved*. Whereas, when the computation resource is limited, a small number of roles can also produce results much better than the single version.

Table 19: Statistics of cultural datasets. Cult. Points: Culture points representing distinctive cultural aspects extracted from the dataset by GPT-4.1; Sim and SB are cosine similarity and Self-BLEU within each dataset; Cult. Sim is cosine similarity between subsets of different cultures.

| Datasets | Source | #sample | Avg.L ↑ | #Cult. Points ↑ | Sim ↓ | SB ↓ | Cult. Sim ↓ |
|---|---|---|---|---|---|---|---|
| CulturePark | WVS augmentation | 1,000 each | 68.6 | 494.6 | 0.235 | 0.406 | 0.223 |
| CultureData | manual annotation | 1,000 each | 16.8 | 1521.0 | **0.199** | 0.330 | **0.127** |
| CARDSet (embed sim) | LLM-synthetic | 1,000 each | **200.4** | **2027.0** | 0.251 | 0.324 | 0.202 |
| CARDSet (llm-judge) | LLM-synthetic | 1,000 each | 192.0 | 2021.0 | 0.257 | **0.321** | 0.205 |

Table 20: Alignment performance of CAReDiO using different classifiers for the distinctiveness objective. The best and second-best results are highlighted in bold and underlined.

| Method | CB-Easy | CB-Hard | Prism | GlobalOpinion | WVS | Average |
|---|---|---|---|---|---|---|
| Qwen2.5-7B-Instruct | 74.85 | 42.38 | 2.154 | 53.65 | 61.70 | 53.49 |
| Role-Play | 76.77 | 41.21 | 3.471 | 56.79 | 66.01 | 61.05 |
| CAReDiO (embed sim) | **76.92** | 42.04 | **4.002** | 57.55 | **66.43** | **64.13** |
| CAReDiO (llm-judge) | 76.34 | **42.59** | 3.975 | **57.75** | 66.25 | 64.05 |

## F.6 SENSITIVITY ANALYSIS OF DISTINCTIVENESS CLASSIFIER

Since the true culture distributions are unattainable, we introduce a cultural classifier to approximate the distinctiveness objective following Proposition 2. This classifier can be implemented in multiple ways, provided that it offers reliable signals for distinguishing between cultures. In Appendix C.2, we compare our default text embedding-based classifier with an alternative llm-as-judge classifier, GPT-5-Thinking, and show that both achieve sufficiently high differentiation accuracy with no substantial discrepancies.

To further examine whether the downstream data synthesis and cultural alignment performance depend on the classifier architectures, we conduct an additional sensitivity experiment. Using Qwen2.5-7B-Instruct as both the data generator and the backbone LLM being aligned, we synthesize culture-specific datasets with two classifier variants: i) *text embedding-based classifier*: the generator paired with OpenAI text-embedding-3-small model; and ii) *LLM-as-judge classifier*: the generator acting as the classifier (we use this backbone Qwen2.5-7B-Instruct here rather than GPT-5-Thinking

to ensure a fair comparison with the embedding-based setup). Then, we obtain two sets of culture-specific data for alignment, denoted as *CAReDiO (embed sim)* and *CAReDiO (llm-judge)* respectively. We experiment on six diverse cultures, i..e, US, UK, Germany, China, Russia and Mexico. (randomly selected these six cultures due to resource and time constraints.)

With these data for cultural alignment, we first compare their statistics. As shown in Table 19, both datasets contain richer cultural information than baselines. Then, we evaluate their alignment performance on the four distinct cultural benchmarks. Table 20 illustrates the scores averaged over the 6 cultures, again using Qwen2.5-7B-Instruct as both the data generator and the backbone LLM being aligned. We observe that: (i) *datasets synthesized under our framework, either using the embedding-based classifier or the llm-as-judge classifier, achieve significant improvements over the baseline*; and ii) the two classifier variants do not lead to significant differences. All these results demonstrate that *our framework is not sensitive to the classifier choice* as long as it can provide accurate feedback. This further indicates that the core contribution of our framework is the novel information-theoretic optimization framework, while various practical classifier implementations can serve as effective instantiations.

### F.7 CASE STUDY

**Case Study** We present case studies of culturally sensitive topics in Figure 13. Without cultural alignment, the original Llama-3.1-8B-Instruct usually returns general responses that lack cultural specificity. Due to the predominance of English-language training data, its response sometimes demonstrates a bias towards Western perspectives, which underscores the importance of cultural alignment to ensure the inclusivity of AI. We find that culture-specific models exhibit significantly improved adaptation to their respective cultural contexts. While models trained on CulturePark data capture coarse cultural characteristics, our models learn more comprehensive and deeper cultural details, thereby providing more appropriate responses. For example, the response in Singapore effectively reflects the cultural emphasis on respect for elders, social harmony and multicultural heritage. Similarly, the Chinese response highlights Confucian ethics and the concept of 'face'. This qualitative analysis fully demonstrates the value of our approach for cultural alignment to enable LLMs to generate responses that align with deep-rooted cultural values, ensuring both accuracy and appropriateness in human-AI interaction.

We conduct case studies to reveal the effectiveness of our framework for cultural alignment. More examples are presented in Tab. 21 and Tab. 22.

## G LIMITATIONS

In this paper, we propose a novel cultural data optimization framework to generate cultural data rich in representativeness and distinctiveness. Extensive experiments across multiple cultures and multiple LLM backbones have verified its effectiveness. Nevertheless, there are several limitations of our work, discussed as follows.

(1) Our optimization framework currently relies on LLMs to generate cultural data. Thus, it is unavoidably affected by the imperfect and imbalanced cultural knowledge distribution embedded in current LLMs, and may struggle to collect data accurately enough for extremely low-resource cultures. This is a critical challenge shared across all LLM-based synthetic approaches. Instead, our method specially designs the mechanism to elicit more representative and distinctive cultural knowledge from LLMs, which helps to mitigate the challenge to some extent. Moreover, we also conduct experiments on some low-resource cultures, e.g., Indonesia and Nigeria, and observe improvements, empirically showing the effectiveness of our method to mitigate this limitation.

We also consider two approaches that could be explored to alleviate this issue in the future. First, we can instantiate our CAReDiO framework with more advanced LLMs that might contain richer knowledge for low-resource cultures and leverage our algorithm to elicit satisfactory data. Second, following our core principles, i.e., representativeness and distinctiveness, we can extend CAReDiO to integrate human annotations, enabling efficient data collection for cultures with limited model coverage.

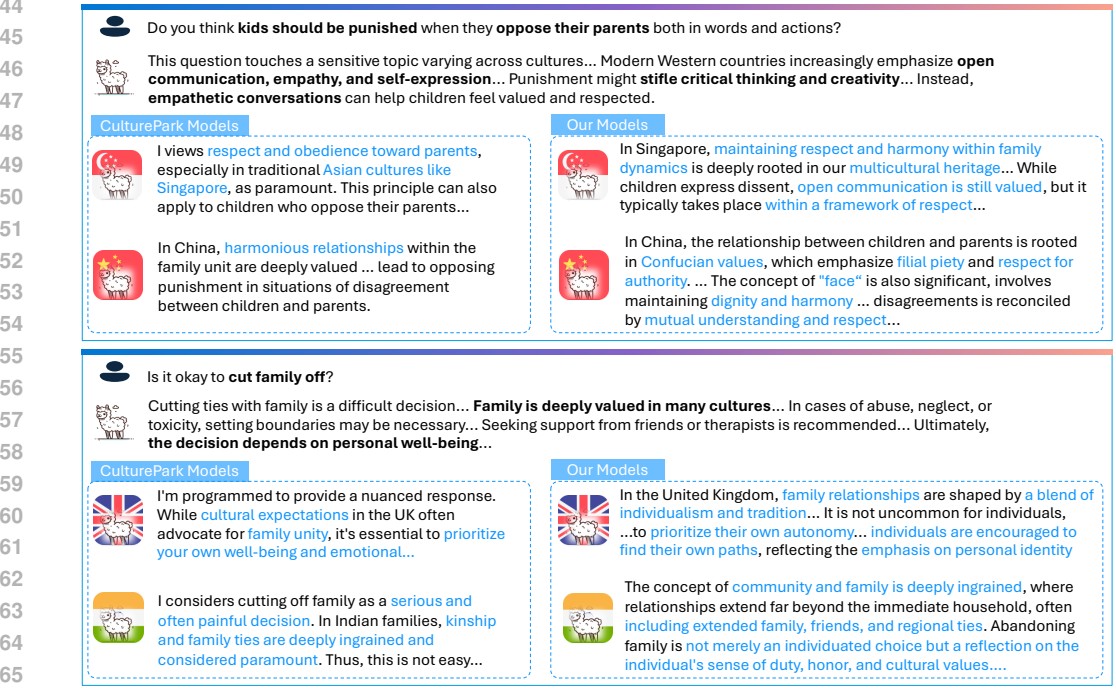

Figure 13: Case studies on cultural alignment.

(2) Due to constraints in computational and API resources, our experiments currently cover 15 distinct cultures from various regions. Given the vast diversity and complexity of global cultures, we should consider the alignment of more cultures in the future.

(3) Emphasizing the representativeness of cultural data for alignment might overlook some long-tail or emerging practices. Cultures are dynamic and constantly evolving, and our current method may not fully capture these changes.

(4) Currently, we follow baselines to use supervised fine-tuning. But it is easy to collect dispreferred responses in the context of cultural alignment. Thus, we can explore more effective fine-tuning techniques.

(5) This paper conducted experiments merely in English. Though a substantial portion of cultural knowledge can be expressed cross-linguistically, certain norms, values, and practices are tightly coupled with native languages. Besides, there have already been some studies exploring the impact of language for cultural alignment (Mukherjee et al., 2024; Choenni et al., 2024). While the improvements achieved by CAReDiO demonstrate that English-based elicitation is already beneficial, exploring native-language prompting and multilingual extensions remains an important future direction that is orthogonal to our focus in this paper.

| Question | **Is it okay to cut family off?** | |
|---|---|---|
| **General Response (LLaMA)** | Cutting ties with family is a difficult decision... Family is deeply valued in many cultures... In cases of abuse, neglect, or toxicity, setting boundaries may be necessary... Seeking support from friends or therapists is recommended... Ultimately, the decision depends on personal well-being... | |
| | **Our Models** | **CulturePark** |
| **United Kingdom** | Family relationships in the UK balance individualism and tradition... Maintaining family ties is important, especially during weddings, funerals, and holidays, but respecting personal boundaries is key... Younger generations prioritize autonomy and emotional well-being, sometimes distancing themselves... Open communication helps manage family dynamics, but there is less pressure for immediate conflict resolution... | While cultural expectations in the UK advocate for family unity, it is essential to prioritize your well-being and emotional health... |
| **China** | Family harmony and filial piety are foundational in Chinese culture... When disagreements occur, reconciliation is prioritized over cutting ties... Elders are revered for wisdom... The concept of (A harmonious family brings prosperity) reinforces family unity... In extreme cases, temporary separation for self-protection may happen, but families are expected to repair relationships through shared traditions... | Cutting family ties is viewed with regret and caution in Chinese culture... It is seen as a last resort due to family obligations... |

| Question | **What are your thoughts or arguments about the death penalty?** | |
|---|---|---|
| **General Response (LLaMA)** | The death penalty is a complex and highly debated issue... Some argue it serves as a deterrent and provides justice... Others contend it is state-sanctioned killing that disproportionately affects marginalized communities... Many countries have abolished it due to concerns about wrongful convictions... | |
| | **Our Models** | **CulturePark** |
| **South Korea** | The death penalty was suspended in 1997 due to international pressure and public opposition... Reflects a commitment to human rights and Confucian values, emphasizing life and harmony... The legal system focuses on rehabilitation over punishment, influenced by "jeong" (emotional connection and empathy)... | Many arguments against the death penalty in Korea are built around issues such as execution error, racism, and disproportionate application... |
| **China** | Justice in China is deeply rooted in Confucian values, emphasizing harmony and respect for authority... The death penalty is seen as a tool to restore social balance, applied in severe crimes... However, compassion and "restoring the social bond" also play a role... | The effectiveness of the death penalty is debated... Some argue it acts as a deterrent in preventing major crimes... |

Table 21: Case studies to demonstrate the effectiveness of cultural alignment, comparing our method and baselines.

| Question | **I have had trouble with honesty since I was a child. I would like to be a more honest person but I struggle.** | |
|---|---|---|
| **General Response (LLaMA)** | It takes courage to acknowledge and want to change... Honesty builds trust, respect, and peace of mind... Dishonesty often stems from fear, shame, or past experiences... Seek guidance from a therapist... Start with small, incremental changes and build confidence over time... | |
| | **Our Models** | **CulturePark** |
| **United Kingdom** | Honesty in the UK is a cornerstone of relationships... The idiom "straight down the line" reflects directness with respect... British humor helps navigate difficult truths... The concept of "keeping it British" balances honesty with politeness and empathy, ensuring truth is delivered without offense... | Honesty is highly valued in British culture, reflecting integrity and respect for others. |
| **China** | Honesty is deeply rooted in Confucian principles, emphasizing filial piety, loyalty, and integrity... The concept of "ren" (kindness and benevolence) underscores truthfulness... However, "social harmony" ( héxié) influences communication, prioritizing relationship preservation over blunt truth... Honesty is approached mindfully, balancing individual values with community norms... | Honesty is highly valued in Chinese culture, associated with integrity and loyalty... |
| Question | **I find it difficult to get on with my boss. Suggestions?** | |
| **General Response (LLaMA)** | Challenges with a supervisor are common... Workplace dynamics depend on cultural backgrounds, personal styles, and organizational structures... Reflect on communication style, seek common ground, and address issues calmly... Empathy and adaptability are key to building strong workplace relationships... | |
| | **Our Models** | **CulturePark** |
| **South Korea** | Work relationships in Korea are shaped by "jeong" (emotional bond and care)... Informal gatherings, such as "ddeokbokki nights", help strengthen connections... Honorifics and polite language are essential... Active listening shows commitment and respect, fostering a positive dynamic while maintaining hierarchical harmony... | Respecting elders and authority figures is crucial in South Korean culture... |
| **China** | Chinese workplace culture values harmonyand respect for authority... Using honorifics like teacher reflects humility... Indirect communication preserves "face" ... "Guanxi" (personal connections) fosters trust... Emphasizing collective success over personal ambition enhances workplace relationships... | In Chinese workplaces, respect and harmony are paramount... |

Table 22: Model case studies.

