# OpenReview forum: "CAReDiO: Enhancing Cultural Alignment of LLM via Representativeness and Distinctiveness Guided Data Optimization"
_ICLR.cc/2026/Conference — Submitted to ICLR 2026_

### Official Review · Reviewer_ai47 · 2025-10-31

**Soundness:** 2
**Presentation:** 2
**Contribution:** 3
**Rating:** 4
**Confidence:** 3

**Summary:**

The authors present a method for generating a dataset for cultural alignment. The algorithm proceeds as follows:
1. start with an initial set of seed prompts;
2. generate model responses for those prompts;
3. score the prompt-response pairs based on a combination of A. how relevant they are to a given culture (via an LLM judge) and B. how distinct the response is to responses associated with other cultures (measured by embedding similarity);
4. use an LLM to refine each question given the response.

The paper creates a dataset using this method. The paper also offers the theoretical insights supporting the relevance and distinctiveness ideas core to the prompting-based approach.

**Strengths:**

1. The paper tackles an important problem and offers a helpful conceptual framework to think about what is desirable in a dataset for cultural alignment.
2. The paper formalizes these concepts mathematically, to add theoretical weight to the intuition.
3. The figures are well-done, and the results include human evaluation.

**Weaknesses:**

1. Theory-aside, the paper offers a prompting-based method for generating a dataset. In that regard, it is not quite clear to me what this method does that’s more than / justifies better performance than other methods. Making this distinction in the practical implementation clear would be helpful.
2. If I understand correctly, there seem to be slight discrepancies between the mathematical objectives, algorithm box, and actual implementation based on prompting in the appendix. Making these more explicit would help with reader understanding. For instance, lines 247-254 seem to suggest inner loop optimization of the responses whereas the algorithm simply generates a response without additional optimization. And the notation for lines 4 and 5 in the algorithm box were a bit confusing as well as I could not find their precise definitions (e.g., with vs. without the subscript).
3. While the authors have various results to measure "cultural alignment," some more clarity on the precise experimental setups (and how they differ) would help. For instance, what is the difference between Figure 4b and Table 3, e.g. GPT-4.1 on CB-Hard?

**Questions:**

1. Focusing on the prompting method itself, what makes the proposed method better than some of the others that generated previous datasets? For instance, how do we know that the performance gap is actually attributable to the central ideas on representativeness + distinctiveness in a generalizable way and not just better models, longer prompts, more iterations, etc.?
2. I noticed the results in main would using CardSet whereas the appendix used CaReDiO. Is there some significant difference in the experimental setup? I had originally thought that the comparison was of fixed datasets.
3. What were the seed questions?
4. Could the authors add more context around each of the different results / figures? Namely, how each is set up and measured?
5. Could authors clean up theorem 1 and the proof? It seems pretty informal as it currently stands.

If the authors could address these questions and concerns, I would be willing to raise my score.

---

> ### Author Response · Authors · 2025-11-25
> **Response to Reviewer ai47 [1/3]**
>
> Thank you very much for your insightful comments and constructive feedback. We provide responses to each Weaknesses (W) and Question (Q) as follows. A revision of our paper has been uploaded, with the revised parts marked in blue.
>
> ### **W1: Theory-aside, it is not quite clear what this method does that's more than / justifies better performance than other methods. Making this distinction in the practical implementation clear would be helpful.**
>
> To address the reviewer's question, we provide a more intuitive explanation for why CAReDiO achieves better performance than other prompting-based methods.
>
> **Assumption**. Following the line of synthesizing cultural data with LLMs, we acknowledge two key assumptions (which often hold in practice).
> (i) The backbone LLMs, e.g., GPT-4.1, have encoded sufficient knowledge about specific cultures and can be elicited in a certain way.
> (ii) The cultural knowledge distribution learned by the LLMs is imperfect and biased, which can be more serious in weaker models.
>
> **Observation**. Based on these assumptions, we observed that most prior synthesis methods directly apply simple role-playing, and usually produce biased or ambiguous data.
>
> **Source of gains**. This improvement by CAReDiO is obtained by explicitly shaping the data distribution along two theoretically grounded criteria derived from cultural theory:
>
> + **Representativeness**: CAReDiO utilizes a multi-LLM ensemble to maximize the shared understandings of the target culture while *minimizing irrelevant noise (corner cases) of generated samples* (Eq.(2)-(3)), which helps *mitigate the bias of single LLM* role-playing. **Evidence**: Optimizing only Representativeness yields + 2.28 on the Prism benchmark compared to naive role-playing (see our response to Q1). In Table 3, with (weaker) Qwen2.5-7B-Instruct as the backbone, on difficult tasks (CB-Hard), Role-Play performs much worse than CAReDiO (40.20 > 36.73).
> + **Distinctiveness**: CAReDiO captures characteristics unique to the target culture rather than generic patterns shared across related cultures (Eq.(4)). *This further distinguishes ambiguous data samples located near cultural boundaries*. **Evidence**: For closely related cultures, e.g., Japan/Korea/China, CAReDiO achieves larger performance gaps between target and non-target cultures, △=5.3 > 2.4, while ensuring better performance on each target one (refer to our response to Reviewer HCb7's W4).
>
> To further enhance optimization efficiency, we embed these two objectives into an RL-style iterative optimization procedure (Algorithm 1). The LLM receives examples with their representativeness and distinctiveness scores and learns to refine questions and responses to produce progressively higher-quality data.
>
> Although CAReDiO is prompt-based, its concrete designs are grounded in information-theoretic derivation, which differentiates it from existing methods and explains its superior empirical performance.
>
> We added these explanations in Appendix F.2 of the revised version.
>
> ---
>
> ### **W2: If I understand correctly, there seem to be slight discrepancies between the mathematical objectives, algorithm box, and actual implementation based on prompting in the appendix. Making these more explicit would help with reader understanding.**
>
> Sincere apologies for the confusion. The mathematical expressions, algorithm, and implementation in the appendix are aligned at the framework level, following the iterative and joint question-response maximization in Eq.(5), but there are some missing details/explanations due to space limits.
>
> To improve clarity and address the concern, we have:
> i) Reorganized Sec. 3.2 and 3.3 to more clearly connect each module's theoretical objective with its practical implementation.
> ii) Polished Algorithm 1 and add more details.
>
> Regarding the reviewer's two specific points, we make the following clarification.
>
> + **Discrepancy between L247-254 and Algorithm 1.**
> The text states that "at the t-th iteration, we fix the $x^{t-1}$ and instruct the LLM to generate $y^t$ that maximizes Eq.(5)." This means that we use the prompt in Fig. 7 (Appendix B.2) to generate multiple responses $y^t$, compute their scores using Eq. (5) with prompts in Fig.8,9 (Appendix B.2), and retain the high-scoring ones. There is no additional inner-loop optimization; that is, we conduct in-context learning based generation, scoring and selection as 'optimization'. This matches the process described in Algorithm 1 (Lines 4–5).
> + **Notation in Algorithm 1 (Lines 4–5).**
> Both $x^{t-1}$ and $y^{t}$ should include the sample index $i$ as $x^{t-1}\_i$ and $y\_i^{t}$, as we synthesize and optimize multiple questions and responses, which we have corrected in the revised pdf. They follow the same primary notation as $(x, y)$ in the main text and denote the individual samples among the whole dataset $\(x_i\)\_{i=1}^N$.

---

> ### Author Response · Authors · 2025-11-25
> **Response to Reviewer ai47 [2/3]**
>
> The optimization process relies on LLM in-context learning (ICL) (to make CAReDiO compatible with black-box LLMs whose generation probabilities are inaccessible), and therefore inevitably involves some approximation in practice. The empirical effectiveness of our approach (Sec.4.2 and Sec.4.3) demonstrates that this is acceptable.
>
> We have carefully checked the paper, fixed and clarified issues, and polished the presentation accordingly. We believe the updated version substantially improves readability and consistency.
>
> ---
>
> ### **W3, Q4: Clarify the experimental setups for different results/figures.**
> Thank you for the valuable suggestions. Due to page limits, most details of experimental setups (e.g., benchmarks), have been placed in **Appendix C.1** of the initial version.
>
> To improve the readability, we have revised the paper accordingly. Below we summarize the key setup for each result (figure or table) to clarify their differences.
>
> **Sec. 4.3 Cultural Alignment Performance.** As claimed in L429-462, this experiment evaluates the performance of LLMs aligned to 15 cultures using both CAReDiO-synthesized data and baseline datasets, e.g., CultureBank.
>
> + Table 3 reports average scores across all 15 cultures on four benchmarks. The "Family" column represents the backbone LLMs being aligned, while the "Method" column means using different datasets to align the same backbone LLM.
> + Fig. 4(a) presents human evaluation results of the alignment performance (L500-505) (aggregated win rate on US, China, Japan and Poland cultures). The backbone being aligned is Qwen2.5-7B-Instruct, compared to the role-playing baseline, the CulturePark baseline, and the role-playing baseline with GPT-4.1 as the backbone.
> + Fig. 4(b) compares alignment results of CAReDiO instantiated by different LLMs, i.e., Qwen2.5-7B-Instruct, GPT-4.1 and GPT-5, to synthesize cultural data, while keeping the backbone being aligned fixed as Qwen2.5-7B-Instruct.
>
> **Sec. 4.4 Hyperparameters Analysis.** These experiments are all conducted with qwen2.5-7b-instruct as the backbone to be aligned.
>
> + Fig. 5(a) shows performance when qwen2.5-7b-instruct is aligned with different amount of CAReDiO-synthesized data. Results are averaged over all 15 cultures.
> + Fig. 5(b) shows alignment performance of CAReDiO with qwen2.5-7b-instruct as the data generator, using data generated in different optimization rounds ($T$ in algorithm 1).
>
> Apologies for the vagueness. We have incorporated these clarifications into the revised PDF to ensure the distinctions among figures and tables are explicit and easy to follow.
>
> We believe these easy-to-fix presentation issues would not impede our core contributions of method, empirical, and theoretical results.
>
> ---
>
> #### **Q1: How do we know that the performance gap is actually attributable to the central ideas on representativeness + distinctiveness in a generalizable way and not just better models, longer prompts, more iterations, etc.?**
>
> 1. **Using the same LLM for synthesis, CAReDiO performs better**. Larger models and longer prompts can also improve performance for prompting-based data synthesis. Actually, *we empirically demonstrated it in Fig.4(b)*, where we endowed CAReDiO with LLMs of varying capability, i.e., Qwen2.5-7b-instruct, GPT-4.1, GPT-5, for data synthesis, and observed consistently better results with stronger models. However, **note that we used the same LLMs for synthesis for all methods in comparison**. In Table 3, all synthesis methods use the *same LLM* (e.g., Qwen2.5-7B-Instruct), both for data generation and as the target model to be aligned afterward. In such a fair comparison, *CAReDiO consistently outperforms all baselines when endowed with the same backbone LLM*.

---

> ### Author Response · Authors · 2025-11-25
> **Response to Reviewer ai47 [3/3]**
>
> 2. **Further validation**. We directly validate the independent contribution of our core principles, i.e., representativeness and distinctiveness, through additional experiments below:
>
> (1) **Correlation Analysis** between the synthesized data with different representativeness + distinctiveness scores (calculated by Eq.(5)) and the alignment results (Fig. 5b). Holding other variables constant (synthetic model, prompt length, etc.), *we observe higher-scoring data yield stronger cultural alignment*, demonstrating that these core principles themselves matter.
>
> (2) **Ablation Studies (Appendix F.2).** We further conduct an additional ablation study to show the contribution of each objective and their combination. As shown in the table below, *both objectives can independently improve the performance significantly*. Only Rep + 15.3% averaged on benchmarks, only Dist + 13.5%, while *their combination (CAReDiO) yields the best (+20%)*, compared to the backbone LLMs.
>
> | **Method**              | **CB-Easy** | **CB-Hard** | **Prism** | **GlobalOpinion** | **Avg** |
> |-------------------------|-------------|-------------|-----------|-------------------|---------|
> | **Qwen2.5-7B-Instruct** | 72.01       | 38.90       | 42.06     | 53.28             | 51.56   |
> | **Role-Play**           | 72.38       | 36.73       | 67.28     | 55.83             | 58.05   |
> | **only Rep**            | **74.53**   | 38.70       | 69.56     | 54.94             | 59.43   |
> | **only Dist**           | 72.18       | 38.14       | 68.72     | 54.95             | 58.50   |
> | **CAReDiO**             | 73.48       | **40.20**   | **77.42** | **56.23**         | **61.83**   |
>
> Please also refer to our response to W1 for an intuitive explanation about why these two objectives work.
>
> In summary, these additional experiments indicate that the proposed representativeness and distinctiveness principles make a clear and measurable contribution to cultural alignment.
>
> ---
>
> #### **Q2: I noticed the results in main would use CardSet whereas the appendix used CAReDiO. Is there some significant difference in the experimental setup?**
>
> We apologize for the confusion. CAReDiO is the proposed framework for cultural data optimization (Algorithm 1), while CARDSet is the dataset synthesized by CAReDiO (as noted in Line 307, Sec.3.3), which is an instantiation of the CAReDiO framework for the 15 cultures studied in this paper.
>
> With this distinction in mind, our usage is consistent in the paper:
> + We refer to CARDSet when discussing experiments regarding the synthesized dataset, e.g., Sec. 4.2 Cultural Data Analysis.
> + We refer to CAReDiO when comparing overall cultural-alignment performance against baseline methods, since the comparison is at the framework level rather than the dataset level.
>
> Importantly, the alignment through CAReDiO uses the same fixed CARDSet dataset to ensure fairness.
>
> We have proofread and revised the paper to make this terminology distinction clearer and avoid further confusion.
>
> ---
>
> #### **Q3: What were the seed questions?**
> As described in *L259–264 of the initial submission (L311-317 in the revised pdf) and Appendix B.1*, our seed questions are constructed in two stages. First, the LLM $p_w$ generates initial questions covering a diverse set of topics from core culture aspects (listed in Appendix B.1). Next, following the Self-Instruct procedure, we synthesize diverse question variants for each topic. We then combine the initial questions with their Self-Instruct expansions to obtain the final set of seed questions $(x^0_i)_{i=0}^N$used for optimization.
>
> Concrete prompts for the initialization and Self-Instruct step are provided in Appendix B.2, Fig.6.
>
> ---
>
> #### **Q5: Could authors clean up theorem 1 and the proof? It seems pretty informal as it currently stands.**
>
> Thank you for the helpful suggestion. In the updated paper, we have carefully reviewed this part to ensure full mathematical clarity and rigor. Specifically, we (i) improve structure and formalization of the theorem and proof, (ii) ensure all notation is formally defined before use, and (ii) provide complete derivations.
>
> ---
>
> **We hope our clarifications and additional experimental results address your concerns, and we are happy to respond to any further questions.**
>
> **We would sincerely appreciate it if you could read our responses and kindly reconsider the assessment of our work.**

---

### Official Review · Reviewer_HCb7 · 2025-10-31

**Soundness:** 2
**Presentation:** 2
**Contribution:** 2
**Rating:** 4
**Confidence:** 4

**Summary:**

This paper addresses the problem of cultural bias in LLMs and argue that existing cultural alignment datasets are insufficient because they fail on two key dimensions, which they ground in established culture theories: (1) Representativeness, or the failure to capture the consensus characteristics of a culture (the "emic" view) , and (2) Distinctiveness, or the failure to capture the unique nuances that differentiate a culture from other, similar ones (the "etic" view). To solve this, the paper proposes CAREDIO, a data optimization framework. CAREDIO is an in-context, LLM-driven process for generating and refining high-quality, culture-specific training data. It formalizes the two dimensions as information-theoretic objectives. The authors then use the approach to generate a new dataset, covering 15 cultures. Their experiments show that fine-tuning both small and large LLMs on as few as 200 samples from the dataset outperforms models trained on existing, often larger, cultural datasets.

**Strengths:**

1. The proposed method is grounded in clear mathematical formalisations, derived directly from information-theoretic quantities and supported with detailed proofs. The formulations for mutual information-driven sample selection and Jensen-Shannon-based culture divergence are actionable.
2. The quantitative dataset analysis is thorough. Table 2 compares diversity, information content, and similarity metrics across datasets, supporting the claim that proposed dataset is both more informative and more diverse.
3. Human evaluation demonstrates the practical significance and that the approach moves beyond benchmark overfitting.
4. The proposed approach needs fewer than 200 training samples for strong performance. Figure 5 shows early high-value data boosts alignment more per sample, which highlights the method's efficiency claims.

**Weaknesses:**

1. The paper's core premise lies in the two dimensions, but the introduction does not clearly articulate them. Figure 1 is also unclear, leaving the reader with a poor intuition for what "representativeness" means independently of distinctiveness.
2. In Section 3.2, the “distinctiveness” objective (Eq. 4) relies on φ(y, x) as a probability that responses are not from other cultures, implemented via a “clustering-based distance measurement”. This is left quite vague: how is this classifier trained, what architectures/embeddings are used, and how sensitive are results to this choice? This is described only briefly.
3. Results demonstrate that the proposed approach is better, but no analysis shows how much each dimension (representativeness / distinctiveness) contributes to final model performance.
4. Theoretically, the distinctiveness objective is vital for closely related cultures (e.g., Japan/Korea/China). However, Table 5–9 largely report aggregate or per-country results, with only sparse discussion of confusion between “neighbour” cultures. It would be informative to see confusion matrices or analysis on these closely related pairs, does CAREDIO really outperform baselines in making fine distinctions?

**Questions:**

1. See Weaknesses for some questions.
2. Can you report results (or discuss limitations) for adapting the proposed approach to “zero-shot” or unseen cultures? What changes are needed to ensure transferability and avoid overfitting to the 15 present cultures?
3. How do you control for or report on annotator disagreement in human evaluations? Do any particular cultures or question types show especially high variance, and does that impact major conclusions?

---

> ### Author Response · Authors · 2025-11-25
> **Response to Reviewer HCb7 [1/5]**
>
> Thank you very much for your insightful comments and constructive feedback. We provide responses to each Weakness (W) and Question (Q) as follows. A revision of our paper has been uploaded, with the revised parts marked in blue.
>
> ### **W1: The paper's core premise lies in the two dimensions, but the introduction does not clearly articulate them. Figure 1 is also unclear, leaving the reader with a poor intuition for what "representativeness" means independently of distinctiveness.**
>
> Thanks for the valuable feedback.
>
> 1. **The meanings of representativeness and distinctiveness were explicitly defined** in L59-70, which have also been mathematically expressed and discussed in L153-160, 210-212, and 241-249. For better understanding, we further clarify the two dimensions below.
>
> + *Representativeness*: An internal view of culture, which means how strongly a sample $(x,y)$ reflects the most salient and central aspects of a culture, rather than trivial or corner cases. Mathematically, how much probability mass is assigned to $(x,y)$ in the true culture distribution, i.e., how large $p_c(x,y)$ is.
> + *Distinctiveness*: An external view to differentiate one culture from others, which captures how a sample $(x,y)$ captures the unique nuances of the target culture, instead of patterns broadly shared across related cultures, as shown in Eq.(1).
>
>     These two dimensions are not always tied together. A highly representative $(x,y)$ may not be distinctive if it is also common in other cultures. Conversely, a concept can be distinctive but not representative if it reflects a rare or peripheral case.
>
> 2. **The explanation of Fig. 1**. The upper-left example ('*talk to
> strangers*') is high in distinctiveness (different answers in China and the US) but is low in representativeness (an uncommon case). The upper-right example is highly representative (the popular 'family relationship' topic) but weakly distinctive (same answers in China and US). Move down, the example illustrates cases that score high on both dimensions by centering on culturally important topics ('balance between individual and society') where the two cultures diverge.
>
> To further help readers' understanding of our work, we've conducted a comprehensive polishing in the revised pdf, including:
>
> i) We polished Fig.1 and added more description in the caption.
>
> ii) We revised the introduction (L58-75) to articulate these two dimensions more explicitly.
>
> iii) We reorganized Sec.3.2 and Sec.3.3, presented each variable and module with concrete examples, and highlighted the two dimensions.
>
> iv) We refined Fig.2 and, involving concrete notations and examples, show how CAReDiO is connected to these two dimensions.
>
> We believe the polishing improves the paper's overall readability, and these easy-to-fix presentation issues would not impede our core contributions of our novel method, empirical and theoretical results.
>
> ### **W2: This is left quite vague: how is the Distinctiveness classifier $\phi(y, x)$ trained, what architectures/embeddings are used, and how sensitive are results to this choice? This is described only briefly.**
>
> Apologies for missing the details of classifier $\phi(x,y)$. We added these details in Sec.3.2 (L250-258), Appendix C.2, Appendix F.6 of the revised version, and explain how it is implemented and how well it performs to address this concern below.
>
> + **Implementation Details**: Due to high-quality annotated culture data is limited, we implement the classifier $\phi(x,y)$ using the backbone LLM for synthesis $p_w$ (e.g., GPT-4.1), together with the openai text-embedding-3-smal model, *without any additional training.* Specifically, for a sample $(x,y)$ and the target culture $c$, we ask the LLM to generate responses for the target culture $y_c$ and for other $K$ non-target cultures $(y_{c_1}, ..., y_{c_K} )$. We encode all responses with the embedding model to obtain $(e_y, e_c, e_{c_1}, ..., e_{c_K})$, and then calculate the probability that the response $y$ does not come from any of the $K$ non-target cultures as:
> $$
> P(y \in c \ \text{and}\ \notin c_i) = \frac{\exp(sim(e_y, e_c))}{\exp(sim(e_y, e_c)) + \sum_{i=1}^{K}(\exp(sim(e_y, e_{c_i})))}, c_i \neq c.
> $$
> This requires $y$ to be semantically closer to same-culture response $y_c$ than to cross-culture ones.
>
> + **Accuracy and Reliability of the Classifier**: We futher invite human experts to manually label 200 samples $(x,y,c,p*)$ for each culture, and conduct an experiment to assess the reliability of our classifier.

---

> ### Author Response · Authors · 2025-11-25
> **Response to Reviewer HCb7 [2/5]**
>
> We report four metrics: i) average probability error $\epsilon=|\phi(x,y) – p*|$) used in Eq.(4); ii) the approximmention error bound $e$ mentioned in Proposition 2; iii) classification accuracy, Acc.; and iv) classification F1. The results are as below:
>
>
> | **culture**  | **United States** | **China** | **Japan** | **Poland** |
> |--------------|-------------------|-----------|-----------|------------|
> | **Acc.**     | 0.9051            | 0.9350    | 0.9450    | 0.9350     |
> | **F1**       | 0.9064            | 0.9366    | 0.9442    | 0.9319     |
> | **$\epsilon$**  | 0.1682            | 0.1803    | 0.0973    | 0.0964  |
> | **$e$**    | 0.0160            | 0.0330    | 0.0290    | 0.0485     |
>
> We can see: a) **Our simple embedding similarity based classifier is highly accurate (with F1 > 0.9)**. This is because semantic differences between cultures are not small, even simple methods perform well, while previous work has largely ignored distinctiveness. b) Note that in Eq.(4), we do not use the binary prediction, but rather the probability $\phi(x,y)$. Therefore, **approximation error bound $e$** matters more, **which is very small (smaller than 0.05)** compared to its upper bound ($logK = 2.64$) in this work. Please refer to Appendix C.2 and our response to Reviewer g389's W1 for more detailed discussion about the classifier error bound.
>
> + **Sensitivity to Other Choice of Classifier**. Besides the embedding-based one, we also tested the accuracy of an alternative LLM-as-judge classifier, *GPT-5-thinking*. The results are as below:
>
>     | **culture**  | **United States** | **China** | **Japan** | **Poland** |
>     |--------------|-------------------|-----------|-----------|------------|
>     | **Acc.** | 0.9650            | 0.9550    | 0.9700    | 0.9100     |
>     | **F1** | 0.9648            | 0.9577    | 0.9714    | 0.9100     |
>     | **$\epsilon$**  | 0.1981            | 0.1358    | 0.1206    | 0.1635
>     | **$e$**    | 0.7952            | 0.5452    | 0.5061    | 0.1902     |
>
>     We can see that LLM-as-judge with GPT-5-thinking achieves slightly better ACC. and F1, but also larger probability error $\epsilon$, and thus leads to much worse error bound $e$, because these off-the-shelf LLMs often fail to provide continuous score/probability. Overall, **both classifiers achieve sufficient accuracy in cultural differentiation.**
>
>     Then, we further compare CAReDiO with the two kinds of classifiers on downstream data synthesis and cultural alignment, denoted as CAReDiO (embed sim) and CAReDiO (llm-judge). With full details in Appendix F.6, we report the key findings two perspectives below.
>
>     (1) *Quality of Synthesized Culture-specific Data*.
>
> | **Dataset**             | **Avg.L $\uparrow$** | **#Cultural Points$\uparrow$** | **Sim $\downarrow$** | **Self-Blue$\downarrow$** | **Cross-Culture Sim$\downarrow$** |
> |-------------------------|-----------|----------------------|---------|---------------|-----------------------|
> | **CulturePark**         | 68.6      | 494.6                | **0.235**   | 0.406         | 0.223                 |
> | **CAREDiO (embed sim)**     | **200.4**     | **2027.0**               | *0.251*   | *0.324*         | **0.202**                 |
> | **CAReDiO (llm-judge)** | *192.0*       | *2021*                 | 0.257   | **0.321**         | *0.205*                 |
>
> **Both variants yield dataset with much richer cultural information** (more unique cultural points), and **lower intra-/inter-cultural similarity** than baselines.
>
> (ii) *Alignment Performance*
>
> | **Method**              | **CB-Easy** | **CB-Hard** | **Prism** | **GlobalOpinion** | **WVS** | **Average** |
> |-------------------------|-------------|-------------|-----------|-------------------|---------|-------------|
> | **Qwen2.5-7B-Instruct** | 74.85      | 42.38      | 2.154     | 53.65            | 61.70   | 53.49       |
> | **Role-Play**           | 76.77      | 41.21      | 3.471     | 56.79            | 66.01   | 61.05       |
> | **CAReDiO (embed sim)**     | **76.92**      | 42.04      | **4.002**     | 57.55            | **66.43**   | **64.13**       |
> | **CAReDiO (llm-judge)** | 76.34      | **42.59**      | 3.975     | **57.75**            | 66.25   | 64.05       |
>
> **Both variants**, either using embedding-based or llm-as-judge classifier, **significantly improve cultural alignment over baselines, without obvious performance gap** (64.13, 64.05 > 61.05).
>
> All the results above demonstrate **our current classifier is sufficiently stable and accurate for our optimization objective. And our framework is robust to this choice**. Please also note that $\phi(x,y)$ can be implemented in different architectures. This is not our core contribution, but just an implementation of our novel optimization method (Algorithm 1). We will explore more structures in the future.

---

> ### Author Response · Authors · 2025-11-25
> **Response to Reviewer HCb7 [3/5]**
>
> ### **W3: No analysis shows how much each dimension (representativeness / distinctiveness) contributes to final model performance.**
>
> Thanks for your valuable suggestion. We add an ablation study on independent optimization objectives. Please refer to Appendix. F.2 for full experimental settings and result analysis. We present the key results below:
> | **Method**              | **CB-Easy** | **CB-Hard** | **Prism** | **GlobalOpinion** | **Avg** |
> |-------------------------|-------------|-------------|-----------|-------------------|---------|
> | **Qwen2.5-7B-Instruct** | 72.01       | 38.90       | 42.06     | 53.28             | 51.56   |
> | **Role-Play**           | 72.38       | 36.73       | 67.28     | 55.83             | 58.05   |
> | **only Rep**            | **74.53**   | 38.70       | 69.56     | 54.94             | 59.43   |
> | **only Dist**           | 72.18       | 38.14       | 68.72     | 54.95             | 58.50   |
> | **CAReDiO**             | 73.48       | **40.20**   | **77.42** | **56.23**         | **61.83**   |
>
> From the table above, we observe:
>
> + **Both objectives can independently improve the performance significantly**. Only Rep + 15.3% averaged on benchmarks, only Dist + 13.5%, while their combination (CAReDiO) yields the best (+20%), compared to the backbone LLMs.
> + **Optimizing representativeness alone outperforms distinctiveness alone**, indicating current LLMs contain biased cultural knowledge, while Eq.(3) more effectively mitigates them. Adding Dist on top of Rep brings additional gains, suggesting that Eq.(4) helps better identify cultural boundaries and resolve ambiguity.
>
> ---
>
> ### **W4: Theoretically, the distinctiveness objective is vital for closely related cultures (e.g., Japan/Korea/China). It would be informative to see confusion matrices or analyses on these closely related pairs.**
>
> Thanks for your valuable suggestion. We conducted additional analyses to directly evaluate whether CAReDiO improves fine-grained distinctiveness among closely related cultures (China/Japan/Korea). We report results from two perspectives.
>
> 1.**Distinctiveness of Synthesized Culture-Specific Data**.
>
> We embed our synthesized data for China, Japan, Korea, and those from the CulturePark baseline, using OpenAI text-embedding-3-small API, then conduct TSNE dimensionality reduction. The visualization is shown in Appendix F.3. **We observe that CAReDiO produces much more clearly separated clusters, while CulturePark exhibits substantial cross-cultural overlap**. To quantify this, *we compute the average inter-cluster centroid distance: CAReDiO: 0.47, CulturePark: 0.29*. This substantial margin indicates that CAReDiO’s synthesized data provides significantly finer distinctiveness than the baseline.
>
> 2.**Confusion Matrices of Cross-Cultural Evaluation**
>
> We further evaluated models aligned to China/Japan/Korea using either CAReDiO or CulturePark, then *measured their alignment to non-target cultures to derive confusion matrices*. We assessed two cultural benchmarks.
>
> **(1) GlobalOpinionQA**
>
> In the Tables below, we can see: i) **Models aligned via CAReDiO perform better on the target culture while worse on non-target cultures**, achieving improved cultural specificity than CulturePark (larger performance gaps △, 5.3 > 2.4).
>
> ii) On the conflicting subset (different answers are expected from distinct cultures), **CAReDiO models exhibit lower alignment on non-target cultures** (Acc. on Conflicting, 31.9 < 34.8).
>
> (a) Results of CulturePark
> | **Country** | **China** | **Japan** | **Korea** | **△**$\uparrow$ | **Acc. on Conflicting**$\downarrow$ |
> |-------------|-----------|-----------|-----------|-------|--------------------------|
> | **China**   | **52.41**     | 53.35     | 51.59     | -0.06 | 38.34  |
> | **Japan**   | 42.74     | **53.38**     | 52.18     | 5.92  | 30.86  |
> | **Korea**   | 49.47     | 50.47     | **51.41**     | 1.44  | 35.13  |
>
> (b) Results of CAReDiO
> | **Country** | **China** | **Japan** | **Korea** | **△**$\uparrow$ | **Acc. on Conflicting**$\downarrow$ |
> |-------------|-----------|-----------|-----------|-------|--------------------------|
> | **China**   | **53.88** | 50.84     | 49.47     | 3.73 | 34.69  |
> | **Japan**   | 45.25     | **54.56** | 53.70     | 5.09 | 33.73  |
> | **Korea**   | 45.23     | 45.54     | **52.46** | 7.08 | 27.29  |

---

> ### Author Response · Authors · 2025-11-25
> **Response to Reviewer HCb7 [4/5]**
>
> **(2) The Prism Dataset**
>
> Using 200 open-ended questions sampled from the Prism dataset, we measured *pairwise similarity between responses produced by China-, Japan-, and Korea-aligned models*. **CAReDiO models display lower cross-cultural similarity (higher Δ) than CulturePark** (0.31 > 0.27), confirming better separation across related cultures.
>
> (a) Results of CulturePark
> | **Country** | **China** | **Japan** | **Korea** | **△** $\uparrow$  |
> |-------------|-----------|-----------|-----------|--------|
> | **China**   | 1         | 0.73      | 0.72      | 0.28   |
> | **Japan**   | 0.73      | 1         | 0.73      | 0.27   |
> | **Korea**   | 0.72      | 0.73      | 1         | 0.27   |
>
> (b) Results of CAReDiO
> | **Country** | **China** | **Japan** | **Korea** | **△**  $\uparrow$   |
> |-------------|-----------|-----------|-----------|--------|
> | **China**   | 1         | 0.71      | 0.70      | 0.30   |
> | **Japan**   | 0.71      | 1         | 0.68      | 0.31   |
> | **Korea**   | 0.69      | 0.68      | 1         | 0.31   |
>
>
> Across all analyses, cluster distinguishability, centroid distances, confusion matrices, and cross-cultural response similarity, **CAReDiO consistently demonstrates a stronger ability to make fine-grained distinctions between closely related cultures**.
>
> Full experimental details and result analysis are provided in Appendix F.3.
>
> ---
>
> #### **Q2: Can you report results (or discuss limitations) for adapting the proposed approach to “zero-shot” or unseen cultures? What changes are needed to ensure transferability and avoid overfitting to the 15 present cultures?**
>
> We would like to clarify that though we instantiated the framework on 15 cultures in experiments, **CAReDiO is culture-agnostic, not tied to these specific cultures**. The distribution set $(p_{c_1} ,…,p_{c_K})$ used in Eq.(1)-Eq.(5) is only a formal placeholder without any constraints on $K$ and each $p_{c_i}$. **Our method can transfer naturally to zero-shot and unseen cultures without requiring any modification to the framework**.
>
> For a new or unseen culture, one can directly follow Algorithm 1 to first generate an initial set of cultural questions and then iteratively optimize for representativeness and distinctiveness to obtain cultural data for alignment. However, CAReDiO currently relies on LLMs' culture competence (mentioned in L202,1883). **Note that it is challenging for extremely low-resource cultures that are poorly represented in the underlying LLMs**. Empirical results show that CAReDiO can obtain large gains in low-resource cultures, e.g., Russia and Poland, but it may still fail in the extreme ones. This is an inherent limitation shared by all LLM-based data synthesis approaches.
>
> Please refer to our response to Reviewer UgnK's W1, Appendix G. Limitations, and Tables 9-13 for more results and discussions about this limitation.
>
> ---
>
> #### **Q3-1: How do you control for or report on annotator disagreement in human evaluations?**
>
> The details of human evaluation, including human recruitment, annotation guidelines, and quality control, have been provided in Appendix C.3 of the initial version (D.3 of the updated version).
>
> We further clarify the control of annotator disagreement as follows.
>
> Our human evaluation measures the cultural alignment of responses generated by LLMs aligned using different methods. Since this task is inherently subjective, we do not enforce a single "correct" label. Instead, we focus on minimizing noise due to inconsistent understanding of the rubric.
>
> To this end, we provide annotators with clear, fine-grained scoring rubrics (listed in Appendix D.3) and conduct mandatory training sessions before annotation. This ensures that disagreements stem from genuine cultural differences rather than misunderstandings of the guidelines.
>
> We also report inter-annotator agreement (IAA). The table below shows agreement scores across cultures and alignment methods:
>
> | **IAA**                | **China** | **Japan** | **Poland** |
> |------------------------|-----------|-----------|------------|
> | **Role-Play (Qwen2.5-7B)**          | 0.879     | 0.596     | 0.654      |
> | **Role-Play (GPT-4.1)** | 0.867     | 0.571     | 0.624      |
> | **CulturePark**        | 0.848     | 0.652     | 0.752      |
> | **CAReDiO**            | 0.806     | 0.880     | 0.795      |
> | **Avg**                | 0.850     | 0.675     | 0.706      |
>
> **The average agreement is 0.85, indicating a strong agreement for subjective NLP/LLM evaluation tasks** (with most values exceeding 0.7). This demonstrates that our annotation process is reliable and that annotator divergence remains within an acceptable and meaningful range.

---

> ### Author Response · Authors · 2025-11-25
> **Response to Reviewer HCb7 [5/5]**
>
> #### **Q3-2: Do any particular cultures or question types show especially high variance, and does that impact major conclusions?**
>
> All detailed per-culture evaluation results across all four benchmarks have been included in Table 5-9 of the initial version. These results consistently show that CAReDiO improves cultural alignment in most settings.
>
> To more directly address this concern, we conduct additional experiments and have the following two key findings:
>
> + **The average and standard deviation of gains (improvement over the backbone LLM) across cultures**. As shown in the Table below, on three benchmarks (Prism, GlobalOpinionQA and WVS), CAReDiO yields small/acceptable variance (std) compared to the average. Besides, we also observe CAReDiO achieves improvements on most cultures, e.g., 100\% improvement ratio on Prism, GlobalOpinionQA and WVS, 73.3\% on CultureBench.
>
>     | **Dataset**         | **Avg** | **Std** | **Ratio of Improved Culture** | **Ratio of Decreased Culture** |
>     |---------------------|---------|---------|-------------------------------|--------------------------------|
>     | **CB-Easy**         | 1.47    | 6.88    | 73.30%                        | 26.70%                         |
>     | **CB-Hard**         | 1.30    | 4.85    | 73.30%                        | 26.70%                         |
>     | **Prism**           | 35.35   | 4.61    | 100%                          | 0                              |
>     | **GlobalOpinionQA** | 2.95    | 3.10    | 100%                          | 0                              |
>     | **WVS**             | 5.00    | 2.47    | 100%                          | 0                              |
>
>
> + **The improvement ratio of CAReDiO across each question type and each culture**. We classify all benchmarks into three categories of question type: i) Multi-choice questions about culture knowledge (CB-Easy, CB-Hard), ii) Value Questionnaire (GlobalOpinionQA and WVS) and iii) Open-ended question (Prism). As shown in the Table below, CAReDiO obtains consistent improvements on most cultures and question types, with two general conclusions:
>
>     a) *CAReDiO performs better on moderately low-resource cultures*, e.g., Germany and Russia. This is because our representativeness objective (Eq.(3)) helps elicit consensus and mitigates LLMs' inherent bias towards minority cultures. Please also refer to our response to Reviewer UgnK's W1 and Appendix C.1 for analysis on low-resource cultures.
>
>     b) *CAReDiO achieves larger improvements on open-ended questions*. This is because our synthesized data takes the form of (x,y) QA pairs. We adopt this format due to that i) most baselines are rooted in multiple-choice questions (e.g., CulturePark and CultureSPA) and tend to overfit to that format, and thus fail to generalize to more realistic open-ended QA tasks; and ii) this format is more informative and better aligns with practical usage of LLMs. The results indicate that *CAReDiO still shows improvements across diverse question types, indicating that this form of data offers better generalization.* Please refer to our response to Reviewer g389's W2 and Appendix D.6 for discussions on format overfiting and leakage.
>
>   | **Question Type**                   | **United States** | **United Kingdom** | **Germany** | **Japan** | **Indonesia** | **Russia** | **Poland** | **Romania** |
>   |-------------------------------------|-------------------|--------------------|-------------|-----------|---------------|------------|------------|-------------|
>   | **Multi-Choice Cultural Knowledge** | 2.60%             | -4.76%             | 13.63%      | 0.78%     | 8.88%         | 4.76%      | 11.78%     | 12.49%      |
>   | **Value Questionnaire**             | 4.24%             | 5.41%              | 6.73%       | 3.06%     | -1.88%        | 13.49%     | 5.34%      | 6.72%       |
>   | **Open-ended Question**             | 62.12%            | 69.82%             | 93.44%      | 60.42%    | 92.86%        | 94.45%     | 89.63%     | 73.83%      |
>
> These findings confirm that our major conclusion, "CAReDiO consistently and significantly enhances cultural alignment", is robust and not driven by high-variance artifacts in specific cultures or question types.
>
> The full experimental details and result analysis have been added to Appendix F.4.
>
> ----
>
> **We hope our clarifications and additional experimental results address your concerns, and we are happy to respond to any further questions.**
>
> **We would sincerely appreciate it if you could read our responses and kindly reconsider the assessment of our work.**

---

### Official Review · Reviewer_g389 · 2025-11-03

**Soundness:** 3
**Presentation:** 3
**Contribution:** 2
**Rating:** 6
**Confidence:** 4

**Summary:**

The paper proposes CAReDiO, an in-context data optimization framework for cultural alignment. It alternates between generating and refining culture-sensitive questions and answers with two information-theoretic objectives: an information-gain objective (representativeness) and a culture-divergence objective (distinctiveness). Using CAReDiO, the authors build CARDSet for 15 cultures and report that small-scale fine-tuning can improve cultural benchmarks across multiple backbones. The framework itself comprises 1) information gain to reduce a model’s cultural uncertainty, 2) divergence to separate target from non-target cultures, and 3) an iterative refinement loop on data questions and responses. The paper evaluates on GlobalOpinionQA and WVS-style setups among others, and compares across a range of cultural benchmarks.

**Strengths:**

S1. The authors present a clear problem framing with two concrete challenges. The paper explicitly separates data quality into representativeness and distinctiveness and ties each to a learning objective, which is conceptually helpful.

S2. The defined objectives are grounded well with theory. Representativeness is operationalized as mutual information and distinctiveness is linked to maximizing a lower bound on generalized Jensen-Shannon divergence across cultures under classifier accuracy and non-overconfidence assumptions. The use of JS echoes prior robustness work employing multi-distribution JS for consistency.

S3. CAREDIO is itself a practical, model-agnostic pipeline. The alternating refinement loop is simple to implement with existing LLMs and can leverage either the target backbone or a larger assistant model to synthesize data.

S4. The new method is evaluated well across a breadth of baselines and benchmarks. The paper situates results against multiple cultural datasets (CB/Prism/GOQA/WVS) covering a range of tasks (multiple choice/survey/open-ended) and methods (CultureX, Role-Play).

S5. There are some risks of circularity due to the use of LLMs as judges. Both objectives estimate culture labels or divergences using LLM-based classifiers, which can encode the same cultural priors the method hopes to correct. Without strong human-grounded calibration, the approach risks amplifying pre-existing biases in the assisting model. The authors mostly mitigate this risk through the use of existing well-developed human preference datasets like PRISM and global value surveys like WVS and the further human evaluations on the final outputs of the CARDSet dataset and the final alignment step showing strong validation of the methods.

**Weaknesses:**

W1. Distinctiveness theory depends on strong, partly unverifiable conditions. Proposition 2 connects the learning objective to a lower bound on GJS only if a culture-membership classifier is sufficiently accurate and not over-confident. The paper does not clearly demonstrate these premises hold across cultures or provide diagnostics of the error bounds in practice.

W2. The benchmark construction offers some leakage concerns. Several compared datasets are built or augmented from WVS or related sources, and the evaluation also uses WVS/GlobalOpinionQA. This creates potential data leakage or style overlap that can inflate gains; the paper mentions the datasets but does not rigorously audit overlap or de-duplication. Durmus et al (2024) show how easy it is for LLMs to overfit survey-style probing.

W3. Another weakness of this work is that the technical novelty relative to prior synthetic pipelines is incremental. CulturePark also uses multi-agent LLMs to generate cross-cultural dialogues, CultureLLM uses WVS to seed augmentation, and PRISM and CulturalBench collect diverse human feedback to ground value statements. CAReDiO’s main novelty is the particular objective pairing, which is interesting but not obviously transformative without stronger ablations.

**Questions:**

Q1. Could the authors provide more details on how the role-playing baseline is developed. In section 3 a number of different types of role-playing are described but it is not clear how they are developed or what the final approach is.

Q2. Did the authors consider ablations on the independent optimization objectives of representativeness and distinctiveness?

Q3. The paper mentions that the english-dominant nature of training corpuses can bias a model towards western cultures. Was any investigation done into the impact of language improving cultural alignment?

---

> ### Author Response · Authors · 2025-11-25
> **Response to Reviewer g389 [1/3]**
>
> Thank you very much for your supportive review and valuable suggestions. Our responses to each Weakness (W) and Question (Q) are as follows. A revision of our paper will be uploaded shortly, with the revised parts marked in blue.
>
> ### **W1: The paper should demonstrate the conditions in Proposition 2 hold across cultures or analyze the error bound in practice.**
>
> We clarify this issue from two perspectives:
>
> (1) **Theoretical Perspective**. The main purpose of Proposition 2 is to i) *provide an interpretation of Eq.(4)*, showing that our method truly exploits the differences between real cultural distributions, and ii) *guarantee our approximated implementation* (Sec.3.3) indeed *optimizes this objective*. **These conditions only affect how tight the error bound is and do NOT affect whether Proposition 2 holds**.
>
> (2) **Empirical Evidence**. Following the suggestion, we added an experiment in Appendix C.2 to show that **the error bound is tight enough in practice**. We construct an evaluation set of $(x, y, p*, c)$ with $(x, y)$ sampled from our synthetic data and $p*$ annotated by three native speakers, covering four cultures (200 samples for each). For each culture, we calculate the classifier error $\epsilon$ as the average of $|\phi(x,y) – p*|$, and the classifier confidence $\eta$ as the max value of $\phi(x,y)$, across the 200 samples.
>
> As shown in the following table, **the approximation error is smaller than 0.05 for most countries**. This is very small compared to the typical range of GJS, which spans $[0, logK]$ for $K+1$ distributions ($K=14$ in this work, and the maximum of GJS is 2.64). Thus, **the approximation is sufficiently accurate for practical use**.
>
> | **culture** | **United States** | **China** | **Japan** | **Poland** |
> |-------------|-------------------|-----------|-----------|------------|
> | **$\epsilon$** | 0.1682            | 0.1803    | 0.0973    | 0.0964     |
> | **$\eta$**     | 0.9519            | 0.9559    | 0.9304    | 0.9675     |
> | **error**   | 0.0160            | 0.0330    | 0.0290    | 0.0485     |
>
> Besides, the efficacy of Eq.(4) (distinctiveness optimization) has also been empirically verified by **the strong overall performance** of CAReDiO across all benchmarks, further **supporting that Proposition 2 holds in practice.**
>
> ---
>
> ### **W2: The benchmark construction offers some leakage concerns. Several compared datasets are built or augmented from WVS or related sources, and the evaluation also uses WVS/GlobalOpinionQA.**
>
> We acknowledged and explicitly mentioned the potential leakage in the submission version (L412-414) but still included WVS data in evaluation for two reasons:
>
> 1. **Comparing the impact of leakage by WVS**. Several baselines, e.g., CultureLLM, CulturePark, CultureSPA, are structurally integrated with WVS, and it's infeasible to remove WVS-related data without fundamentally altering these baselines. **Note that CAReDiO uses only demographics in WVS, not its questions or responses**. As discussed in Sec.4.3 (L484-485), we keep WVS in evaluation to show baselines' limitations, which perform well only on these two survey-style benchmarks but fail on other OOD ones, indicating potential leakage and overfitting.
>
> 2. **Comparing the generalization capability of different methods**. To mitigate the possible leakage from WVS , we take two actions: (i) **We remove the WVS split in GlobalOpinionQA** and only keep the GAS subset for evaluation (L356-357). (ii) **We intentionally introduce two additional benchmarks in different formats** for comparison. As shown in Table 3, our method achieves significantly larger gains on these non-WVS benchmarks than on WVS/GlobalOpinionQA, while baselines, especially the three rooted in WVS, only perform well on WVS (on non-WVS sets, even worse than Role-Play). This demonstrates CAReDiO's improvements are not driven by leakage and supports its robustness and generality.
>
> We included this discussion on data leakage in Appendix  D.6 of the revised pdf to further support our experimental design.

---

> ### Author Response · Authors · 2025-11-25
> **Response to Reviewer g389 [2/3]**
>
> ### **W3: The technical novelty relative to prior synthetic pipelines is incremental. CAReDiO’s main novelty is the particular objective pairing, which is interesting but not obviously transformative without stronger ablations.**
>
> We clarify this issue from the following three perspectives.
>
> (1) **Clarification of our novelty and contribution.** As claimed in Line 91-96, our technical novelty is three-fold:
> + *A novel perspective for modeling culture through representativeness and distinctiveness*.
> + *A new cultural data synthesis method* (Algorithm 1), whose novelty does NOT lie in seed augmentation or multi-agent LLM (these are just fundamental ingredients like Transformer or GPT), but lies in the specific information-theoretic methods. For "multi-agent dialogue extraction", we introduce *Eq. (3), a novel information-gain-based consensus elicitation method*; for "measuring cultural differences", we propose *Eq. (4), a new score function for distinctiveness optimization*.
> + *Theoretical contributions*. Different from most heuristic or engineering-driven frameworks for cultural data synthesis, CAReDiO is grounded in theoretical guarantees for its effectiveness. These theoretical analyses also contribute significantly, especially for ICLR/ICML.
>
> (2) **Empirical validation of effectiveness.** The extensive comparisons with popular synthetic pipelines (in Table 3), e.g., CultureLLM, CulturePark and CultureSPA, have empirically validated the superiority of our novel designs (i.e., Eq.(3) and Eq.(4)).
>
> **We also added an ablation study to justify our method**. We provided full details and results in Appendix F.2, and summarize the key findings here:
>
> | **Method**              | **CB-Easy** | **CB-Hard** | **Prism** | **GlobalOpinion** | **Avg** |
> |-------------------------|-------------|-------------|-----------|-------------------|---------|
> | **Qwen2.5-7B-Instruct** | 72.01       | 38.90       | 42.06     | 53.28             | 51.56   |
> | **Role-Play**           | 72.38       | 36.73       | 67.28     | 55.83             | 58.05   |
> | **only Rep**            | **74.53**   | 38.70       | 69.56     | 54.94             | 59.43   |
> | **only Dist**           | 72.18       | 38.14       | 68.72     | 54.95             | 58.50   |
> | **CAReDiO**             | 73.48       | **40.20**   | **77.42** | **56.23**         | **61.83**   |
>
> As shown in the Table, **the representativeness** (only Rep, Eq.(3)) and **distinctiveness** (only Dist, Eq.(4)) **objectives can each yield certain improvements** compared to simple Role-play, particularly significant on the OOD CB-Hard and Prism, while combining both achieves the best performance.
>
> (3) **Scope of comparison and novelty.** Our study focuses on the task of cultural alignment and proposes a novel theoretical framework for cultural data optimization. Techniques like multi-agent data synthesis, seed augmentation, and diverse human feedback collection are orthogonal research fields that require dedicated studies. Our technical novelty mainly lies in the designed scoring function (Eq.(2),(3),(4)) and the alternative refinement framework (Algorithm 1). Incorporating multi-agent and feedback is less central to novelty.

---

> ### Author Response · Authors · 2025-11-25
> **Response to Reviewer g389 [3/3]**
>
> #### **Q1: Please provide more details on the role-playing baselines and the role-playing developed in Section 3.**
>
> Apologies for missing the details of the role-playing baseline. We instruct LLMs to simulate individuals from specific cultural backgrounds with the system prompt (added to Appendix D.4)
>
>     "You are a {country} chatbot that know {country} culture very well. Please answer the following questions according to your knowledge about the culture of {country}."
>
> The role-playing used in Eq.(3), Sec.3.3, has been provided in Appendix B.2 (Fig.6,7,8) of the initial version (currently Fig 7,8,9,10 in the revised pdf). In summary, we provide system messages to role-play three types of individuals, including:
> + **General people with various demographics sampled from WVS**. The system message is "*Now you are a {country} citizen with the following demographic profile: {demographic_info} You understand {country} culture based on your own experiences and viewpoints as this persona, including the cultural values, beliefs, and social practices. Your task is …*"
> + **Cultural experts**, such as sociologists and cultural psychologists, the system prompt is *"You are a {country} culture expert, with 15 years of experience as a {role}. Thus, you have deep knowledge of {country} culture values, traditions and social practices. Your task is to …  from your own viewpoints as this expert"*
> + **Cross-cultural researchers** to introduce cultural knowledge from an external perspective. The system prompt is *"You are a {foreign} culture expert, with 15 years of experience as a cross-cultural researcher. Thus, you have deep knowledge of cultural values, beliefs, and social practices of various cultures across the world. Your task is to … from your own viewpoints as this foreign culture expert".*
>
> We also included these details in Appendix B.2 of the revised pdf to enhance reproducibility.
>
> ---
>
> #### **Q2: Ablations on independent optimization objectives of representativeness and distinctiveness.**
>
> Following the valuable suggestion, we added an ablation study on the Representativeness (Rep) and Distinctiveness (Dist) objectives. Please refer to **our response to Weakness 3 (W3)** and Appendix. F.2 in our updated PDF for more details. The key findings are as below:
>
> + **Both objectives can independently improve the performance significantly**. Only Rep + 15.3% averaged on benchmarks, only Dist + 13.5%, **while their combination (CAReDiO) yields the best** (+20%), compared to the backbone LLMs.
> + **Optimizing representativeness alone outperforms distinctiveness alone**, indicating current LLMs contain biased cultural knowledge, while Eq.(3) more effectively mitigates them. Adding Dist on top of Rep brings additional gains, suggesting that Eq.(4) helps better identify cultural boundaries and resolve ambiguity.
>
> ---
>
> #### **Q3. Was any investigation done into the impact of language on improving cultural alignment?**
>
> (1) We mentioned the English corpus (L38) not for highlighting language, but emphasizing that disproportionately Western, English-speaker–created pretraining data induces cultural imbalance and the dominance of Western culture. This is a widespread phenomenon observed in culture-related studies [1,2,3]. However, we did not intend to imply that the English language itself is the source of the bias.
>
> (2) The impact of language for cultural alignment is indeed an important research question, and there have  already been several studies around it [4,5]. This is orthogonal to our focus and out of this work's scope. We leave it to future work.
>
> In the revised pdf, we further clarified it in Sec.1 and discussed potential language-related issues in the Limitations section (Appendix G).
>
> ---
>
> **We hope our responses, additional results, and the revised paper could address your concerns. We are more than willing to respond to any further questions regarding our methods and experiments.**
>
> **We would sincerely appreciate it if you could review our responses and kindly reconsider the assessment of our work.**
>
> ---
>
> ### Reference
>
> [1] Naous et al., Having Beer after Prayer? Measuring Cultural Bias in Large Language Models. AC
>
> [2] Masoud et al.,Cultural Alignment in Large Language Models: An Explanatory Analysis Based on Hofstede’s Cultural Dimensions. COLING 2025.
>
> [3] Durmus et al., Towards Measuring the Representation of Subjective Global Opinions in Language Models. COLM 2024.
>
> [4] Mukherjee et al., Global Gallery: The Fine Art of Painting Culture Portraits through Multilingual Instruction Tuning. NAACL 2024.
>
> [5] Choenni et al., The Echoes of Multilinguality: Tracing Cultural Value Shifts during Language Model Fine-tuning. ACL 2024.

---

### Official Review · Reviewer_UgnK · 2025-11-04

**Soundness:** 3
**Presentation:** 2
**Contribution:** 3
**Rating:** 6
**Confidence:** 4

**Summary:**

The paper proposes a framework for dataset construction for cultural alignment, where it aims to optimize for two objectives (representativeness and distinctiveness). The framework entails generating cultural QA pairs, scored for whether the QA pair is representative of that culture through consensus of multiple LLMs and whether it is distinct from other cultural QA pairs by measuring JS divergence between the answer distributions. The authors theoretically ground their objective in cultural theories and create a dataset for 15 cultures, showing that fine-tuning on data generated using the proposed framework leads to better performance.

**Strengths:**

- Both objectives seem sound, are well motivated, and grounded in theory
- The proposed approach shows moderate improvements compared to the baselines
- The writing is clear
- In depth details and fine-grained results are provided in the Appendix

**Weaknesses:**

- For the representativeness optimization objective, a fundamental assumption behind the consensus elicitation approach is that multiple LLMs with the right conditioning can simulate a group of individuals from a target culture c. This isn’t obvious to me and would be something that needs empirical experiments to prove its efficacy, persona based prompting for survey simulation is still a research field. Thus, repeatedly using the Cultural Consensus theory for grounding seems a bit of a stretch.
- The weaknesses or limitations of the paper are not discussed in sufficient detail. For instance, since all parts of the pipeline depend on LLMs already knowing or inferring something about the culture, a fundamental limitation would be the approach not working for cultures not well represented in current LLMs. This is a core limitation that should be discussed in the paper.
- Presentation could be improved
  - A lot of the space in the paper is given to outlining the framework and grounding it in theory. I appreciate the effort the authors put into grounding the approach but this results in framework becoming concrete much later in the paper, which is worse for readability.
  - Several important details necessary for understanding the paper in depth are in the Appendix.
  - Figure 2 is not visually representative of the framework, the reader isn’t walked through the Figure in text. Clear examples or a better figure would aid the reader in clearly understanding the iterative data generation and optimization process.

**Questions:**

- Minor:
  - WVS is cited with multiple references in different parts of the paper (Xu et al., AlKhamessi et al., Tao et al.), none of which correspond to the actual reference: Haerpfer et al. (2022).
- Suggestion:
  - Could shorten the framework introduction and current motivation, propositions which are quite verbose, make the framework concrete earlier in the paper with the dataset, add a discussion section at the end.

---

> ### Author Response · Authors · 2025-11-25
> **Response to Reviewer UgnK [1/3]**
>
> Thank you very much for your supportive review and valuable suggestions. Our responses to each Weaknesses (W) and Questions (Q) are as follows. A revision of our paper has been uploaded, with the revised parts marked in blue.
>
> ### **W1. As the base of consensus elicitation with Cultural Consensus Theory, multiple LLMs can simulate a group of individuals from a target culture, is not obvious and requires empirical experiments to prove its efficacy.**
>
> We clarify both the reasonability and effectiveness of introducing multiple LLM-simulated culture individuals to elicit more representative items from three points.
>
> (1) **The effectiveness of prompting LLMs to simulate culture individual has been widely validated**. Known as *In-Context Alignment (ICA)*, conditioning LLMs with attributes, such as role, persona and culture, has been a method widely adopted in cultural alignment and shown the efficacy [1,2,3,4]. Moreover, our paper also demonstrates that the Role-Play baseline (in Table 3) significantly improves the original base model (57.75 $\rightarrow$ 68.62 on Gemma-3-27B-IT), showing *ICA indee enhances LLMs' cultural competence*.
>
> (2) **Further enhancement by multiple role-playing LLMs**. The core motivation behind consensus elicitation (Eq.(3)) is that one single LLM may encode incomplete and biased knowledge of the target culture. Multiple LLMs prompted with different personas can help extract cultural knowledge from diverse perspectives, mitigating bias.
>
> We empirically verify this advantage by *assessing multiple role-playing LLMs and their consensus on cultural benchmarks*. We also *compare the final alignment performance of using a single or multiple individuals in representativeness optimization*. We add full experiments in Appendix C.1, F.5 and summarize key results below:
>
> + ***Different roles show divergent cultural perceptions***. We evaluated different LLM-roles on GlobalOpinionsQA, and measure the agreement among of individual roles below:
>
>     | **Culture** | **United States** | **Germany** |  **Japan** | **Russia** | **Poland** | **Mexico** | **Nigeria** |
>     |-------------|-------------------|-------------|-----------|---------------|------------|------------|-------------|
>     | **Agreement of Roles**    | 0.86              | 0.85      | 0.81      | 0.8        | 0.81       | 0.79       | 0.8         |
>
>     We observe obvious differences in their outputs. This indicates incorporating multiple personas helps activate diverse cultural knowledge.
>
> + ***Consensus of multiple roles further improves stability and accuracy***. We compare the performance between single role and multiple (n=7) roles below:
>
> | **Method** | **United States** | **Germany** | **Japan** | **Russia** | **Poland** | **Mexico** | **Nigeria** | **Avg**$\uparrow$ | **Std** $\downarrow$ |
> |------------|-------------------|-------------|-----------|------------|------------|------------|-------------|---------|---------|
> | **Single** | 64.50              | 62.81       | 56.47     | 56.65      | 59.71      | 59.40      | 58.84       | 59.77   | 2.76 |
> | **Multi**  | **65.17**          | **64.99**   | **58.24** | **59.21**  | **62.63**  | **60.45**  | **59.94**   | **61.52**| **2.57** |
>
> Obviously, **aggregating multiple roles yields better and more stable (lower std across cultures) results**, with especially larger gains in low-resource settings, e.g., Russia and Poland, where model biases are more prominent.
>
> + ***Consensus elicited by multiple individuals further enhances cultural alignment.*** We compare the performance of CAReDiO using different numbers of individuals (n=1, 10, 23) in representativeness optimization below:
>
> | **Dataset** | **Avg** |**CB-Easy** | **CB-Hard** | **Prism** | **GlobalOpinionQA** | **WVS** |
> |-----------------|-------------|-------------|-----------|---------------------|---------|---------|
> | **n=1** | 60.39  | 72.44  | 42.61      | 3.61    | 53.57              | 61.12 |
> | **n=10** | *61.23*   | *72.96*  | *43.94*      | *3.65*    | *54.28*              | *62.04* |
> | **n=23** | **63.76**  | **75.38**      | **45.22**      | **3.92**    | **55.57**   | **64.09**  |
>
> We can see that **data produced by more LLM-simulated individuals yield better cultural alignment** *(63.7 (n=23) > 61.23 (n=10) > 60.39
> (n=1))*, indicating diverse voters identify more accurate consensus.
>
> **These results collectively demonstrate the effectiveness and reliability of consensus elicitation (CCT) from multiple LLMs.**

---

> ### Author Response · Authors · 2025-11-25
> **Response to Reviewer UgnK [2/3]**
>
> (3) **CAReDiO does not rely on perfect persona simulation**. We recognize precisely simulating individuals is an unresolved challenge, but it is **NOT** our purpose. Instead, CCT only requires agents with diverse background and sufficient *cultural competence* (L198). As specified by Eq.(2) and Eq.(3), our consensus elicitation method leverages such diverse, culturally conditioned agents to identify shared cultural knowledge. The experiments in Table 3 and Appendix F.5 show that such approximated persona playing is sufficient to obtain cultural improvement.
>
> We add the full implementation details, experimental settings and results analysis in Appendix C.1, F.5.
>
> ### **W2. The weaknesses or limitations are not discussed sufficiently, especially for cultures not well represented in current LLMs.**
>
> **We have already discussed these limitations in Appendix F of the initial version**, including i) inapplicability to low-resource cultures, (ii) limited number of cultures, (iii) the potential neglect of non-mainstream subcultures, and (iv) the need to further extend alignment methods.
>
> Note that handling low-resource cultures is a critical challenge for all synthetic approaches [5,6], we never claimed CAReDiO can fully address it. Instead, our method is specially designed to elicit more representative and distinctive cultural knowledge from LLMs, which helps to mitigate this challenge to some extent. We conduct experiments on Indonesia and Nigeria and observe improvements (see Table 10-14 in Appendix F.1 of the revision), which empirically show our method mitigates, though does not eliminate, this limitation.
>
> We added more discussions and potential future directions of this issue in Appendix G of the revised version.
>
> ---
>
> ### **W3-1, Q2: Too much space for outlining theoretical framework before making the framework concrete, affecting readability.**
>
> Thanks for your constructive suggestions. We presented the full mathematical design of CAReDiO before instantiating it, since we aimed to highlight its fundamental nature and generality. We acknowledge this may hurt readability and clarity.
>
> **To further improve the readability of our work, we've conducted a comprehensive review and polishing**. Key changes include:
> + We **reorganized Sec.3.2 and Sec.3.3**, presented each variable and module with concrete examples, and introduced their implementations earlier.
> + We added a **comprehensive notation table** of all symbols used in our mathematical expressions in Appendix A.
> + We refined Fig.2, including concrete notations and examples to better align it with the description.
> + We polished Sec.3.3 and made it focus mainly on specifying hyperparameters, the backbone LLMs, and some configurations to create a cultural dataset as an instantiation.
>
> Details can be referred to in the revised version.
>
> ### **W3-2: Several details necessary for understanding the paper in depth are in the Appendix.**
>
> Due to strict page limits, we have to place most of the details in Appendix, e.g., mathematical derivations, cultural topic list to construct the dataset, implementation details, and additional supportive results. **All of these have been explicitly referenced in the main text** to ensure smooth navigation.
>
> To further help readers' understanding, in the revised pdf, we: i) added some key implementation details in Sec.3.2, ii) presented concrete examples alongside the mathematical expressions, and ii) further polished Fig.2.
>
> Let us know if you have any specific concerns about missing details; we would be more than happy to address them directly.
>
> ### **W3-3: Figure 2 is not visually representative of the framework; the reader isn’t walked through the Figure in text.**
>
> Thank you for the valuable suggestion. We have provided an improved version of Fig.2 in the revised pdf, in which we (i) explicitly highlight the iterative data optimization process, (ii) include concrete notations and examples to better align the figure with the textual description, and (iii) add more descriptive guidance, marks, and a detailed caption to help readers follow the workflow directly from the figure.
>
> ----
>
> We believe the polishing improves the paper's overall readability, and these easy-to-fix presentation issues (W3) would not impede our core contributions of our novel method, empirical and theoretical results.

---

> ### Author Response · Authors · 2025-11-25
> **Response to Reviewer UgnK [3/3]**
>
> #### **Q1: WVS cited in the paper should correspond to the actual reference.**
>
> Thank you for pointing this out. In several places, we discuss current studies using WVS for cultural alignment or evaluation, so we intentionally cite application papers (e.g., AlKhamissi et al., 2024 in Line 103, Xu yet al. 2024a in Line 345), rather than the WVS dataset itself. For direct references to WVS appearing in Line 321 and Line 1072, we have corrected such instances to the proper citation: Haerpfer et al. (2022).
>
> ----
>
> **We hope that our responses and additional experimental results have addressed your concerns, and we are more than willing to answer any further questions.**
>
> ----
>
> ### Reference
> [1] Durmus et al., Towards Measuring the Representation of Subjective Global Opinions in Language Models. 2023.
>
> [2] Kwok et al., Evaluating Cultural Adaptability of a Large Language Model via Simulation of Synthetic Personas. 2024.
>
> [3] Kharchenko et al., How Well Do LLMs Represent Values Across Cultures? Empirical Analysis of LLM Responses Based on Hofstede Cultural Dimensions. 2025.
>
> [4] Choenni and Shutova. Self-Alignment: Improving Alignment of Cultural Values in LLMs via In-Context Learning. 2024.
>
> [5] Li et al., CulturePark: Boosting Cross-cultural Understanding in Large Language Models. NeurIPS 2024.
>
> [6] Chiu et al., CulturalBench: a Robust, Diverse and Challenging Benchmark on Measuring the (Lack of) Cultural Knowledge of LLMs. ACL 2025.

---

### Author Response · Authors · 2025-12-03
**General Response to New Acs Regarding the Main Concerns [1/2]**

We sincerely thank all the reviewers and ACs for their efforts and valuable comments. To help reduce the new AC's workload caused by the accident, we'd like to summarize the reviewers' flagged Weaknesses (W) into **six** key concerns and explain how we address them. *Our corresponding substantial clarifications and improvements have extended the updated paper by 11 full pages* (28 pages $\rightarrow$ 39 pages), with the revised parts marked in blue.

#### **1. Novelty and Contribution of CAReDiO** (Reviewers g389–W3, ai47-W1)
As CAReDiO is implemented as an LLM-empowered in-context learning (ICL) method, both reviewers questioned about its novelty and improvement beyond prior synthetic pipelines. We clarify this from three aspects:
+ We emphasized our claimed novelty and contributions (Lines 91-96 in the initial paper) are: i) **A novel, theory-grounded perspective for modeling culture via representativeness and distinctiveness**; ii) **A new synthesis method** whose novelty lies not in seed augmentation or prompting methods (fundamental ingredients), but **in two information-theoretic optimization objectives** (Eq.(2)-Eq.(5)); and iii) **theoretical analysis** (Propositions 1\&2) providing effectiveness guarantee, consistent with ICLR/ICML standards.
+ We provided *an intuitive explanation* (in Appendix F.2) that the improvement of CAReDiO is obtained by **explicitly shaping better cultural data distribution along two culture theory-motivated criteria**, i.e., *representativeness and distinctiveness*.
+ Empirical results in Table 3 shows CAReDiO's superiority over popular synthetic pipelines, and an additional ablation study (Appendix F.2) shows that representativeness and distinctiveness independently contributes measurable gains.

**These justify the novelty and improvement of CAReDiO both theoretically and empirically.**

#### 2. **Reliability of Multi-LLM Simulation for Consensus Elicitation** (Reviewer UgnK – W1)
Empirical evidence is required by the reviewer for the efficacy of using multiple LLMs to simulate a group of cultural individuals for representativeness optimization. **Following this requirement**,
+ We confirmed **the validity of cultural individual simulation** with the improvement of the Role-Play baseline over the original base model (*57.75 $\rightarrow$ 68.62 on Gemma-3-27B-IT*).
+ We empirically verified that the **consensus aggregated across $n=7$ roles outperforms the single role-play**, especially for low-resource cultures (e.g., *56.65 $\rightarrow$ 59.21 for Russia; 59.71 $\rightarrow$ 62.63 for Poland*).
+ We compared CAReDiO using different numbers of individuals (n=1, 10, 23) for optimization and showed that **multiple LLMs enhances cultural alignment** ((63.7 (n=23) > 61.23 (n=10) > 60.39(n=1)).

**These results collectively demonstrate the effectiveness and reliability of consensus elicitation from multiple LLMs.**

#### 3. **Error and Architecture Sensitivity of the Distinctiveness Classifier** (Reviewer g389 – W2, HCb7 – W2)
We added extensive experiments in *Appendix C.2, F.6* to fully diagnose the approximation error,  reliability and sensitivity of the classifier, showing that:
+ **The practical error bound is tight enough**, $<0.05$ for most countries, very small compared to typical GJS range (2.64 in our setting).
+ **Our text embedding similarity based classifier is sufficiently accurate and reliable for optimization**, with F1 > 0.9 and error $e < 0.05$.
+ We test an alternative LLM-as-judge classifier, GPT-5-thinking, observing **both classifiers achieve sufficient accuracy and show no substantial differences in downstream data synthesis and cultural alignment performance**. *CAReDiO (embedding-based) achieves 64.13 on cultural benchmarks, CAReDiO (llm-as-judge) 64.05 > 61.05 (the role-play baseline)*.

**These confirm the validity and reliability of our current classifier, as well as the robustness of our framework to this choice**.

---

> ### Author Response · Authors · 2025-12-03
> **General Response to New Acs Regarding the Main Concerns [2/2]**
>
> #### 4. **Additional Experiments on Model Validity and Reliability** (Reviewer g389 – Q2, HCb7 – W3,W4,Q3, ai47 – Q1)
> **We added extensive experiments to substantially strengthen empirical validation**. Specifically,
> + We added **an ablation study** to demonstrate the contribution of each independent optimization objective (Appendix F.2). Only representativeness $\rightarrow$ +15.3%; Only distinctiveness $\rightarrow$ +13.5%, the combination (CAReDiO) +20%.
> + We analyzed **performance among closely related cultures** (Japan / Korea / China) to showed the contributions of **distinctiveness objective**: i) for data synthesis, CAReDiO captures clear separation while the baseline exhibits cross-cultural overlap (inter-culture centroid distance 0.47 > 0.29); and ii) for alignment, CAReDiO shows larger performance gaps between the target culture and close non-target cultures ($\Delta$ 5.3 > 2.4 on GlobalopinionQA, $\Delta$ 0.31>0.27 on Prism).
> + We added **an analysis on the hyperparameter, the number of individuals set for consensus elicitation** (Appendix F.5). More LLM-simulated individuals lead to better cultural alignment (63.7 (n=23) > 61.23 (n=10) > 60.39(n=1), demonstrating the validity for our design.
> + We added **a sensitivity analysis of the distinctiveness classifier** (Appendix F.6), observing that our framework is robust to this choice: CAReDiO (embedding-based) achieves 64.13 on cultural benchmarks, CAReDiO (llm-as-judge) 64.05 > 61.05 (the role-play baseline)
> + We reported detailed inter-annotator agreement of human evaluation, with the average 0.85 indicating **strong reliability for our subjective evaluations**.
> + We analyzed **CAReDiO's gains across different cultures and question types**, showing that **CAReDiO consistently improves alignment** for almost all cultures (100% on Prism, GlobalOpinionQA and WVS, 73.3% on CultureBench) and for diverse question types).
>
> **These additional experiments significantly enhance the validity, reliability and robustness of our method**.
>
> #### 5. **Potential Benchmark Leakage in Baselines** (Reviewer g389-W2)
> We explicitly acknowledged the potential WVS data leakage in the initial submission (L412-414) and clarified the reasons of including WVS for evaluation in rebuttal (Appendix D.6 in the revised version):
> + **Keeping WVS evaluation helps reveal baselines' limitations**. Since some baselines are inherently tied to WVS and perform well only on WVS-like benchmarks but weak on other OOD ones, indicates potential leakage and overfitting, while **our method does not use WVS data**.
> + We have taken two actions to mitigate the impacts: i) removing the WVS split from GlobalOpinionQA and ii) adding two additional benchmarks in different formats, where CAReDiO achieves significantly larger gains.
>
> Thus, **we are fully aware of the leakage issue, leverage it to reveal baseline weaknesses, and mitigate it to ensure reliability of the final conclusion.**
>
> #### **6. Improvements of Paper Presentation** (Reviewer UgnK- W3, g389 - Q1, HCb7 – W1, W2, Q2, ai47 – W2,W3, Q3-Q5)
> To improve the clarity and readability of our paper, **we conducted a comprehensive proofreading and polishing**, including:
> + Clarified and added details of method and experiment in Appendix B.2, D.4.
> + Polished Fig.1 and detailed the caption; Redrew Fig.2, including concrete notations and examples, to align it with the description.
> + Revised the introduction to articulate the representativeness and distinctiveness objective more clearly.
> + Reorganized Sec.3.2 and Sec.3.3, presented each variable and module with concrete examples, and introduced their implementations earlier.
> + Added more details in Appendix and highlighted them in the main text to ensure smooth navigation.
>
> **The polishment significantly improves the paper's overall readability, and transparency and reproducibility of our method.**
>
> **For other minor issues**, we i) corrected the reference of WVS (Reviewer UgnK – Q1); ii) improved formalization and clarity of theorem 1 (Reviewer ai47 – Q2); iii) added a complete notation table of all symbols in Appendix A, and *iv) added more discussions about inapplicability to low-resource cultures, potential neglect of subcultures and future directions in Appendix G*. We believe these easy-to-fix presentation issues would not impede the core contributions of our novel method, empirical and theoretical results.
>
> *Please note that Reviewer ai47 (current rating: 4) explicitly indicated willingness to raise the score in the initial comment (before the accident) if concerns were resolved, and our revision fully addresses all the points*.
>
> In summary, we believe these clarifications, revisions, expansions, and additional results further improve our work and address all the concerns. We would sincerely appreciate it if you could read the reviews and our responses, and kindly consider the fair assessment of our work.
>
> Best regards,
>
> The Authors

---

### Meta-Review · Area_Chair_2kSv · 2025-12-24

**Summary:**

The reviewers repeatedly signal a lack of support for various claims and choices in the paper, as well as some writing suggestions. The authors did make a sincere effort to response to those, but this is much to change during rebuttal and while at least now discussing some of the issues not mentioned before (why the assumptions hold and why would consensus know what culture needs).  I am still skeptical this suffices (e.g. theres an agreement number, it is quite similar across language, I would expect a large drop in cultures models are weak at, how do we know they don't agree on wrong things that we try to measure etc.)

**Reviewer Concerns:**

Reviewers had multiple concerns about the writing of the paper (while direct issues where tackled, probably others where not explicitly stated by reviewers and are worth tackling).
Reviewers also discussed the assumptions and the supporting evidence behind why the method is expected to work.

**Reviewer Scores:**

That is not a fair, relevant or meaningful question. I protest the way this was all handled.
A Reviewers are not here, and ToM is weak, at least mine and the one literature study. I will not try to predict people.
B Scores are, anyway, a weak signal of interest; a paper should not be accepted or rejected just based on it. An AC's job is to look at the specific weaknesses and translate them into a recommendation. Here for examples the scores are quite high, but the issues reviewers had merrit more revisions.
C There are about 100 pages of discussions for me to read overall, in addition to the discussions I monitored and were just replaced, this is beyond my personal ability to do fairly. I did my best effort.


One of the reviews was misplaced, but they can't correct it now...

---

### Decision · Program_Chairs · 2026-01-26

Reject